# Beyond Deep Heuristics: A Principled and Interpretable Orbit-Based Learning Framework

## Abstract

We introduce *Weighted Backward Shift Neural Networks* (WBSNNs), a general-purpose learning paradigm that works across modalities and tasks without per-dataset customization and replaces stacked nonlinearities with structured *orbit dynamics*. WBSNNs comprise a purely linear, operator-theoretic stage that constructs an orbit dictionary that exactly interpolates selected anchors, thereby yielding a faithful geometric scaffold of the dataset, and subsequent predictions reuse this scaffold for generalization by forming data-dependent linear combinations of these orbits—making the model inherently interpretable, as each prediction follows explicit orbit paths on this scaffold, tied to a small, structured subset of the data. While the architecture is built entirely from linear operators, its predictions are nonlinear—emerging from the selection and reweighting of orbit elements rather than deep activation stacks. We further extend this exact-interpolation guarantee to infinite-dimensional sequence Banach spaces (e.g., $\ell^p$, $c_0$), positioning WBSNNs as suitable for operator-learning problems in these spaces. WBSNNs demonstrate robust generalization; we provide a formal proof that their structure induces implicit regularization in stable dynamical regimes—regimes which, in most of our experiments, emerged automatically without any explicit penalty or constraint on the core orbit dynamics—and consistently match or outperform strong baselines across different tasks involving nontrivial manifolds, high-dimensional speech and text classification, sensor drift, air pollution forecasting, financial time series, image recognition with compressed features, distribution shift and noisy low-dimensional signals—often using as little as $1\%$ of the training data to build the orbit dictionary. Despite its mathematical depth, the framework is computationally lightweight and requires minimal engineering to achieve competitive results, particularly in noisy low-dimensional regimes. These findings position WBSNNs as a highly interpretable, data-efficient, noise-robust, and topology-aware alternative to conventional neural architectures.

## 1 Introduction

Deep learning excels across vision, language, and time series, but its practical success still leans on stacked nonlinearities, heavy overparameterization, and empirical tricks (e.g., attention), which obscures mechanism and often demands large labeled datasets and careful tuning. We ask whether a model built around simple, structured orbit dynamics can both explain and perform. We introduce Weighted Backward Shift Neural Networks (WBSNNs)—an orbit-based architecture rooted in operator theory. WBSNNs replace deep activation stacks with orbits of weighted backward shifts: first, the model builds an orbit dictionary and achieves exact interpolation of selected anchor subsets using linear operators only; then a lightweight MLP forms data-dependent linear combinations of orbit elements to generalize. Although the components are linear, the predictions are nonlinear in the input through the combinatorics of orbits, yielding a model that is interpretable (every prediction decomposes into explicit orbit paths), data-efficient, and topology-aware.

A distinctive property of WBSNN is its general-purpose design: the same orbit-based mechanism extracts structure and generalizes across varied, demanding conditions—without bespoke per-dataset pipelines. To evaluate WBSNN's versatility and inductive capacity, we test it across a wide

spectrum of datasets spanning multiple modalities, structural challenges, and learning objectives. This includes distribution shift, financial time series (including temporally indexed data with volatility and long-range dependencies), air-pollution forecasting, high-dimensional speech and text classification, synthetic low-dimensional regimes with noise, compressed-feature image recognition, and geometrically structured manifolds with both synthetic and natural noise: WBSNN matches or outperforms strong baselines while often using as little as $1\%$ of the training data to build the orbit dictionary, offering a lightweight alternative to attention-based deep networks amid growing environmental concerns over the compute and energy demands of modern AI systems. We prove that the architecture induces an implicit regularization mechanism (Lemma 3.10) and we provide an exact-interpolation theorem for finite dimensional spaces (Theorem 3.7 below), that explains the observed data efficiency.

While all experiments are finite-dimensional, our theory is not. We also prove an exact interpolation theorem in infinite-dimensional sequence Banach spaces (e.g., $\ell^p$, $c_0$): see Theorem A.1, Appendix A. This establishes that WBSNN's orbit-based representation is well-posed beyond finite dimensional spaces. This establishes a mathematically rigorous foundation for applications such as operator learning for PDE-governed systems, weather–climate forecasting on space–time fields, medical imaging (e.g., CT/MRI inverse problems), control and reinforcement learning for distributed PDE dynamics, continuous-time signal processing, neuroscience, and implicit neural fields (e.g., radiance or level-set representations), where functions or operators are the core objects.

At a holistic, system-level view, WBSNN invites a new perspective: datasets need not be arbitrary collections of points but may admit a canonical orbit-based decomposition—a geometric signature—emerging from an optimal WBSNN configuration. In this view, WBSNN is not only a learning architecture but also a functor, translating datasets into structured geometric representations whose topological and dynamical properties may encode generalization. This suggests a principled framework for comparing datasets and understanding learning via orbit geometry and category-theoretic structure.

## 2 BACKGROUND

We acknowledge foundational texts such as (2) and (3) that have shaped the operator-theoretic context of this work. Our construction is rooted in the theory of linear dynamics, where the weighted backward shift operator plays a foundational role. In infinite-dimensional separable sequence Banach spaces, this operator acts on sequences $x = (x_0, x_1, x_2, \dots)$ by shifting coordinates backward and scaling by a bounded weight sequence $w = (w_i)_{i \geq 0}$:

$$B_w x = (w_1 x_1, w_2 x_2, w_3 x_3, \dots). \tag{1}$$

This simple mechanism gives rise to surprisingly rich dynamical behavior—such as dense orbits and hypercyclicity—making it a central object of study in the theory of linear dynamics in infinite-dimensional spaces. To adapt this dynamic to finite-dimensional settings, we introduce the Weighted Backward Shift Neural Network (WBSNN), a structured transformation that cyclically reintroduces lost input components to preserve richness. While the operators remain linear, our model's predictions become nonlinear due to their dependence on orbit evolution, offering both interpretability and expressive capacity. This shift-based structure is used as the backbone for a three-phase learning architecture that blends operator theory with modern neural design.

**WBSNN in an Operator-Theoretic Machine Learning Context.** Operator theory has shaped machine learning in diverse ways, primarily through Koopman operator frameworks, which linearize nonlinear dynamical systems via spectral decompositions (often approximated in practice via DMD or neural networks) for tasks like prediction and control (9), (8), (4). Extensions such as Koopman autoencoders further embed states into linear subspaces for generative modeling (1). Another class of models, such as DeepONet (7) and neural integral equations (11), learns mappings between infinite-dimensional function spaces, often for PDEs, using architectures like branch–trunk networks for DeepONet or attention-based solvers for neural integral equations. In contrast, models like Linformer (10) proposing a low-rank projection to reduce self-attention complexity from quadratic to linear, while NAIS-Net (5) enforces stability through non-autonomous differential equation dynamics. Collectively, these methods prioritize universal operator approximation or architectural constraints for efficiency and stability, yet they are not orbit-based and thus do not build representations

through iterates of dynamic operators. These models serve as baselines in our experiments, against which WBSNN demonstrates superior data efficiency and noise robustness across diverse tasks.

In contrast, WBSNNs are not based on approximating nonlinear operators or constraining spectra for stability. Instead, they are rooted directly in dynamics of linear operators: the study of how iterates of a single operator generate rich geometric and topological behavior. A central object here is the hypercyclic operator—one admitting a vector whose orbit under iteration is dense in the space—with the weighted backward shift as the archetypal example, see (2). At first glance this notion seems exotic, since proving hypercyclicity often requires delicate constructions. Yet the key discovery is that hypercyclicity is not exceptional but typical: once one dense orbit exists, many do—and even more surprising, the set of hypercyclic operators living on any infinite-dimensional separable Banach space is itself topologically large, see Theorem 1.2, Proposition 2.20 and Corollary 2.9 (2). Motivated by these striking and unexpected facts, linear dynamics has become a crossroads where seemingly unrelated areas of mathematics converge, revealing a rich source of insights. WBSNNs borrow this spirit, grounding representation learning in orbit dynamics to bring interpretability and expressive power into machine learning. By harnessing linear dynamics, WBSNN achieves data-efficient, noise-robust generalization—often with as low as $1\%$ of training data—while maintaining exact interpolation and topological awareness, outperforming baselines in diverse regimes from manifolds to noisy signals.

## 3 WBSNN

**Weighted Backward Shift Neural Network's Backbone.** WBSNN is a three-phase learning framework introduced in Definition 3.9. At its core lies a structured, shift-based architecture $W^{(L)}$ (defined below), which cyclically shifts and scales input features across layers. Unlike traditional feedforward networks, its modulo-$d$ design preserves all input dimensions throughout depth, enabling persistent feature recombination. This recurrence simulates orbit dynamics from infinite-dimensional operator theory, where iterated linear actions generate structured, dense trajectories. With only $d$ parameters, $W^{(L)}$ builds global representations from local shifts, providing an interpretable and expressive backbone for interpolation and generalization.

**Definition 3.1.** $W^{(L)}$ *is a layered linear model of depth $L$ with $d$ predictors satisfying the following: given a layer's activations $x^{(l)} = (x_n^{(l)})$ with $0 \leq l \leq L$ and $0 \leq n \leq d - 1$, the activations at the next layer $l + 1$ follow the recurrence:*

$$x_{i \bmod d}^{(l+1)} = w_{(i+1) \bmod d} \cdot x_{(i+1) \bmod d}^{(l)} \quad \text{for } 0 \leq l \leq L - 1.$$

*where $x^{(0)}$ represents the input vector, $x^{(L)}$ the output vector and $(w_n)_{0 \leq n \leq d-1}$ is a weight finite sequence.*

We refer the reader to Figure 1 in the Appendix B for a visual illustration of how $W^{(L)}$ operates in the case $d = 5$ and $L = 11$.

Given $L$, we extend $x^{(0)}$ by appending its first $L$ entries wrapped modulo $d$, and denote the result by $x_{\text{ext},L}^{(0)} \in \mathbb{R}^{d+L}$. That is, we define $(x_{\text{ext},L}^{(0)})_i := x_{i \bmod d}^{(0)}$ for all $0 \leq i \leq d + L - 1$. Then the output of $W^{(L)}$ can be given in matricial form by

$$(W^{(L)} x_{\text{ext, L}}^{(0)})_i := \left( \prod_{k=0}^{L-1} w_{(i+k) \bmod d} \right) \cdot x_{(i+L-1) \bmod d}^{(0)} \tag{2}$$

for $1 \leq i \leq d$, where the matrix $W^{(L)} \in \mathbb{R}^{d \times (d+L)}$ is defined as:

$$W_{i,j}^{(L)} = \begin{cases} \prod_{k=0}^{L-1} w_{(i+k) \bmod \text{d}}, & \text{if } j = i + L \\ 0, & \text{otherwise} \end{cases}, \tag{3}$$

for $1 \leq i \leq d$ and $1 \leq j \leq d + L$. Each row $i = 1, \ldots, d$ of $W^{(L)}$ thus contains exactly one nonzero entry, located at column $i + L$, representing a product of $L$ weights extended modulo $d$.

**Note:** For practical visualization, the reader should observe that $W^{(L)}x_{\text{ext}}^{(0)}$ is a shift-scaling of $x_{\text{ext}}^{(0)}$, beginning at the $L$-th entry of $x_{\text{ext}}^{(0)}$ and preserving the order of components modulo $d$.

**Remark 3.2.** *Throughout the paper, in order to keep the notation simple, whenever we apply $W^{(L)}$ to a vector $x^{(0)} \in \mathbb{R}^d$, the input is implicitly extended modulo $d$ so that all required components $x_d^{(0)}, x_{d+1}^{(0)}, \ldots x_{d+L-1}^{(0)}$ are defined via $x_{d+k}^{(0)} := x_k^{(0)}$ for $k = 0, \ldots, L-1$.*

**Example 3.3.** *Let $d = 4$, $L = 2$, and weights $(w_0, w_1, w_2, w_3)$. Then for an input vector $x^{(0)} = (x_0^{(0)}, x_1^{(0)}, x_2^{(0)}, x_3^{(0)}) \in \mathbb{R}^4$, then*

$$W^{(2)}x^{(0)} = \begin{bmatrix} 0 & 0 & w_1 w_2 & 0 & 0 & 0 \\ 0 & 0 & 0 & w_2 w_3 & 0 & 0 \\ 0 & 0 & 0 & 0 & w_3 w_0 & 0 \\ 0 & 0 & 0 & 0 & 0 & w_0 w_1 \end{bmatrix} \begin{pmatrix} x_0^{(0)} \\ x_1^{(0)} \\ x_2^{(0)} \\ x_3^{(0)} \\ x_0^{(0)} \\ x_1^{(0)} \end{pmatrix} = \begin{pmatrix} w_1 w_2 x_2^{(0)} \\ w_2 w_3 x_3^{(0)} \\ w_3 w_0 x_0^{(0)} \\ w_0 w_1 x_1^{(0)} \end{pmatrix}.$$

*Here we adopted the convention outlined in Remark 3.2.*

**Remark 3.4.** *This extension modulo $d$ structure ensures that no information is lost, even for large $L$, and allows the model to propagate and recombine features across all positions. $W^{(L)}$ thus provides a structured linear backbone for learning with global receptive fields and low parameter complexity.*

**Example 3.5** (Weight regimes ($d = 3, L = 2$)). *Let $w = (c, c, c)$. Then*

$$W^{(2)}x^{(0)} = \left( c^2 x_2^{(0)}, \ c^2 x_0^{(0)}, \ c^2 x_1^{(0)} \right).$$

*Thus: $c = 0.5$ (vanishing) $\Rightarrow$ decay by $0.5^2$; $c = 1$ (neutral) $\Rightarrow$ pure permutation (no scaling); $c = 1.5$ (exploding) $\Rightarrow$ amplification by $1.5^2$. In general, the product $\prod_{k=0}^{d-1} w_k$ governs energy growth/decay, providing a simple knob for stability and expressivity.*

The following lemma reveals a key structural property of $W$ that limits the distinctiveness of large powers. (Proof deferred to Appendix A.1).

**Lemma 3.6.** *Given $W$ with $d$ predictors, for every $X \in \mathbb{R}^d$ and every $L \geq 1$, we have that $W^{(L)}X = \lambda W^{(L \bmod d)}X$, where $\lambda = \left( \prod_{k=0}^{d-1} w_k \right)^m$ with $L = md + (L \mod d)$.*

The following is our main theoretical result. It shows that, under a technical independence condition, any finite dataset admits an exact realization within a class of transformed weighted shifts. We now present and prove an explicit construction of such a realization, which will serve as the foundation for our learning algorithm. (Proof deferred to Appendix A.2).

**Theorem 3.7** (Universal Representation via Transformed Weighted Shifts). *Let $W$ on $\mathbb{R}^d$ with weights $w_n \neq 0$, and let $\{(X_i, Y_i)\}_{i=1}^N \subset \mathbb{R}^d \times \mathbb{R}^d$ be a finite set of input-output pairs dataset with $N \leq d$. Suppose for each $i = 1, \ldots, N$, there exists $L_i \in \{0, \ldots, d-1\}$ such that*

$$W^{(L_i)}X_i \notin \text{span}\{W^{(L_1)}X_1, \ldots, W^{(L_{i-1})}X_{i-1}\}. \tag{4}$$

*Then, there exists a linear operator $J$ on $\mathbb{R}^d$ such that $Y_i = JW^{(L_i)}X_i$ for all $i = 1, \ldots, N$.*

**Remark 3.8.** *On $J$ as an Isomorphism: Define the invertibility metric $\delta := \max_{1 \leq i \leq N} \|Y_i - J_{i-1}W^{(L_i)}X_i\|$. By iterative construction, $J = I_d + \sum_{i=1}^N (Y_i - J_{i-1}W^{(L_i)}X_i) \otimes f_i^*$, so $\|J - I_d\| \leq N\delta \cdot \max_i \|f_i^*\|$. Thus, $J$ is an isomorphism if $\|J - I_d\| < 1$ (see Lemma 2.1, p 192 (6)), achieved when $\delta < 1/(N \cdot \max_i \|f_i^*\|)$.*

*The restriction $L_i \in \{0, \ldots, d-1\}$ in Theorem 3.7 follows from Lemma 3.6, as $W^{(L)}X$ cycles with period $d$ up to scaling. Beyond this, higher $L$ only rescales earlier terms, adding no new linear independence information.*

**Optimization Framework.** The constructive proof of Theorem 3.7 motivates a three-phase learning framework grounded in orbit structure; here we detail Phases 1-3. We select a subset $D$ of the training set and partition it into $K$ subsets $D_k$ for $k \in \{0, \ldots, K-1\}$ so that the recursive linear

independence condition (4) holds within each $D_k$. In practice this condition is easy to satisfy; however, the cap we impose on the size of each $D_k$ can influence generalization, and choosing this cap optimally is left for future work (discussed later).

**Phase 1.** Learn a global shift operator $W$ such that the orbit points $W^{(L_i)}X_i$ satisfy condition (4) across the selected subsets $D_k$.

**Phase 2.** Apply Theorem 3.7 independently to each $D_k$: fit a local linear map $J_k$ on $D_k$ to achieve exact interpolation (equivalently, apply the theorem $K$ times, once per subset).

**Phase 3.** Mimicking dynamics in infinite dimensions, we define Phase 3 of WBSNN as generalization over the space $\sum_{k=0}^{K-1} J_k \cdot \text{span}\left\{W^{(m)}X : m \geq 0\right\}$. Note that this formulation is the reason why we considered unwrapping $W^{(L)}$ modulo $d$; otherwise, standard backward shifts would vanish beyond depth $d$, preventing generalization along the infinite orbit. By Lemma 3.6, in finite-dimensional spaces our generalization expression collapses—up to scalar factors—via *MLP-learned coefficients* $\alpha_{k,m}(X)$ to the prediction rule:

$$\hat{Y}_{\text{new}} = \sum_{k=0}^{K-1} J_k \sum_{m=0}^{d-1} \alpha_{k,m}(X_{\text{new}}) \cdot W^{(m)} X_{\text{new}}, \tag{5}$$

which reuses the operators from Phases 1–2 without compromising the interpolation guarantees on each $D_k$.

Note that Phase 3 is defined on infinite orbits and therefore Phases 1–3 apply to infinite-dimensional datasets, as long as we can prove an exact interpolation theorem for infinite-dimensional Banach spaces. As shown in Theorem A.1, Appendix A.4, the requisite theorem holds, providing the theoretical foundations of WBSNNs in infinite dimensions. In what follows, we restrict our attention to the finite-dimensional case; the infinite-dimensional extension is treated in Appendix A.

We are now ready to introduce WBSNNs in the finite-dimensional setting.

**Definition 3.9** (WBSNN in finite-dimensional datasets). *A* Weighted Backward Shift Neural Network (WBSNN) *on a $d$-dimensional vector space (with $d$ finite) is a three-phase learning model defined by: (i) a fixed shift operator $W \in \mathbb{R}^{d \times d}$,[1] (ii) local linear maps $\{J_k\}$ learned over subsets $D_k$, and (iii) a data-dependent weight function $\alpha_{k,m}(X)$, parametrized via an MLP, for combining orbit elements. The final prediction for input $X \in \mathbb{R}^d$ is given by* (5).

**Visualizing WBSNN in Action: Interpretability on Swiss Roll.** To illustrate WBSNN's interpretability and operational mechanics, we examine its application to the Swiss Roll dataset—a classic nonlinear manifold benchmark—with $d = 3$, 120 samples, and added Gaussian noise of 0.5. In this run, Phase 1 and Phase 2 use approximately 6% of the training set ($\sim$5 points) for learning the shift operator $W$ and subsets $D_k$ for interpolation, while Phase 3 leverages the full training data for generalization. Critical numerical details appear in Appendix B, Table 1.

Phase 1 begins by selecting a subset of training points and optimizing $W = [0.8, 0.8, 0.8]$, partitioning them into 2 subsets $D_0$ and $D_1$ (total 5 points) with $\Delta = 3.1146$. In Phase 2, local operators $J_0$ and $J_1$ are constructed to achieve exact interpolation (norm differences $\approx 0$ within numerical precision, e.g., $7.77 \times 10^{-16}$ for Point 1 under $J_0$), confirming Theorem 2.7's guarantees. This exact fit on subsets ensures the model captures local geometry without overfitting the full dataset. The final performance, shown in Table 1, positions WBSNN as data-efficient and robust: achieving a test accuracy of 0.9583 and train accuracy of 0.9868, competitive with baselines despite using far fewer samples in its core phases. This extreme data efficiency (6% vs. 100% for baselines) underscores WBSNN's ability to capture manifold structure with minimal data. Unlike baselines, which may memorize the small dataset, WBSNN leverages orbit dynamics for robust generalization. Phase 3's MLP outputs $\alpha_{k,m}$ weights, which assign explicit contributions to each projection and dimension, enabling direct inspection of decision factors unlike black-box baselines, enabling traceable predictions via Equation (5), i.e. $\hat{Y} = \sum_{k=0}^{1} J_k \sum_{m=0}^{2} \alpha_{k,m}(X) \cdot W^{(m)} X$.

To visualize WBSNN's interpretability, we examine predictions for two points: (1) a held-out training point not used in Phase 1 (seen by the model during Phase 3 training but excluded from subset

---

[1]In our notation, $W$ denotes the base shift operator of type $\mathbb{R}^d \to \mathbb{R}^d$, while $W^{(L)}$ is a derived object of type $\mathbb{R}^{d+L} \to \mathbb{R}^d$ constructed to simulate $L$ applications of $W$ over extended input.

construction in Phases 1–2), and (2) an unseen test point (never encountered during any phase). This selection highlights WBSNN's generalization: the held-out training point tests interpolation beyond Phase 1 subsets (demonstrating robustness within the training distribution), while the test point evaluates true out-of-sample performance (showing adaptability to unseen manifold regions). Both points yield correct predictions, underscoring the framework's ability to leverage orbit dynamics for reliable, traceable decisions.

For the held-out training point, the prediction sum from Phase 3 gives (2.328883) yielding a sigmoid probability of 0.911241, confidently predicting label 1.0 (matching true). The high probability (0.911241) reflects strong confidence in Class 1, driven by positive terms $\alpha_{k,m}$ from dimensions 1 and 2, aligning with the spiral's geometry. For the unseen test point, the sum $(-2.475847)$ yields a sigmoid probability of 0.077569, predicting label 0.0 (matching true). The low probability (0.077569) indicates robust separation from Class 1, with negative $\alpha_{k,m}$ suppressing projections across all dimensions, illustrating WBSNN's ability to generalize to unseen points by adapting weights to the input's position on the manifold.

These examples, detailed in Appendix B, Table 1, highlight WBSNN's transparency: each prediction decomposes into per-dimension contributions. The selection of these points underscores WBSNN's strengths: the held-out training point shows how the model interpolates within the training distribution without relying on Phase 1 subsets, while the test point validates true extrapolation. This dual view emphasizes interpretability—each prediction traces back to explicit orbit combinations—positioning WBSNN as a transparent alternative to black-box models.

**On $\alpha_{k,m}$ modalities.** Concerning the MLP-learned coefficient $\alpha_{k,m}$ of our prediction rule (5), we consider four WBSNN modalities along two axes—head expressivity and backbone sharing. **(1)** $\alpha_{k,m}$: a full per-$k$, per-state head that gates all dynamical states $m$ within each subset $k$ (maximal flexibility). **(2)** $\alpha_{k,L}$: a middle-ground head that gates only the states established by Phase 1 and 2, meaning for each $0 \leq k \leq K-1$, the coefficient $\alpha_{k,L} = 0$ for all $L \notin \mathcal{L}_k$, where $\mathcal{L}_k$ is the set of $L_i$ satisfying condition (4) for $X_i \in D_k$, reducing variance while retaining within-$k$ selectivity. **(3)** $\alpha_k$ (shared backbone): a per-$k$ head trained on a trunk *primed by* $\alpha_{k,m}$; operationally it aggregates over $m$ using features learned for the richer head, offering strong generalization with lower overfitting. **(4)** $\alpha_k$ (separate backbones): the same per-$k$ head trained on its own trunk, decoupled from $\alpha_{k,m}$. So, $\alpha_{k,L}$ is an intermediate head that sits between $\alpha_{k,m}$ and $\alpha_k$. Our main protocol uses variants (1) and (3), i.e. full $\alpha_{k,m}$ and $\alpha_k$ with a shared, $\alpha_{k,m}$-primed backbone, balancing accuracy, robustness, and efficiency. Appendix E reports the behavior of $\alpha_{k,m}$ and both $\alpha_k$ variants across datasets. We reserve the variant $\alpha_{k,L}$ exclusively for the infinite-dimensional formulation of WBSNNs (Appendix A.4), where it plays a central role.

**Interpolation modalities.** While Theorem 3.7 provides a constructive proof of exact interpolation via sequential updates of the operator $J$, our implementation adheres to the same design principles while allowing a relaxed (non-exact) interpolation adapting to realistic settings for stability, efficiency, and robustness. This last variant intentionally relaxes the exact fit to illustrate the model's behavior under noisy conditions, offering a broader view of orbit-based generalization and adaptive weighting phenomena. Details are deferred to Appendix D. This dual approach allows us to adopt exact interpolation in noiseless or low-noise regimes and to switch to non-exact formulations when data is noisy or ill-conditioned.

**Practical considerations.** Phase 1 evaluates $W^{(L)}X$ across depths $L \leq d$ and tests span membership via a least–squares residual with tolerance $\tau$ (a numerically cheap proxy for exact independence; see Appendix D); the work grows roughly quadratically with the subset size. To keep costs tractable without losing diversity, we optimize $W$ on mini-subsets and form $D_k$ from a subsampled fraction of the training pool (diversity-aware anchor policies—e.g., coverage/k-center ideas—instead of uniform subsampling are left for future work). In Phase 2 we compute $J_k$ using either the constructive rank-one update scheme given by the proof of Theorem 3.7 (yielding exact interpolation), the closed-form pseudoinverse (still giving exact interpolation when full-rank), or a regularized pseudoinverse for noisy/ill-conditioned cases yielding relaxed interpolation, details in (Appendix D). Our main results default to the regularized form for stability in noisy/compressed regimes, with exact variants used when independence is clean. Crucially, we handle vector and scalar targets uniformly in the same $d$-dimensional space: classification uses one-hot label vectors, and regression embeds the scalar along a fixed direction and reads it back through a fixed linear functional. Phase 3 then reuses the same orbit-weighted combination, preserving interpretability.

**Empirical Observation (implicit regularization).** Across nine datasets and varied configurations of input dimension and sample size, the learned shift $W$ almost always lies in the vanishing regime, $\prod_{i=0}^{d-1} w_i < 1$. This contraction behaves as an implicit regularizer—improving generalization and noise robustness—while rare cases favor neutral or mildly expanding dynamics to preserve signal, reflecting a regularization–expressivity trade-off. This motivates the next lemma.

**Noise Suppression via Orbit Decay.** WBSNN generalizes over the infinite orbit $\{W^{(m)}X\}_{m\in\mathbb{N}}$, which, by Lemma 3.6, collapses—up to scalars—to its first $d$ iterates. In the vanishing regime—where $\rho := \prod_{i=0}^{d-1} |w_i| < 1$—each $W^{(m)}\varepsilon$ decays as $\rho^{\lfloor m/d \rfloor}$, producing spectral regularization. Though Phase 3 only uses shallow iterates, the full decay structure biases the model toward stability. This yields noise suppression without explicit penalties (like dropout layers), driven solely by the orbit geometry of the shift operator itself. The result is stable learning behavior in noisy or low-sample settings, as formally shown below. (Proof deferred to Appendix A.3).

**Lemma 3.10** (Noise Suppression via Orbit Decay). *Let $\varepsilon \in \mathbb{R}^d$ be an additive noise vector and let $W^{(m)}$ with weights $w_0, \ldots, w_{d-1} \in \mathbb{R}$ such that $\rho := \prod_{i=0}^{d-1} |w_i| < 1$. Then for all $m \geq 1$,*

$$\|W^{(m)}\varepsilon\| \leq \rho^{\lfloor m/d \rfloor} \cdot C,$$

*for some constant $C > 0$ depending only on $\varepsilon$ and $d$. In particular, $\|W^{(m)}\varepsilon\| \to 0$ as $m \to \infty$.*

**Remark 3.11.** *This shows that although Phase 3 in WBSNN only uses orbit powers $m \leq d - 1$, these terms implicitly encode decay across the full infinite orbit $\{W^{(m)}\varepsilon\}_{m\geq 0}$ via scalar rescaling. The vanishing regime $\prod_{i=0}^{d-1} w_i < 1$ induces exponential suppression of long-term perturbations, acting as an intrinsic regularizer in the orbit geometry.*

For a concrete instantiation of our main protocol: relaxed (non-exact interpolation), see Algorithm 1 in Appendix C.

## 4 EXPERIMENTS

**Protocol and compute.** We report results across nine datasets under one protocol: train-only standardization, small PCA bottlenecks, and tiny Phase-1/2 "discovery" budgets to build the orbit dictionary. Unless noted otherwise, $\alpha_k$ denotes the shared-trunk head (the main protocol), and $\alpha_{k,m}$ is the richer head gating per subset and per orbit state. All experiments run in a lightweight CPU-only Jupyter environment—no GPUs—so the numbers reflect true data/geometry leverage rather than horsepower.

*Baselines.* We report tuned baselines spanning: linear (LogReg, LinearSVM), kernel (RBF-SVM, KRR, Nyström, RFF), tree ensembles (RF, XGBoost, ExtraTrees), neural (MLP, CNN/LeNet/ResNet-18), sequence/dynamical (LSTM, Linformer, NAIS-Net), operator-learning (DeepONet, FNO), spectral/graph and nonparametric (Diffusion Maps+LogReg, Laplacian-LogReg, Scattering2D, Hankel-SVD+LDA, $k$-NN/Label Propagation), and domain-specific (CORAL, time-delay ridge, EDMD).

**CIFAR-100 (compressed, capped).** On a small-but-stubborn CIFAR-100 variant (30 classes, frozen ResNet-18 features, PCA to $d \in \{10, 20\}$ and 1–3% discovery), compressing to $d=10$ with a 3k budget keeps only ~31% variance and creates a low-margin regime: linear and kernel heads hover around 50%, while $\alpha_{k,m}$ sits between 45-47% and outperforms $\alpha_k$ at this extreme compression. Widening to $d=20$ (~43% variance) and a 5k budget pushes $\alpha_{k,m}$ into the high-50s, narrowing the gap to the best shallow baselines despite discovery still capped at 3%—evidence that widening the bottleneck and modestly enlarging the pool helps the orbit dictionary organize what signal survives projection, see Tables 2 and 3 in the Appendix for details.

**Gas Sensor Drift (chronological, distribution shift).** The chronology (10 batches over 36 months; strict out-of-time split; PCA $d \in \{5, 10, 15\}$; discovery 1–7%) stresses noise and drift simultaneously. Here, WBSNN's stability under drift and noise shone through. At $d=5$ with $n \approx 1.6$k and 5–7% discovery, all WBSNN heads land ~59–61% with $\alpha_k$ (separate) 7% discovery outperforming the strongest baseline overall under drift. Opening to $d=10$ and $n=3$k (EVR=96%) reveals the scaling law cleanly: $\alpha_{k,m}$ peaks near 68% at 3% discovery and remains best overall; both $\alpha_k$ variants are more budget-hungry but converge within a point or two by 5%. A 20-seed sweep at $d=10$ (10% dis-

covery) shows $\alpha_{k,m}$ $0.647 \pm 0.0367$, $\alpha_k$(shared) $0.645 \pm 0.051$, $\alpha_k$(separate) $0.65 \pm 0.037$—modest spread under drift and reproducible gains from sparse discovery. More details in Tables 4–7.

**IMDb (compressed text, mean-pooled).** With mean-pooled GloVe-100 embeddings, PCA $d \in \{10, 20, 50\}$, and 1–15% discovery, WBSNN scales predictably despite erasing word order. With 2k samples at $d=10$ (EVR≈0.52), $\alpha_{k,m}$ holds near 0.75 (best overall); with 4k at $d=20$ (EVR≈0.67) it rises to ∼0.77 (best overall with Random Forest); with 25k at $d=50$ (EVR≈0.87) and just 1% discovery it reaches ∼0.793, keeping pace with RBF-SVM/Nyström and outrunning logistic/GBDT. Exact-interpolation runs confirm the bias–variance trade: at low data they match or beat non-exact (e.g., ∼0.73–0.77 at $d \in \{10, 20\}$ with 2–4k) with zero anchor residuals, while non-exact becomes preferable as samples and horizon grow. More details in Tables 8–12.

**MNIST (PCA bottlenecks).** Compressed regimes expose structure vs. capacity. At $d=5$ with 2k samples (∼33% variance), accuracy is limited by information loss; WBSNN remains competitive with SVM and consistently beats logistic/MLP/forest. At $d=15$ with 5k samples (∼58% variance), both heads climb; $\alpha_{k,m}$ benefits more from the extra signal. At $d=30$ with 10k samples (∼73% variance), WBSNN exceeds 95% test accuracy—rivaling strong PCA baselines (in both $d = 15, d = 30$) without image-specific machinery—while accuracy rises monotonically with Phase-budget. Details in Tables 13–15.

**Noisy synthetic gauntlet (heavy tails, spurious cue).** A 15k×50D task with few informative co-ordinates, heavy-tailed heteroskedastic noise, boundary-peaked label flips, and a 0.6-correlated spurious feature separates information-seekers from shortcut-takers. Under tight discovery budgets, WBSNN recovers signal reliably: at small $d$ it achieves exact interpolation (Phase-2 residuals ≈ 0) yet generalizes; as $d$ (5→20) or $n$ (1.5k→12k) grow, accuracy rises to ≈ 0.75–0.78, matching or slightly edging strong tabular/kernel baselines (LogReg, RBF-SVM, RFF+LogReg, ExtraTrees) and a tuned 1-layer MLP. Random forests overfit, while WBSNN remains stable and spurious-resistant, with predictable gains from modest increases in $d$ or $n$. More details in Tables 16–20.

**FI-2010 (limit order book).** Chronological splits; PCA $d \in \{10, 20, 40\}$; discovery 1–10%; $n = 1k–30k$. In the harsh corner $d=10, n=1k$, $\alpha_{k,m}$ ∼0.458 and $\alpha_k$ 0.404→0.476 (as budget rises) sit with LR/RBF-SVM/Nyström/MLP/RF around 0.40–0.47; Phase-2 already aligns most anchors exactly. At $d=20, n=2k$, both heads firm near 0.49 (best overall), ahead of Linformer/NAIS-Net/DeepONet/LR/MLP. Push to $d=20, n=10k$: WBSNN jumps to ∼0.62; at $d=20, n=30k$ with just 1–3% discovery, both heads reach ∼0.65 while baselines lag in the 0.44–0.59 range. A 20-seed small-data sweep ($d=10, n=2k$, 10%) yields $0.504 \pm 0.046$ for $\alpha_{k,m}$ and $0.500 \pm 0.051$ for $\alpha_k$: typical low-$n$ dispersion that tightens as information grows. Details in Tables 21–26.

**PRSA-2017 (air-quality regression).** Four meteorological features; strict chronological split where the last 20% (∼82k hours) is the fixed test window. WBSNN's $\alpha_{k,m}$ scales from $R^2 \approx 0.27$ at $n=1k$ to $0.29-0.30$ at 3k and $0.32-0.33$ at 10k with discovery ranging 1–15%, while MAE stays ∼0.58–0.60. The head hierarchy is consistent: $\alpha_{k,m}$ leads; $\alpha_k$(shared) tracks closely and beats $\alpha_k$(separate) as data grow. Exact vs. non-exact shows the intended trade-off: exact wins at $n=1k$ (e.g., $R^2 \approx 0.283 - 0.284$ vs. ∼ 0.27) and stays similar at 3k, but non-exact pulls ahead at 10k (∼$0.326-0.332$ vs. ∼$0.297-0.320$), suggesting a small slack in Phase-2 improves drift-robustness on a massive future window. Proportional 80/20 ablations (e.g., 3k/600 and 10k/2k) mirror this: exact is stable on small tests; non-exact scales better on broader-horizon evaluations. Tables 27–31.

**ISOLET (speech; 26-way).** A clean compression/budget sweep. In the compressed corner ($d=5, n=2000$), $\alpha_{k,m}$ climbs from 62.0%→64.8% as discovery rises 5%→15%, trailing only RBF-SVM and beating logistic/MLP/RF/k-NN/Nyström/Label-Prop; $\alpha_k$(shared) sits just behind and clearly outperforms $\alpha_k$(separate). A 20-seed study at $d=5$, 10% shows $\alpha_{k,m}$ $0.62 \pm 0.011$, $\alpha_k$(shared) $0.58 \pm 0.019$, $\alpha_k$(separate) $0.55 \pm 0.022$—sharing the trunk lowers variance. At $d=10, n=4000$, $\alpha_{k,m}$ reaches 77.1% (5%), essentially tied with RBF-SVM (77.6%) and ahead of other baselines. On the full train, tight $d=5$ caps at ∼65%, while $d=20$ yields ∼90% and $d=35$ 91.9–92.4% with just 3–5% discovery—competitive with logistic/MLP and within a few points of RBF-SVM, all while forests/k-NN overfit. More details in Tables 32–37.

**Swiss Roll + RFF (classification & regression).** Three tough classification tracks and a heteroskedastic regression confirm that tiny discovery budgets suffice when geometry is organized. In noisy 5-class, small runs (RFF20→PCA10, $M_{\text{train}}=800$, 15%) already hit ∼95% ($\alpha_{k,m}$) in line with RBF-SVM; scaling to RFF30→PCA15 with $M_{\text{train}}=30k$ and just 7% anchors keeps both heads

$\sim 96\%$. In low-sample + label-noise ($\sim 20\%$ flips), the simpler $\alpha_k$ (shared) can edge $\alpha_{k,m}$ at higher $d$ and $M$ (e.g., $\sim 75\%$ at RFF20$\rightarrow$PCA15, 1k, 10%), consistent with a helpful bias under uniform flip noise. On multi-roll (4 spirals), WBSNN rides the topology: $\sim 94$–$95\%$ at RFF20$\rightarrow$PCA10, 2k/10% and $\sim 97\%$ at RFF30$\rightarrow$PCA15, 20k/7%, neck-and-neck with the best kernels/MLPs. For regression, $\alpha_k$ is the safer bias-controlled choice at scale (e.g., $R^2 \approx 0.83$ at 20k vs. $\sim 0.78$ for $\alpha_{k,m}$), showing that wider heads add variance when signal is essentially 1-D. Further details in Tables 38–46.

**Swiss Roll + polynomial embedding.** With explicit quadratics + interactions + six spurious channels, WBSNN behaves like a well-regularized model. At mild noise (0.2), it tracks 98.1% and logistic wins by a hair 98.2%—exactly what you expect in a quasi-linearized space. At noise 0.5, WBSNN holds mid-95s, with smaller $d$ acting as a built-in denoiser (e.g., 95.3% at $d=10$, 5k vs. 93.6% at $d=15$, 5k); at noise 0.8, small-$d$ plus small budgets remains the safe recipe, and WBSNN stays competitive while kernel/linear baselines occasionally nip it by 0.5–1 point—again consistent with bias–variance under noisy algebraic features. More details in Tables 47–50.

**Swiss Roll (raw 3D).** No feature maps, just standardization—WBSNN treats geometry as home turf. With three noisy classes ($\sigma=0.5$) and only $\sim 10\%$ discovery, it is already 99.4% at 10k. Logistic stalls at 61%, underscoring that this is about geometry, not head capacity. Under scarcity + label noise (10-way, 10% flips, 1k points), WBSNN holds 91.5% and matches or edges kernels. On the 3-roll topology at 20k, it again rides with the leaders ($\sim 98.7\%$) with only 7% budget. For regression on the unwrapped angle, the shared $\alpha_k$ head ages better as scale increases (e.g., $R^2 \approx 0.986$ at 2k vs. $\sim 0.969$ for $\alpha_{k,m}$, and $R^2 \approx 0.961$ at 10k vs. $\sim 0.938$), highlighting that a narrower head can be the right bias when the target is essentially 1-D. Further details in Tables 51–56.

**Conclusion.** Across a deliberately eclectic suite of datasets—under stressors including heavy/heteroskedastic noise, distribution drift, extreme PCA compression, label flips, spurious cues, and multi-roll topology—the pattern is consistent. With tiny discovery budgets (often 1–10%) and a small head, WBSNN scales with information: once the orbit dictionary has enough anchors, it matches or beats strong linear, kernel, tree, neural, and operator-learning baselines (e.g., DeepONet, FNO, NAIS-Net, EDMD). Grounded in operator theory and explicitly orbit-driven, WBSNN shines among operator-learning counterparts with consistent improvements. The richer $\alpha_{k,m}$ head is the geometry workhorse on nonlinear class boundaries, while the shared $\alpha_k$ head is the safer bias for noisy 1-D regressions and flip-noise regimes. Error-bar studies (20 seeds) show moderate spreads that contract as $n$ or PCA retained variance grows; exact fits are preferred at small data or small $d$, while relaxed fits win as samples or horizons increase. In short, WBSNN learns the data's geometry rather than memorizing examples—delivering competitive accuracy under CPU-only constraints.

## 5 FUTURE WORK.

**Extension to Infinite Dimensions.** We established WBSNN's foundations in infinite-dimensional sequence Banach spaces (Appendix A.4). Future work will systematically explore this regime on inherently infinite-dimensional datasets, leveraging WBSNN's orbit dictionary to model complex, high-dimensional data structures while maintaining computational efficiency and interpretability.

**Toward a Geometric Theory of Datasets.** While this work introduces WBSNN and demonstrates its empirical and theoretical potential, we are still far from understanding how to optimally configure it for a given dataset $\mathcal{D}$. Interestingly, WBSNN consistently selects a vanishing regime ($\prod_{i=1}^{d} w_i < 1$) during Phase 1 optimization—promoting orbit stability and interpolation quality. This behavior, though not enforced, suggests the existence of dataset-specific configurations $(K_\mathcal{D}, W_\mathcal{D}, J_\mathcal{D})$ such that WBSNN acts as a functor extracting the intrinsic geometry or topology of $\mathcal{D}$. Such a perspective may ultimately allow us to reason about learning not only through data and models, but through the geometry they induce. If formalized, this could ground a geometric or category-theoretic framework for learning, where orbit representations encode generalization structure.

## ETHICS STATEMENT

We adhere to the ICLR Code of Ethics. Our work uses only public, non-identifiable datasets and does not involve human subjects or sensitive attributes. We release code and full result tables to support transparency and responsible use.

## Reproducibility Statement

An anonymous repository with all code, configs, and per-dataset notebooks/markdowns is available at `https://github.com/wbsnn2025/wbsnn_for_reviewer_iclr2026`. We pin exact package versions and provide commands to regenerate all tables and figures from a clean clone. We use train-only standardization and PCA across all datasets; splits and seeds are specified in the repo README.

All assumptions and complete proofs (including the infinite-dimensional foundations for WBSNNs) are in Appendix A. Algorithmic pseudocode appears in Appendix C (Algorithm 1); Appendix D explains the theorem–algorithm correspondence, the Phase-1 independence proxy, and the three Phase-2 interpolation modalities. A Swiss-Roll interpretability example is summarized in Table 1. Comprehensive per-dataset results and ablations are in Appendix E.

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

## A  Proofs

### A.1  Proof of Lemma 3.6

***Proof of Lemma 3.6.*** Let $W$ with $d$ predictors, where $W^{(L)} \in \mathbb{R}^{d \times (d+L)}$ applies an extended modulo $d$ shift as defined in (3), with the action on $X \in \mathbb{R}^d$ (implicitly extended to $X_{\text{ext},L}$) given by

(2). For $1 \leq i \leq d$, (2) gives $(W^{(L)}X)_i = \left(\prod_{k=0}^{L-1} w_{(i+k) \bmod d}\right) X_{\text{ext},L,(i+L-1) \bmod d}$, where $X_{\text{ext},L,j} = X_{j \bmod d}$ for all $0 \leq j \leq d + L - 1$. Similarly,

$$(W^{(L \bmod d)}X)_i = \left(\prod_{k=0}^{(L \bmod d)-1} w_{(i+k) \bmod d}\right) X_{\text{ext},L \bmod d,(i+(L \bmod d)-1) \bmod d}.$$

Since $(i+L-1) \bmod d = (i+(L \bmod d)-1) \bmod d$, the indices match, so $X_{\text{ext},L,(i+L-1) \bmod d} = X_{\text{ext},L \bmod d,(i+(L \bmod d)-1) \bmod d}$. For $L \geq 1$, there exists $m \geq 0$ such that $L = md + (L \bmod d)$, where $0 \leq L \bmod d < d$. The product in $W^{(L)}$ splits as:

$$\prod_{k=0}^{L-1} w_{(i+k) \bmod d} = \left(\prod_{n=0}^{m-1} \prod_{k=0}^{d-1} w_{(i+nd+k) \bmod d}\right) \prod_{k=0}^{(L \bmod d)-1} w_{(i+md+k) \bmod d}.$$

Indeed, the first $md$ terms of $\prod_{k=0}^{L-1} w_{(i+k) \bmod d}$ are $m$ full cycles (of length $d$) with indices $i + nd + k$ for $n$ from $0$ to $m-1$ and $k$ from $0$ to $d-1$. So, for each $n$, we get

$$w_{(i+nd) \bmod d} \cdot w_{(i+nd+1) \bmod d} \cdot \cdots \cdot w_{(i+nd+(d-1)) \bmod d}$$

which covers one full cycle of indices. The remaining $L \bmod d$ terms (from $k = md$ to $k = L - 1$) are

$$w_{(i+md) \bmod d} \cdot w_{(i+md+1) \bmod d} \cdot \cdots \cdot w_{(i+md+(L \bmod d)-1) \bmod d}$$

which is exactly $\prod_{k=0}^{(L \bmod d)-1} w_{(i+md+k) \bmod d}$.

For each $n$, the inner product $\prod_{k=0}^{d-1} w_{(i+nd+k) \bmod d} = \prod_{k=0}^{d-1} w_{(i+k) \bmod d} = \prod_{k=0}^{d-1} w_k$, since the indices $(i+nd+k) \bmod d$ cycle through $i, i+1, \ldots, i+d-1$. Thus, the first term is $\left(\prod_{k=0}^{d-1} w_k\right)^m$. Hence, $(W^{(L)}X)_i = \left(\prod_{k=0}^{d-1} w_k\right)^m (W^{(L \bmod d)}X)_i$, and the scalar $\lambda = \left(\prod_{k=0}^{d-1} w_k\right)^m$ satisfies the lemma. $\qquad\square$

## A.2 Proof of Theorem 3.7 on Universal Representation via Transformed Weighted Shifts

***Proof of Theorem 3.7.*** Let $W$ on $\mathbb{R}^d$ with weights $w_n \neq 0$. Let also $(X_i, Y_i)_{i=1}^N \subset \mathbb{R}^d \times \mathbb{R}^d$ be a finite set of input-output pairs satisfying (4).

Initialize: Set $J_0 = I_d$, the identity operator on $\mathbb{R}^d$.

Iterative construction: For $i = 1$. Pick any $L_1 \in \{0, \ldots, d-1\}$ and find a linear functional $f_1^* \in (\mathbb{R}^d)^*$ with $f_1^*(W^{(L_1)}X_1) = 1$. Define $J_1 v := J_0 v + (Y_1 - J_0 W^{(L_1)}X_1) \otimes f_1^*(v) = v + (Y_1 - W^{(L_1)}X_1) \otimes f_1^*(v)$. Thus, $J_1 W^{(L_1)}X_1 = W^{(L_1)}X_1 + (Y_1 - W^{(L_1)}X_1) \cdot f_1^*(W^{(L_1)}X_1) = W^{(L_1)}X_1 + (Y_1 - W^{(L_1)}X_1) \cdot 1 = Y_1$.

For $i \geq 2$: Assume

$$J_{i-1}W^{(L_j)}X_j = Y_j \quad \text{for } j \leq i - 1, \tag{6}$$

and pick $L_i \in \{0, \ldots, d-1\}$ satisfying condition (4). By Hahn-Banach Theorem (see Corollary 3.15 on p.112 (6)), there exists a linear functional $f_i^* \in (\mathbb{R}^d)^*$ such that $f_i^*(W^{(L_j)}X_j) = 0$ for $j \leq i - 1$, and $f_i^*(W^{(L_i)}X_i) = 1$. Define $J_i$ as follows:

$$J_i v := J_{i-1}v + (Y_i - J_{i-1}W^{(L_i)}X_i) \otimes f_i^*(v). \tag{7}$$

Thus,

$$J_i W^{(L_i)}X_i = J_{i-1}W^{(L_i)}X_i + (Y_i - J_{i-1}W^{(L_i)}X_i) \cdot 1 = Y_i, \quad \text{and}$$

$$J_i W^{(L_j)}X_j = J_{i-1}W^{(L_j)}X_j = Y_j \text{ for } j \leq i - 1 \text{ (by } f_i^*(W^{(L_j)}X_j) = 0 \text{ and (6))}.$$

By induction, for $1 \leq i \leq N$: $J_i W^{(L_j)}X_j = Y_j$ for $j \leq i$. Finally, by setting $J := J_N$, we have that $Y_i = J W^{(L_i)}X_i$ for all $1 \leq i \leq N$.

$\qquad\square$

## A.3 PROOF OF LEMMA 3.10 ON NOISE SUPPRESSION VIA ORBIT DECAY

***Proof of Lemma 3.10.*** By Lemma 3.6, for every $m \geq 1$, there exists a scalar $\lambda_m > 0$ such that

$$W^{(m)}\varepsilon = \lambda_m \cdot W^{(m \bmod d)}\varepsilon.$$

Let $m = qd + r$, with $0 \leq r < d$ and $q \in \mathbb{N}$. Then, $\lambda_m = \left(\prod_{i=0}^{d-1} w_i\right)^q = \rho^q$. Hence,

$$\|W^{(m)}\varepsilon\| = \lambda_m \cdot \|W^{(r)}\varepsilon\| \leq \rho^q \cdot \max_{0 \leq r < d} \|W^{(r)}\varepsilon\| = \rho^{\lfloor m/d \rfloor} \cdot C,$$

where $C := \max_{0 \leq r < d} \|W^{(r)}\varepsilon\|$ depends only on $\varepsilon$ and $d$. Since $\rho < 1$, it follows that $\|W^{(m)}\varepsilon\| \to 0$ exponentially as $m \to \infty$. $\qquad\square$

## A.4 WBSNNs IN INFINITE-DIMENSIONAL DATASETS.

To extend WBSNNs to infinite dimensions, we require an exact interpolation theorem in a sequence Banach space. We work with the standard weighted backward shift $B_w$ as defined in (1), acting on a sequence Banach space $Z$ (e.g., $\ell^p$, $c_0$), assuming $\sup_n |w_n| < \infty$ so that $B_w$ is bounded. In this setting we no longer wrap coordinates modulo $d$; orbits are formed by the genuine iterates $B_w^L$.

**Theorem A.1** (Universal Representation via Transformed Weighted Backward Shifts in infinite-dimensional sequence Banach spaces.). *Let $B_w$ be a weighted backward shift as defined in (1) acting on an infinite-dimensional sequence Banach space $Z$. Let $\mathcal{D}$ be a dataset living on $Z$, and for $N \geq 1$ consider $(X_i, Y_i)_{i=1}^N \subset \mathcal{D}$ a finite sequence of input-output pairs. Suppose for each $i = 1, \ldots, N$, there exists $L_i \geq 1$ such that*

$$B_w^{L_i} X_i \notin \mathrm{span}\{B_w^{L_1} X_1, \ldots, B_w^{L_{i-1}} X_{i-1}\}. \tag{8}$$

*Then, there exists a linear and bounded operator $J$ on $Z$ such that $Y_i = JB_w^{L_i} X_i$ for all $i = 1, \ldots, N$.*

Because the Hahn–Banach Theorem holds in Banach spaces and the proof of Theorem 3.7 is dimension-free—relying only on linearity, Hahn–Banach separation, and boundedness, with no finite-dimensional arguments—it carries over verbatim from $\mathbb{R}^d$ to the infinite-dimensional setting. Thus Theorem A.1 follows by replacing the wrapped operator $W^{(L)}$ (modulo $d$) with the genuine iterate $B_w^L$ (assuming $B_w$ is bounded).

**How to apply Theorem A.1.** Given a countable sequence of input-output pairs dataset $\mathcal{D} = (X_i, Y_i)_{i=1}^\infty$ on a infinite-dimensional sequence Banach space, denote by $\mathcal{T}$ the training set and pick $K$ disjoint subsets $D_k$ with $0 \leq k \leq K - 1$, where

$$D_k = \{(i, L_i) : (X_i, Y_i) \in \mathcal{T}, Y_i = J_k B_w^{L_i} X_i\}$$

and $J_k$ is the linear bounded operator guaranteed by Theorem A.1 applied to $D_k$. Note that the number of elements of $\cup_{k=1}^K D_k$ can be strictly less than the training set size allowing for small budgets of WBSNN's Phase 1 and 2. For an unseen $X$, our prediction rule becomes

$$\hat{Y}_{\text{new}} = \sum_{k=0}^{K-1} J_k \sum_{L \in L_k} \alpha_{k,L}(X_{\text{new}}) \cdot B_w^L X_{\text{new}}, \tag{9}$$

where $L_k = \{L_i : (i, L_i) \in D_k\}$.

Computationally, we restrict the inner sum to a finite window $L \in L_k$ (or alternatively, we could enforce $\ell^1$-summability/decay constraints on $\{\alpha_{k,L}\}_{L \in \mathbb{N}_0}$), ensuring the prediction map is well-posed and efficiently implementable. For computational convenience we adopt the middle-ground head variant $\alpha_{k,L}$ as the default head (see definition in 'On $(\alpha_{k,m})$ modalities' subsection), which aggregates over depth $L$ per subset $k$ without enumerating all residue classes $m$.

By Theorem A.1, the Phase 1–3 pipeline extends to an infinite-dimensional sequence Banach space $Z$, assuming $B_w$ is bounded (i.e., $\sup_n |w_n| < \infty$): Phase 1 learns $B_w$; Phase 2 fits bounded linear maps $J_k$ over disjoint $D_k$; Phase 3 combines orbit features $\{B_w^L X\}$ via $\alpha_{k,L}(X)$ to generalize beyond the interpolation sets. Unlike the finite-dimensional case, there is no modulo-$d$ collapse; unleashing the broad potential of WBSNNs in infinite-dimensional settings.

**Definition A.2** (WBSNN in $\infty$-dimensional datasets). *A Weighted Backward Shift Neural Network (WBSNN) on an infinite-dimensional sequence Banach space $Z$ is a three-phase learning model defined by: (i) a weighted backward shift $B_w$ acting on $Z$; (ii) local bounded linear maps $\{J_k\}$ learned over subsets $D_k$; and (iii) a data-dependent weight function $\alpha_{k,L}(X)$, parametrized via an MLP, for combining orbit elements. The final prediction for input $X \in Z$ is given by (9). For computational well-posedness, we restrict the inner sum to a finite window $L \in L_k$ or enforce $\ell^1$-summability/decay constraints on $\{\alpha_{k,L}(X)\}_{L \geq 0}$.*

Although learning occurs in finite dimension, increasing the number of selected training pairs $N$ in Theorem A.1 and allowing larger $D_k$ subsets expands the set of orbit positions used in Phases 1–2. This yields a finer approximation of the orbit geometry that would arise in infinite dimensions, while remaining computationally controlled via the learned head.

# B   WBSNN BACKBONE VISUALIZED

The following diagram illustrates the computation of $W^{(L)}x^{(0)}$ for a WBSNN with $d = 5$ and $L = 11$.

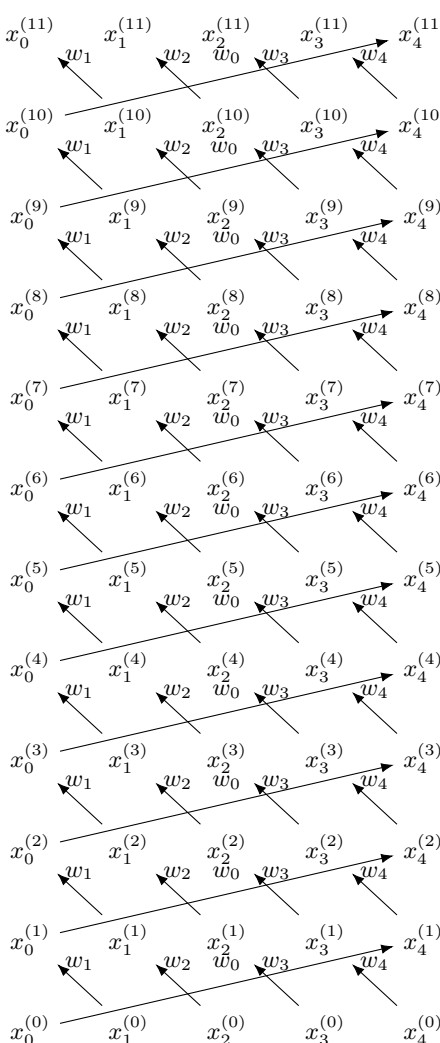

Figure 1: WBSNN with $d = 5$ predictors and depth $L = 11$.

**Dynamics of $W^{(L)}$:** WBSNN's backbone leverages a structured weight matrix $W$ to generate orbit vectors, as illustrated in Figure 1 for $d = 5$ and $L = 11$. For an input $x^{(0)} \in \mathbb{R}^d$, each application of $W$ shifts the components cyclically, where $x_{i-1}^{(l+1)} = w_i x_i^{(l)}$ (indices modulo $d$), with weights $w_0, w_1, ..., w_{d-1}$ applied sequentially. After $L$ iterations, $W^{(L)}x^{(0)}$ emerges as a vector in $\mathbb{R}^d$, capturing a rich trajectory of transformations. This orbit underpins Phase 1's subset construction, enabling the model to exploit temporal and spatial patterns in the input data.

Table 1: Interpretability Details for WBSNN on Swiss Roll ($d = 3$, 120 samples, noise=0.5).

| **Phase 1 Subset Points** | Point $X_1$: $[1.6078, 1.2816, 0.6559]$, Label: 1.0 |
| | Point $X_2$: $[-2.0611, -1.0536, -0.3633]$, Label: 1.0 |
| | Point $X_3$: $[0.4151, 1.5876, 0.5976]$, Label: 0.0 |
| | Point $X_4$: $[-2.0067, -0.6176, -0.0628]$, Label: 1.0 |
| | Point $X_5$: $[0.5160, -1.6329, -0.1805]$, Label: 0.0 |
| **Phase 1 Weights** | $W = [0.8, 0.8, 0.8]$, $\Delta = 3.1146$, $X_1 \in D_0, X_2 \in D_0, X_3 \in D_0, X_4 \in D_1, X_5 \in D_1$ |
| **Phase 2 Interpolation** | Point $X_1$: $L_1 = 1$, Norm of $Y_1 - J_0 W^{(2)} X_1 = 7.77 \times 10^{-16}$ |
| | Point $X_2$ : $L_2 = 2$, Norm of $Y_2 - J_0 W^{(2)} X_2 = 6.66 \times 10^{-16}$ |
| | Point $X_3$ : $L_3 = 2$, Norm of $Y_3 - J_0 W^{(2)} X_3 = 7.39 \times 10^{-16}$ |
| | Point $X_4$ : $L_4 = 2$, Norm of $Y_4 - J_1 W^{(1)} X_4 = 8.88 \times 10^{-16}$ |
| | Point $X_5$ : $L_5 = 2$, Norm of $Y_5 - J_1 W^{(2)} X_5 = 7.24 \times 10^{-16}$ |

| | Model | Train Acc. | Test Acc. | Train Loss | Test Loss |
|---|---|---|---|---|---|
| | WBSNN | 0.9868 | 0.9583 | 0.1011 | 0.1930 |
| **Phase 3 + baselines** | Logistic Reg. | 0.5921 | 0.7917 | 0.6636 | 0.6026 |
| | Random Forest | 1.0000 | 1.0000 | 0.5284 | 0.5685 |
| | SVM (RBF) | 0.9868 | 1.0000 | 0.3147 | 0.3330 |
| | MLP 1HL | 1.0000 | 1.0000 | 0.5134 | 0.5233 |

| **Held-out Train Point** $\alpha_{k,m}(X)$ **Matrix** | $X = [0.3204, -0.7333, 1.8079]$, Label: 1.0 |
| | $$\begin{bmatrix} \alpha_{0,0}(X) & \alpha_{0,1}(X) & \alpha_{0,2}(X) \\ \alpha_{1,0}(X) & \alpha_{1,1}(X) & \alpha_{1,2}(X) \end{bmatrix} = \begin{bmatrix} 1.0000 & -1.0000 & 1.0000 \\ 0.5943 & -0.7987 & 0.2296 \end{bmatrix}$$ |
| **Prediction Breakdown** | Dim 1: $[J_0 W^{(0)} X, J_1 W^{(0)} X] = [0.0250, 0.0763]$, so we have Term $= [\alpha_{0,0}(X), \alpha_{1,0}(X)] \cdot [J_0 W^{(0)} X, J_1 W^{(0)} X]^T = 0.070374$. |
| | Dim 2: $[J_0 W^{(1)} X, J_1 W^{(1)} X] = [-0.6677, -0.9095]$, so we have Term $= [\alpha_{0,1}(X), \alpha_{1,1}(X)] \cdot [J_0 W^{(1)} X, J_1 W^{(1)} X]^T = 1.394070$. |
| | Dim 3: $[J_0 W^{(2)} X, J_1 W^{(2)} X] = [0.8810, -0.0718]$, so we have Term $= [\alpha_{0,2}(X), \alpha_{1,2}(X)] \cdot [J_0 W^{(2)} X, J_1 W^{(2)} X]^T = 0.864439$. |
| | Sum of Terms: 2.328883, sigmoid probability $\sigma(2.328883) = 0.911241$, indicating high confidence in Class 1 due to positive terms, Pred Label: 1.0, True Label: 1.0. |
| **Test Point (Unseen)** $\alpha_{k,m}(X)$ **Matrix** | $X = [0.4585, 0.6298, 0.0940]$, Label: 0.0 |
| | $$\begin{bmatrix} \alpha_{0,0}(X) & \alpha_{0,1}(X) & \alpha_{0,2}(X) \\ \alpha_{1,0}(X) & \alpha_{1,1}(X) & \alpha_{1,2}(X) \end{bmatrix} = \begin{bmatrix} 0.0399 & -1.0000 & 0.1973 \\ 1.0000 & 0.8567 & 1.0000 \end{bmatrix}$$ |
| **Prediction Breakdown** | Dim 1: $[J_0 W^{(0)} X, J_1 W^{(0)} X] = [0.2504, -0.8192]$, so we have Term $= [\alpha_{0,0}(X), \alpha_{1,0}(X)] \cdot [J_0 W^{(0)} X, J_1 W^{(0)} X]^T = -0.809212$. |
| | Dim 2: $[J_0 W^{(1)} X, J_1 W^{(1)} X] = [0.5423, -0.3129]$, so we have Term $= [\alpha_{0,1}(X), \alpha_{1,1}(X)] \cdot [J_0 W^{(1)} X, J_1 W^{(1)} X]^T = -0.810256$. |
| | Dim 3: $[J_0 W^{(2)} X, J_1 W^{(2)} X] = [-0.2839, -0.8004]$, so we have Term $= [\alpha_{0,2}(X), \alpha_{1,2}(X)] \cdot J_0 W^{(2)} X, J_1 W^{(2)} X]^T = -0.856379$. |
| | Sum of Terms: -2.475847, sigmoid probability $\sigma(-2.475847) = 0.077569$, reflecting strong confidence in Class 0 via negative terms, Pred Label: 0.0, True Label: 0.0. |

# C PSEUDOCODE FOR WBSNN

---

**Algorithm 1:** WBSNN Optimization Framework

---

**Input:** Dataset $\{(X_i, Y_i)\}_{i=1}^M$, dimension $d$, new input $X_{\text{new}}$

**Output:** Predicted $\hat{Y}_{\text{new}}$

**1 Phase 1: Subset Construction and Operator Optimization**

**2** Initialize shift weights $W \in \mathbb{R}^d$ (e.g., $W = \mathbf{1}$); set residual threshold $\tau > 0$.

**3** Randomly sample a subset $\{(X_i, Y_i)\}_{i \in \mathcal{S}}$, $\mathcal{S} \subset \{1, \ldots, M\}$ (e.g., 10% of training data).

**4** Define $D_k = \{(i, L_i) \mid i \in \mathcal{S}_k \subset \mathcal{S}, L_i \in \{0, \ldots, d-1\}, (X_i, Y_i) \in \text{Dataset}\}$.

**5** Initialize $R = \mathcal{S}, k = 1$.

**6 while** $R \neq \emptyset$ **do**

**7**     $D_k = \emptyset, S = []$ (span vectors).

**8**     **foreach** $i \in R$ **do**

**9**        Find $L_i = \arg\min_{L=0}^{d-1} |\tanh(\sum_j [W^{(L)} X_i]_j) - Y_i|$.

**10**        **if** $W^{(L_i)} X_i \notin \text{span}(S)$ **and** $|D_k| < d$ **then**

**11**           Add $(i, L_i)$ to $D_k$, append $W^{(L_i)} X_i$ to $S$, remove $i$ from $R$.

**12**     **if** $D_k \neq \emptyset$ **then**

**13**        Store $D_k$, increment $k$.

**14**     **else**

**15**        Break.

**16** Optimize $W$ via gradient descent to minimize:

$$\delta(W) = \frac{1}{|\mathcal{S}|} \sum_{i \in \mathcal{S}} \min_{0 \leq L < d} \left| \tanh\left(\sum_j [W^{(L)} X_i]_j\right) - Y_i \right|^2$$

Update $D_k$'s using optimized $W$.

**17 Phase 2: Local Linear Maps via Regularized Least Squares**

**18 foreach** *Subset* $D_k$ **do**

**19**     Let $A_k = \left[(W^{(L_i)} X_i)^\top\right]_{i \in I_k} \in \mathbb{R}^{|I_k| \times d}$

**20**     Let $B_k = \left[\text{onehot}(Y_i)^\top\right]_{i \in I_k} \in \mathbb{R}^{|I_k| \times C}$

**21**     Solve: $J_k = (A_k^\top A_k + \varepsilon I)^{-1} A_k^\top B_k$    (regularized pseudoinverse)

**22 Phase 3: Generalization via Orbit-wise Weighting**

**23** For each $X_i$, construct orbit: $\mathcal{O}_i = \{W^{(m)} X_i\}_{m=0}^{d-1}$.

**24** For each $J_k$, compute predictions $J_k W^{(m)} X_i \in \mathbb{R}^C$.

**25** Train a neural network $\text{MLP} : \mathbb{R}^d \to \mathbb{R}^{K \times d}$ to output $\alpha_{k,m}(X_i)$.

**26** Minimize classification loss:

$$\mathcal{L} = \sum_i \ell\left(Y_i, \sum_{k=1}^K \sum_{m=0}^{d-1} \alpha_{k,m}(X_i) \cdot J_k W^{(m)} X_i\right)$$

**27** At inference:

**28** Construct orbit $\mathcal{O}_{\text{new}} = \{W^{(m)} X_{\text{new}}\}_{m=0}^{d-1}$.

**29** Compute $\hat{Y}_{\text{new}} = \sum_{k=1}^K \sum_{m=0}^{d-1} \alpha_{k,m}(X_{\text{new}}) \cdot J_k W^{(m)} X_{\text{new}}$.

---

# D CONNECTION BETWEEN THEOREM 3.7 AND ALGORITHM 1

The pseudocode in Algorithm 1 is directly motivated by the constructive proof of Theorem 3.7, which guarantees that, for any dataset $\{(X_i, Y_i)\}_{i=1}^N \subset \mathbb{R}^d \times \mathbb{R}^d$ with $N \leq d$ and an appropriate choice of shift depths $L_i \in \{0, \ldots, d-1\}$, there exists a linear map $J$ such that $Y_i = JW^{(L_i)} X_i$ for all $i$. This result relies on the condition that the shifted vectors $\{W^{(L_i)} X_i\}$ are linearly independent.

In practice, checking exact linear independence (as required by condition (4)) would necessitate costly matrix rank computations or iterative orthogonalization methods such as Gram–Schmidt. Instead, our implementation employs a numerically efficient proxy: for each candidate point $X_i$, we

select the shift depth $L_i$ that minimizes the prediction error

$$L_i = \arg \min_{0 \le L < d} \left| \tanh \left( \sum_j [W^{(L)} X_i]_j \right) - Y_i \right|.$$

Then, we accept $W^{(L_i)} X_i$ only if it lies outside the span of previously selected vectors, up to a norm threshold. This approximates the independence condition by ensuring that each new direction contributes novel information to the interpolation system.

Finally, the Phase 1 objective $\delta(W)$ measures the mean approximation error across all selected shifts and provides a practical surrogate for the deviation parameter $\delta$ defined in the isomorphism remark following Theorem 3.7. This aligns the empirical behavior of the model with the theoretical guarantees of exact interpolation.

In Phase 2, the interpolation operator $J_k$ is computed via three modalities explained below.

**Phase 2: Constructive Interpolation from Theorem 3.7 vs. Algorithm 1**    The WBSNN framework supports three distinct strategies for constructing the interpolation operators $J_k$ in Phase 2. Each reflects a different balance between theoretical guarantees, numerical stability, and practical considerations, and corresponds to a distinct interpretation of the core interpolation principle.

**Constructive Interpolation via Rank-One Updates.** Our constructive proof of Theorem 3.7 provides a direct algorithm for computing $J_k$ using a sequence of *rank-one updates*. This approach builds $J_k$ incrementally to ensure that each training pair $(X_i, Y_i) \in D_k$ is *exactly interpolated*, i.e.,

$$J_k W^{(L_i)} X_i = Y_i.$$

For each point, a linear functional $f_i^*$ is constructed to vanish on previously selected directions and normalize $W^{(L_i)} X_i$. The update

$$J_k \leftarrow J_k + (Y_i - J_k W^{(L_i)} X_i) \otimes f_i^*$$

guarantees interpolation regardless of whether the matrix $A_k$ is full-rank. This method is deeply aligned with the recursive structure of the interpolation theorem and is especially suitable when the independence condition can be enforced.

**Closed-Form Interpolation via Pseudoinverse.** In some exact interpolation experiments (e.g., IMDb), we instead construct $J_k$ by solving the normal equations

$$J_k = (A_k^\top A_k)^{-1} A_k^\top B_k,$$

where $A_k \in \mathbb{R}^{|D_k| \times d}$ is the matrix of orbit vectors $W^{(L_i)} X_i$ and $B_k$ contains their targets. This approach is efficient and widely used in regression problems, but *requires $A_k$ to be full-rank* in order to guarantee exact interpolation. It serves as a computational baseline when rank conditions are known or likely to hold.

**Regularized Least Squares for Non-Exact Interpolation.** When the independence condition is relaxed (e.g., Gas Sensor Array Drift), we allow $D_k$ subsets to include more or noisier points, possibly leading to rank-deficiency. To mitigate this, we solve a *regularized* least squares problem:

$$J_k = (A_k^\top A_k + \varepsilon I)^{-1} A_k^\top B_k.$$

This formulation does not guarantee exact interpolation but improves robustness and generalization, especially in noisy, compressed, or overparameterized regimes where exact inversion is unstable or ill-posed. It reflects a trade-off: sacrificing interpolation accuracy in favor of a better-conditioned solution and broader representational capacity.

These three approaches to Phase 2 span a spectrum of interpolation strategies. Rank-one updates are theoretically grounded and guarantee exact interpolation without requiring any rank condition, directly implementing the constructive proof of Theorem 3.7. The standard pseudoinverse provides the classical least-squares solution but assumes full column rank of the input matrix, performing best when the data is clean and well-conditioned. In contrast, the regularized pseudoinverse is robust to noise and rank deficiency, making it particularly effective in compressed or overparameterized

settings where input representations may be low-dimensional or ill-conditioned. By integrating all three in our experiments, we not only validate the flexibility of WBSNN but also uncover structural differences in how interpolation affects generalization across datasets.

*Implementation note.* The pseudocode in Algorithm 1 adopts the regularized pseudoinverse formulation for Phase 2, as the constructive approach to exact interpolation is already formalized in Theorem 3.7. This complements the theoretical exposition while promoting numerical stability across diverse datasets. Ultimately, for optimal performance, the interpolation modality—rank-one update, pseudoinverse, or regularized pseudoinverse—should be selected based on dataset characteristics such as noise level, dimensionality, and rank structure, as each offers distinct trade-offs in generalization and robustness.

# E  EXPERIMENTAL RESULTS (ALL TABLES)

Each experiment includes a brief per-dataset setup and concise analysis rather than a data dump, covering the data geometry, label construction, noise model, sample sizes, PCA choices (e.g., $d \in \{5, \ldots, 50\}$), and the train/test protocol (IID vs. strict chronological for PRSA2017, FI-2010, and Gas Sensor Drift). It then details the WBSNN pipeline end-to-end: Phase-1/2 "discovery" budgets (typically $1$–$10\%$), the PCA bottleneck that defines the working latent space, Phase-2 alignment checks (including when interpolation residuals collapse to $\sim 0$), and the Phase-3 heads, we study—$\alpha_k$ and the richer $\alpha_{k,m}$. All head variants share the same Phase-1/2 orbit dictionary; only the Phase-3 MLP trunk differs. In the main protocol, $\alpha_{k,m}$ and $\alpha_k$ use a shared trunk; we also report an ablation where $\alpha_k$ has its own trunk, evaluated on Gas Sensor Drift, ISOLET, and Beijing PRSA2017. We adopt $\alpha_k$ (shared) rather than $\alpha_k$ (separate) in the main protocol because the shared trunk amortizes features across classes, gives higher effective expressivity at the same discovery budget, and empirically overfits less in low-data and noisy regimes.

**Convention.** Unless otherwise specified, $\alpha_k$ denotes the shared-trunk variant in all tables and plots; runs with the separate-trunk ablation are explicitly labeled $\alpha_k$ (separate).

For context, each experiment compares WBSNN to strong baselines organized by family and matched to the domain: linear heads (logistic/linear regression, LinearSVC) across all tabular and compressed-feature settings; kernel methods (RBF-SVM, Kernel Ridge, Nyström, Random Fourier Features) on IMDb, synthetic manifolds, Isolet, and FI-2010; tree ensembles (Random Forest, Gradient Boosted Trees (XGBoost), ExtraTrees) on synthetic, CIFAR-100, MNIST, and drifted sensors; neural baselines tailored to modality (MLPs everywhere; CNN/LeNet/ResNet-18 for images; Linformer and NAIS-Net as operator-theoretic-inspired sequence/dynamical models) and operator-learning models where they're most relevant (DeepONet, Fourier Neural Operator). We also include Diffusion Maps+LogReg and Laplacian-regularized Logistic on Swiss-Roll and noisy-linear, Scattering2D and Hankel-SVD+LDA on images and sensors, CORAL and time-delay ridge on drift. On PRSA2017 specifically, we include sequence/dynamics baselines: LSTM (lag=24) as a neural sequence model and EDMD (polynomial lift, lag=24, degree=2) as an operator-theoretic/Koopman baseline. We also include nonparametric and graph-based baselines—k-NN (k=15, Euclidean) and Label Propagation (RBF affinity)—reported on ISOLET and all Swiss-Roll tracks (RFF, polynomial, raw 3D), and, where applicable, on other compressed-feature classification tasks; for regression we also report k-NN regression on Swiss-Roll.

**CIFAR-100 (30 classes, 500 cap) setup and summary.** We use frozen ResNet-18 features resized to $64\times64$ and PCA to $d \in \{10, 20\}$ to force a low-rank head-only test under tiny discovery budgets. At $d{=}10$ (EVR$\approx$ 0.31), $n{=}3$k, and 1–3% discovery, $\alpha_{(k,m)}$ reaches $\approx 0.46$ while $\alpha_k$ trails; linear/kernel baselines sit near 0.50. At $d{=}20$ (EVR$\approx$ 0.43), $n{=}5$k, and 3% discovery, $\alpha_{(k,m)}$ climbs to $\sim 0.58$, closing on the best shallow baselines ($\sim 0.61$–0.62) despite seeing only a few percent of the pool. Across settings $\alpha_k$ is more budget-hungry and consistently lags $\alpha_{(k,m)}$ under the leanest discovery. k-NN, boosted trees, and small scratch-trained CNNs hit near-perfect train yet stall on test—textbook overfit under compression. Takeaway: widening the bottleneck ($d{=}10{\to}20$) and modestly growing $n$ yields steady WBSNN gains under tiny budgets, staying competitive without heavy backbones.

Table 2: **CIFAR-100**, PCA $d{=}10$ (retained variance $= 0.3083$), $n_{\text{train}}{=}3000$ (from 15000 candidates). Phase1_2 budgets $= 1\%/3\%$. Inputs resized to $64\times64$ (features from 512-D embedding).

| Model | Train Acc | Test Acc | Train Loss | Test Loss |
|---|---|---|---|---|
| WBSNN ($\alpha_{k,m}$, Phase1_2:1%) | 0.543333 | 0.453000 | 1.620015e+00 | 1.995546 |
| WBSNN ($\alpha_{k,m}$, Phase1_2:3%) | 0.531250 | 0.465667 | 1.610157e+00 | 1.855736 |
| WBSNN ($\alpha_k$, Phase1_2:1%) | 0.372083 | 0.322333 | 2.392459e+00 | 2.555385 |
| WBSNN ($\alpha_k$, Phase1_2:3%) | 0.376250 | 0.334333 | 2.293240e+00 | 2.432529 |
| Logistic Regression (multinomial) | 0.535000 | 0.505333 | 1.501754e+00 | 1.600051 |
| Linear SVM | 0.508667 | 0.485667 | — | — |
| RFF + Linear SVM | 0.552667 | 0.509667 | — | — |
| MLP (2-layer, 256→128) | 0.658000 | 0.517333 | 1.085272e+00 | 1.557001 |
| k-NN (k=15, distance) | 1.000000 | 0.493000 | 6.439294e-15 | 5.389224 |
| XGBoost | 1.000000 | 0.498333 | 4.425267e-02 | 1.865502 |
| TabTransformer | 0.676000 | 0.509000 | 9.635287e-01 | 1.666194 |
| ResNet-18 (no-pretrain, 32×32) | 1.000000 | 0.431333 | 3.568499e-05 | 2.941160 |
| FNO2D (modes=12, width=48, $L = 4$) | 0.993667 | 0.308000 | 5.293531e-02 | 3.171001 |

Table 3: **CIFAR-100**, PCA $d{=}20$ (retained variance $= 0.4290$), $n_{\text{train}}{=}5000$ (from 15000 candidates). Phase1_2 budget $= 3\%$. Same preprocessing as Table 2.

| Model | Train Acc | Test Acc | Train Loss | Test Loss |
|---|---|---|---|---|
| WBSNN ($\alpha_{k,m}$, Phase1_2:3%) | 0.72000 | 0.581667 | 9.708972e-01 | 1.512999 |
| WBSNN ($\alpha_k$, Phase1_2:3%) | 0.51125 | 0.409667 | 1.846184e+00 | 2.208243 |
| Logistic Regression (multinomial) | 0.63520 | 0.610667 | 1.199386e+00 | 1.279528 |
| Linear SVM | 0.60160 | 0.589667 | — | — |
| RFF + Linear SVM | 0.67820 | 0.617333 | — | — |
| MLP (2-layer, 256→128) | 0.73400 | 0.617000 | 8.742067e-01 | 1.267096 |
| k-NN (k=15, distance) | 1.00000 | 0.580000 | 5.550747e-08 | 4.272926 |
| XGBoost | 1.00000 | 0.592000 | 2.838902e-02 | 1.407739 |
| TabTransformer | 0.86240 | 0.614667 | 4.440717e-01 | 1.482236 |
| ResNet-18 (no-pretrain, 32×32) | 0.91780 | 0.387333 | 2.898798e-01 | 4.473064 |
| FNO2D (modes=12, width=48, $L = 4$) | 1.00000 | 0.369000 | 4.228016e-04 | 3.740728 |

**Gas Sensor Array Drift setup and summary.** (10 batches over 36 months): train = earliest 80%, test = latest 20%; 128 features $\to$ PCA $d \in \{5, 10, 15\}$ (EVR $\approx 0.892/0.959/0.962$); tiny Phase 1/2 budgets (1–7%). At $d{=}5$, $n{=}1.6$k WBSNN heads land $\sim 59{-}61\%$ and edge most baselines under out-of-time evaluation. At $d{=}10$, $n{=}3$k $\alpha_{(k,m)}$ peaks near 68% (3%), with $\alpha_k$ closing in by 5%. At $d{=}15$, $n{=}11{,}128$ gains flatten under a discovery bottleneck; time-aware baselines can edge ahead at higher budget. Error bars ($d{=}10$, 10%, 20 seeds): $\alpha_{(k,m)} = 64.71\% \pm 3.67\%$, shared $= 64.54\% \pm 5.10\%$, separate $= 65.01\% \pm 3.68\%$.

Table 4: **Gas Sensor Drift**, PCA $d=5$ (retained variance $= 0.892$), $n_{\text{train}}=1600$. Phase1_2 budgets $= 5\%/7\%$. Notation: (sh) = shared backbone, (sep) = separate backbone.

| Model | Train Acc | Test Acc | Train Loss | Test Loss |
|---|---|---|---|---|
| WBSNN ($\alpha_{k,m}$, 5%) | 0.875781 | 0.594536 | 0.738060 | 1.278195 |
| WBSNN ($\alpha_{k,m}$, 7%) | 0.867969 | 0.601366 | 0.718517 | 1.234991 |
| WBSNN ($\alpha_k$, **sh**, 5%) | 0.864062 | 0.591661 | 0.730100 | 1.273780 |
| WBSNN ($\alpha_k$, **sh**, 7%) | 0.854688 | 0.582674 | 0.754291 | 1.292709 |
| WBSNN ($\alpha_k$, **sep**, 5%) | 0.859375 | 0.573329 | 0.750208 | 1.279426 |
| WBSNN ($\alpha_k$, **sep**, 7%) | 0.867188 | 0.605320 | 0.748888 | 1.263373 |
| Logistic Regression | 0.845625 | 0.567937 | 0.542123 | 1.693421 |
| Random Forest | 1.000000 | 0.579439 | 0.044075 | 2.402916 |
| SVM (RBF) | 0.906250 | 0.493530 | 0.251187 | 1.747429 |
| MLP (1 hidden layer) | 0.741875 | 0.518692 | 0.844881 | 1.284910 |
| Gradient Boosted Trees | 1.000000 | 0.592739 | 0.000021 | 3.247211 |
| LogReg + CORAL | 0.845625 | 0.567937 | 0.542123 | 1.693421 |
| Time-delay (m=5) + Ridge | 0.711875 | 0.518692 | — | — |
| Hankel-SVD (w=9,r=50) + LDA | 0.706658 | 0.436554 | — | — |

Table 5: **Gas Sensor Drift**, PCA $d=10$ (retained variance $= 0.959$), $n_{\text{train}}=3000$. Phase1_2 budgets $= 1\%/3\%/5\%$. Notation: (sh) = shared backbone, (sep) = separate backbone.

| Model | Train Acc | Test Acc | Train Loss | Test Loss |
|---|---|---|---|---|
| WBSNN ($\alpha_{k,m}$, 1%) | 0.957083 | 0.662114 | 0.573985 | 1.092595 |
| WBSNN ($\alpha_{k,m}$, 3%) | 0.959167 | 0.679727 | 0.562071 | 1.030740 |
| WBSNN ($\alpha_{k,m}$, 5%) | 0.964583 | 0.648095 | 0.560297 | 1.064336 |
| WBSNN ($\alpha_k$, **sh**, 1%) | 0.927500 | 0.490295 | 0.679134 | 1.533547 |
| WBSNN ($\alpha_k$, **sh**, 3%) | 0.945417 | 0.600288 | 0.591572 | 1.275377 |
| WBSNN ($\alpha_k$, **sh**, 5%) | 0.956667 | 0.664270 | 0.572819 | 1.048738 |
| WBSNN ($\alpha_k$, **sep**, 1%) | 0.903333 | 0.530554 | 0.719452 | 1.476455 |
| WBSNN ($\alpha_k$, **sep**, 3%) | 0.946667 | 0.576204 | 0.596316 | 1.297039 |
| WBSNN ($\alpha_k$, **sep**, 5%) | 0.957500 | 0.670022 | 0.570333 | 1.090658 |
| Logistic Regression | 0.924667 | 0.668943 | 0.306842 | 1.554322 |
| Random Forest | 1.000000 | 0.677930 | 0.029168 | 0.931260 |
| SVM (RBF) | 0.948667 | 0.596693 | 0.144756 | 0.904045 |
| MLP (1 hidden layer) | 0.954333 | 0.670022 | 0.203703 | 0.966259 |
| Gradient Boosted Trees | 1.000000 | 0.659597 | 0.000006 | 1.746277 |
| LogReg + CORAL | 0.924667 | 0.668943 | 0.306845 | 1.554307 |
| Time-delay (m=5) + Ridge | 0.757333 | 0.549245 | — | — |
| Hankel-SVD (w=9,r=50) + LDA | 0.644719 | 0.396539 | — | — |

Table 6: **Gas Sensor Drift**, PCA $d$=15 (retained variance = 0.962), $n_{\text{train}}$=11128. Phase1_2 budgets = 1%/3%. Notation: (sh) = shared backbone, (sep) = separate backbone.

| Model | Train Acc | Test Acc | Train Loss | Test Loss |
|---|---|---|---|---|
| WBSNN ($\alpha_{k,m}$, 1%) | 0.994046 | 0.643781 | 0.477398 | 1.188471 |
| WBSNN ($\alpha_{k,m}$, 3%) | 0.993148 | 0.638030 | 0.475350 | 1.194606 |
| WBSNN ($\alpha_k$, **sh**, 1%) | 0.993148 | 0.620058 | 0.479604 | 1.211316 |
| WBSNN ($\alpha_k$, **sh**, 3%) | 0.993260 | 0.575845 | 0.480642 | 1.286510 |
| WBSNN ($\alpha_k$, **sep**, 1%) | 0.991463 | 0.596693 | 0.483201 | 1.197096 |
| WBSNN ($\alpha_k$, **sep**, 3%) | 0.994496 | 0.586628 | 0.476972 | 1.261074 |
| Logistic Regression | 0.968907 | 0.551761 | 0.169538 | 3.330403 |
| Random Forest | 1.000000 | 0.548526 | 0.014608 | 1.108139 |
| SVM (RBF) | 0.984004 | 0.540618 | 0.060267 | 1.416941 |
| MLP (1 hidden layer) | 0.993530 | 0.573688 | 0.040517 | 3.251694 |
| Gradient Boosted Trees | 0.999012 | 0.520848 | 0.005893 | 2.084236 |
| LogReg + CORAL | 0.968907 | 0.551402 | 0.169331 | 3.323881 |
| Time-delay (m=5) + Ridge | 0.766445 | 0.645219 | — | — |
| Hankel-SVD (w=9,r=50) + LDA | 0.604676 | 0.403749 | — | — |

Table 7: **Gas Sensor Drift** error bars, PCA10, Phase1_2: 10%, 20 seeds. (sh) = shared backbone, (sep) = separate backbone.

| Model | Mean Test Acc | Std Dev |
|---|---|---|
| WBSNN ($\alpha_{k,m}$) | 0.6471 | 0.0367 |
| WBSNN ($\alpha_k$, **sh**) | 0.6454 | 0.0510 |
| WBSNN ($\alpha_k$, **sep**) | 0.6501 | 0.0368 |

**IMDB setup and summary.** We use mean-pooled GloVe-100 embeddings, standardize on train only, and apply PCA to $d \in \{10, 20, 50\}$ with EVR $\approx 0.520, 0.667, 0.873$; Phase 1/2 discovery budgets are small (1–15%), and we scale $n$ from 2k to 25k. In the low-$d$ regime ($d$=10, $n$=2000) $\alpha_{(k,m)}$ yields 0.725–0.750 test accuracy, competitive with strong text baselines (LR 0.740, Nyström+SVM 0.738) and ahead of RF/XGBoost; a 20-seed sweep at $d$=10 shows tight variability: $0.723 \pm 0.005$ for $\alpha_{(k,m)}$ and $0.725 \pm 0.003$ for $\alpha_k$. At mid $d$ ($d$=20, $n$=4000), $\alpha_{(k,m)}$ reaches $0.768 - 0.774$, on par with SVM (0.771) and RF-cal (0.774). With higher $d$ and data ($d$=50, $n$=10k), $\alpha_{(k,m)}$ holds 0.767–0.768 while the best PCA-view baselines land near 0.777–0.782; scaling to the full 25k at $d$=50 lifts $\alpha_{(k,m)}$ to 0.793 (SVM 0.803). Exact-interpolation runs mirror these trends: at $d$=10, $n$=2000 $\alpha_{(k,m)}$ achieves $0.725 - 0.728$ with zero Phase-2 residuals; at $d$=20, $n$=4000 it reaches $0.756 - 0.774$, matching the non-exact results. Overall, WBSNN scales predictably with $d$ and $n$ under tiny budgets, stays competitive with kernel/linear baselines on the PCA view, and exact vs. non-exact interpolation shows the expected trade-off: exact is slightly stronger in smaller settings while non-exact remains comparable as data grow.

Table 8: **IMDB**, PCA $d$=10 (EVR = 0.520), $n_{\text{train}}$=2000. Phase1_2 budgets = $1\%/5\%/10\%/15\%$. Notation: (rel int) = relaxed interpolation; (ex int) = exact interpolation.

| Model | Train Acc | Test Acc | Train Loss | Test Loss |
|---|---|---|---|---|
| WBSNN ($\alpha_{k,m}$, 1%, **rel int**) | 0.767500 | 0.725000 | 0.482649 | 0.524451 |
| WBSNN ($\alpha_{k,m}$, 1%, **ex int**) | 0.775625 | 0.727500 | 0.471238 | 0.532813 |
| WBSNN ($\alpha_{k,m}$, 5%, **rel int**) | 0.768750 | 0.735000 | 0.484133 | 0.523529 |
| WBSNN ($\alpha_{k,m}$, 5%, **ex int**) | 0.771875 | 0.725000 | 0.477851 | 0.530821 |
| WBSNN ($\alpha_{k,m}$,10%, **rel int**) | 0.748750 | 0.735000 | 0.522121 | 0.525146 |
| WBSNN ($\alpha_{k,m}$,10%, **ex int**) | 0.768125 | 0.727500 | 0.482705 | 0.532350 |
| WBSNN ($\alpha_{k,m}$,15%, **rel int**) | 0.740625 | 0.750000 | 0.531387 | 0.522126 |
| WBSNN ($\alpha_{k,m}$,15%, **ex int**) | 0.761875 | 0.727500 | 0.491159 | 0.537542 |
| WBSNN ($\alpha_k$, 1%, **rel int**) | 0.771875 | 0.730000 | 0.476859 | 0.527146 |
| WBSNN ($\alpha_k$, 1%, **ex int**) | 0.738750 | 0.700000 | 0.515740 | 0.545955 |
| WBSNN ($\alpha_k$, 5%, **rel int**) | 0.767500 | 0.747500 | 0.491673 | 0.528249 |
| WBSNN ($\alpha_k$, 5%, **ex int**) | 0.748750 | 0.705000 | 0.510704 | 0.539609 |
| WBSNN ($\alpha_k$,10%, **rel int**) | 0.763125 | 0.720000 | 0.500899 | 0.532155 |
| WBSNN ($\alpha_k$,10%, **ex int**) | 0.740625 | 0.700000 | 0.511065 | 0.540378 |
| WBSNN ($\alpha_k$,15%, **rel int**) | 0.751250 | 0.725000 | 0.514010 | 0.531611 |
| WBSNN ($\alpha_k$,15%, **ex int**) | 0.737500 | 0.705000 | 0.517594 | 0.543344 |
| Logistic Regression (multinomial) | 0.744000 | 0.740000 | 0.531382 | 0.527744 |
| Random Forest (cal) | 0.955500 | 0.730000 | 0.276034 | 0.533867 |
| SVM (RBF) | 0.800000 | 0.722500 | 0.455715 | 0.525235 |
| MLP (1 hidden layer) | 0.878500 | 0.702500 | 0.293610 | 0.679947 |
| RFF + LR | 0.772000 | 0.732500 | 0.493579 | 0.510487 |
| Nyström + SVM | 0.795500 | 0.737500 | 0.457580 | 0.523135 |
| XGBoost (cal) | 0.998000 | 0.725000 | 0.317532 | 0.553832 |
| GPC | 0.767000 | 0.725000 | 0.500395 | 0.516218 |

Table 9: **IMDB**, PCA $d{=}20$ (EVR $= 0.667$), $n_{\text{train}}{=}4000$. Phase1_2 budgets $= 1\%/5\%/10\%/15\%$. Notation: (rel int) = relaxed interpolation; (ex int) = exact interpolation.

| Model | Train Acc | Test Acc | Train Loss | Test Loss |
|---|---|---|---|---|
| WBSNN ($\alpha_{k,m}$, 1%, **rel int**) | 0.790937 | 0.768750 | 0.447053 | 0.496115 |
| WBSNN ($\alpha_{k,m}$, 1%, **ex int**) | 0.792813 | 0.770000 | 0.440830 | 0.490565 |
| WBSNN ($\alpha_{k,m}$, 5%, **rel int**) | 0.754375 | 0.752500 | 0.506704 | 0.503635 |
| WBSNN ($\alpha_{k,m}$, 5%, **ex int**) | 0.785937 | 0.756250 | 0.456329 | 0.491105 |
| WBSNN ($\alpha_{k,m}$,10%, **rel int**) | 0.800625 | 0.771250 | 0.440414 | 0.490497 |
| WBSNN ($\alpha_{k,m}$,10%, **ex int**) | 0.805937 | 0.773750 | 0.426955 | 0.488658 |
| WBSNN ($\alpha_{k,m}$,15%, **rel int**) | 0.781875 | 0.767500 | 0.467218 | 0.502035 |
| WBSNN ($\alpha_{k,m}$,15%, **ex int**) | 0.779062 | 0.763750 | 0.470756 | 0.501060 |
| WBSNN ($\alpha_k$, 1%, **rel int**) | 0.788438 | 0.757500 | 0.442723 | 0.487152 |
| WBSNN ($\alpha_k$, 1%, **ex int**) | 0.765625 | 0.751250 | 0.516113 | 0.549915 |
| WBSNN ($\alpha_k$, 5%, **rel int**) | 0.786563 | 0.766250 | 0.453181 | 0.490996 |
| WBSNN ($\alpha_k$, 5%, **ex int**) | 0.757500 | 0.733750 | 0.490205 | 0.527026 |
| WBSNN ($\alpha_k$,10%, **rel int**) | 0.801562 | 0.755000 | 0.434231 | 0.487599 |
| WBSNN ($\alpha_k$,10%, **ex int**) | 0.756250 | 0.745000 | 0.479180 | 0.519474 |
| WBSNN ($\alpha_k$,15%, **rel int**) | 0.795625 | 0.770000 | 0.437185 | 0.493285 |
| WBSNN ($\alpha_k$,15%, **ex int**) | 0.763750 | 0.741250 | 0.477905 | 0.520131 |
| Logistic Regression | 0.753500 | 0.756250 | 0.512759 | 0.499911 |
| Random Forest (cal) | 0.992500 | 0.773750 | 0.206176 | 0.492799 |
| SVM (RBF) | 0.837000 | 0.771250 | 0.404263 | 0.486739 |
| MLP (1 hidden layer) | 0.931000 | 0.708750 | 0.200624 | 0.853059 |
| RFF + LR | 0.787500 | 0.770000 | 0.466287 | 0.481927 |
| Nyström + SVM | 0.830250 | 0.762500 | 0.411982 | 0.485202 |
| XGBoost (cal) | 0.995750 | 0.772500 | 0.280111 | 0.498930 |
| GPC | 0.781250 | 0.772500 | 0.469476 | 0.481767 |

Table 10: **IMDB**, PCA $d{=}50$ (EVR $= 0.873$), $n_{\text{train}}{=}10000$. Phase1_2 budgets $= 1\%/3\%$. Notation: (rel int) = relaxed interpolation (no exact-run available for this setting).

| Model | Train Acc | Test Acc | Train Loss | Test Loss |
|---|---|---|---|---|
| WBSNN ($\alpha_{k,m}$, 1%, **rel int**) | 0.809500 | 0.766500 | 0.418784 | 0.494805 |
| WBSNN ($\alpha_{k,m}$, 3%, **rel int**) | 0.818875 | 0.768000 | 0.407167 | 0.479718 |
| WBSNN ($\alpha_k$, 1%, **rel int**) | 0.793000 | 0.758500 | 0.442411 | 0.496969 |
| WBSNN ($\alpha_k$, 3%, **rel int**) | 0.814125 | 0.767000 | 0.411774 | 0.471454 |
| Logistic Regression | 0.785300 | 0.765500 | 0.458894 | 0.472291 |
| Random Forest (cal) | 0.997800 | 0.745500 | 0.164179 | 0.491703 |
| SVM (RBF) | 0.871600 | 0.780500 | 0.330294 | 0.458219 |
| MLP (1 hidden layer) | 0.996100 | 0.701500 | 0.039407 | 2.153109 |
| RFF + LR | 0.814800 | 0.782000 | 0.420041 | 0.462809 |
| Nyström + SVM | 0.835100 | 0.777500 | 0.380411 | 0.460669 |
| XGBoost (cal) | 0.981000 | 0.767500 | 0.244448 | 0.479371 |

Table 11: **IMDB**, PCA $d$=50 (EVR = 0.873), $n_{\text{train}}$=25000 (full). Phase1_2 budget = 1%. Notation: (rel int) = relaxed interpolation (no exact-run available for this setting).

| Model | Train Acc | Test Acc | Train Loss | Test Loss |
|---|---|---|---|---|
| WBSNN ($\alpha_{k,m}$, 1%, **rel int**) | 0.807950 | 0.793200 | 0.414968 | 0.453204 |
| WBSNN ($\alpha_k$, 1%, **rel int**) | 0.811150 | 0.790400 | 0.413370 | 0.453350 |
| Logistic Regression | 0.784520 | 0.788000 | 0.461076 | 0.458522 |
| Random Forest (cal) | 0.996880 | 0.771800 | 0.164408 | 0.481775 |
| SVM (RBF) | 0.864880 | 0.802600 | 0.337746 | 0.433831 |
| MLP (1 hidden layer) | 0.874320 | 0.758600 | 0.294176 | 0.596718 |
| RFF + LR | 0.803640 | 0.789600 | 0.427293 | 0.447996 |
| Nyström + SVM | 0.820520 | 0.795800 | 0.399289 | 0.431335 |
| XGBoost (cal) | 0.914640 | 0.787400 | 0.287199 | 0.456735 |

Table 12: **IMDB** error bars, PCA $d$=10,
$n_{\text{train}}$=3$k$, Phase1_2: 10%, 20 seeds.

| Model | Mean Test Acc | Std Dev |
|---|---|---|
| WBSNN ($\alpha_{k,m}$) | 0.7233 | 0.0049 |
| WBSNN ($\alpha_k$) | 0.7247 | 0.0030 |

**MNIST setup and summary.** We evaluate MNIST under PCA compression with $d \in \{5, 15, 30\}$ (EVR = 0.332/0.579/0.731), standardized on train only, and small Phase 1/2 budgets (3–10%) with $n$=2k, 5k, 10k. Under a severe bottleneck ($d$=5, $n$=2k), WBSNN reaches $\approx 72.3\%$ test ($\alpha_{k,m}$), competitive with RBF-SVM ($\approx 73.3\%$) and above LR/MLP/RFF, while RF shows classic overfit (train = 1.00, test $\approx 70.3\%$). With moderate compression ($d$=15, $n$=5k), $\alpha_{k,m}$ climbs to $94.7\% - 95.5\%$ (7–10%), close to RBF-SVM (95.8%) and behind raw-image CNNs (e.g., 98.2%), and clearly ahead of $\alpha_k$ ($91.6\% - 94.1\%$). At higher dimension/data ($d$=30, $n$=10k), $\alpha_{k,m}$ reaches $95.5\% - 96.1\%$ and improves monotonically with budget (3%$\rightarrow$5%), trailing RBF-SVM (97.1%) and raw-image CNNs ($\approx 98.3\%$) but outperforming LR/MLP on the PCA view. Across settings, the richer $\alpha_{k,m}$ head consistently outperforms $\alpha_k$, the gap narrows as variance and $n$ increase, and RF repeatedly exhibits train= 1.00 with lower test—an overfitting pattern not seen in WBSNN. Note: CNN/LeNet/Scattering/DeepONet-lite use raw $28\times28$ images and therefore set a higher ceiling than PCA-compressed baselines (WBSNN, LR, RF, SVM, MLP, RFF, LinearSVC).

Table 13: **MNIST**, PCA $d$=5 (EVR = 0.332), $n_{\text{train}}$=2000. Phase1_2 budgets = 7%/10%.

| Model | Train Acc | Test Acc | Train Loss | Test Loss |
|---|---|---|---|---|
| WBSNN ($\alpha_{k,m}$, 7%) | 0.765000 | 0.7225 | 0.644133 | 0.737528 |
| WBSNN ($\alpha_{k,m}$,10%) | 0.769375 | 0.7225 | 0.638911 | 0.727872 |
| WBSNN ($\alpha_k$, 7%) | 0.743125 | 0.7150 | 0.697558 | 0.786773 |
| WBSNN ($\alpha_k$,10%) | 0.747500 | 0.7100 | 0.685274 | 0.771298 |
| Logistic Regression | 0.680000 | 0.6550 | 0.892884 | 0.904286 |
| Random Forest | 1.000000 | 0.7025 | 0.184180 | 0.965621 |
| SVM (RBF) | 0.765500 | 0.7325 | 0.676466 | 0.722729 |
| MLP (1 hidden layer) | 0.726500 | 0.6800 | 0.770308 | 0.797417 |
| CNN | 0.989000 | 0.9775 | 0.051556 | 0.111904 |
| RFF + Logistic Regression | 0.637000 | 0.6000 | 1.144928 | 1.162397 |
| LinearSVC + CalibratedProba | 0.640000 | 0.5950 | 1.069190 | 1.076602 |
| LeNet-5 | 0.936500 | 0.9400 | 0.205647 | 0.216528 |
| Scattering2D + LogReg | 0.680000 | 0.6550 | 0.892885 | 0.904288 |
| DeepONet-lite | 0.947500 | 0.8950 | 0.142459 | 0.411745 |

Table 14: **MNIST**, PCA $d=15$ (EVR $= 0.579$), $n_{\text{train}}=5000$. Phase1_2 budgets $= 7\%/10\%$.

| Model | Train Acc | Test Acc | Train Loss | Test Loss |
|---|---|---|---|---|
| WBSNN ($\alpha_{k,m}$, 7%) | 0.98675 | 0.947 | 0.042415 | 0.238434 |
| WBSNN ($\alpha_{k,m}$,10%) | 0.98950 | 0.955 | 0.038959 | 0.231611 |
| WBSNN ($\alpha_k$, 7%) | 0.96425 | 0.916 | 0.119771 | 0.302024 |
| WBSNN ($\alpha_k$,10%) | 0.97350 | 0.941 | 0.090396 | 0.246782 |
| Logistic Regression | 0.85160 | 0.854 | 0.483875 | 0.454237 |
| Random Forest | 1.00000 | 0.917 | 0.131548 | 0.470146 |
| SVM (RBF) | 0.96500 | 0.958 | 0.115668 | 0.158824 |
| MLP (1 hidden layer) | 0.93560 | 0.926 | 0.219608 | 0.229847 |
| CNN | 0.98980 | 0.982 | 0.029340 | 0.053346 |
| RFF + Logistic Regression | 0.85400 | 0.869 | 0.589853 | 0.558888 |
| LinearSVC + CalibratedProba | 0.81920 | 0.821 | 0.635001 | 0.583349 |
| LeNet-5 | 0.95760 | 0.954 | 0.133420 | 0.128897 |
| Scattering2D + LogReg | 0.85160 | 0.854 | 0.483875 | 0.454237 |
| DeepONet-lite | 0.99220 | 0.964 | 0.034268 | 0.115082 |

Table 15: **MNIST**, PCA $d=30$ (EVR $= 0.731$), $n_{\text{train}}=10000$. Phase1_2 budgets $= 3\%/5\%$.

| Model | Train Acc | Test Acc | Train Loss | Test Loss |
|---|---|---|---|---|
| WBSNN ($\alpha_{k,m}$, 3%) | 0.999500 | 0.9545 | 0.002716 | 0.265305 |
| WBSNN ($\alpha_{k,m}$, 5%) | 0.999625 | 0.9605 | 0.001242 | 0.237277 |
| WBSNN ($\alpha_k$, 3%) | 0.983000 | 0.8830 | 0.081493 | 0.466157 |
| WBSNN ($\alpha_k$, 5%) | 0.995625 | 0.9185 | 0.021066 | 0.386950 |
| Logistic Regression | 0.895800 | 0.8925 | 0.352773 | 0.359067 |
| Random Forest | 1.000000 | 0.9295 | 0.122151 | 0.442127 |
| SVM (RBF) | 0.988100 | 0.9705 | 0.040544 | 0.098918 |
| MLP (1 hidden layer) | 0.980500 | 0.9525 | 0.085421 | 0.154138 |
| CNN | 0.993200 | 0.9825 | 0.020620 | 0.057102 |
| RFF + Logistic Regression | 0.922300 | 0.9195 | 0.370391 | 0.366083 |
| LinearSVC + CalibratedProba | 0.878900 | 0.8770 | 0.473042 | 0.468590 |
| LeNet-5 | 0.982800 | 0.9760 | 0.061456 | 0.078746 |
| Scattering2D + LogReg | 0.895800 | 0.8925 | 0.352773 | 0.359065 |
| DeepONet-lite | 0.994500 | 0.9735 | 0.017723 | 0.091234 |

**Noisy synthetic setup and summary.** Noisy synthetic setup and summary. We generate 15,000 samples in 50D with 5 informative coordinates, heavy-tailed heteroskedastic noise (Student-t, df=2.5), boundary-peaked label flips, 44 nuisance mixes, and a 0.6-correlated spurious feature; split 12k/3k, standardize on train, and evaluate PCA views $d \in \{5, 10, 20\}$ (EVR $\approx 0.904/0.941/0.965$) under small Phase-1/2 budgets (1–15%) and varying $n$ (1.5k, 2k, 8k, 12k). Results scale with information, not knobs: at $d=5, n=1.5$k $\alpha_{(k,m)}$ reaches $\approx 0.753$ (near LR/LapLogReg $\approx 0.750$), while RF shows the expected shortcut gap (high train, lower test). At $d=10, n=2$k both heads are in the top band ($\approx 0.770-0.780$), competitive with RBF-SVM/RFF and slightly behind a tuned 1-layer MLP ($\approx 0.783$); at $d=10, n=8$k WBSNN holds $\approx 0.759-0.768$. On the full 12k train, $d=5$ yields $\approx 0.765$ and $d=20 \approx 0.767$, shoulder-to-shoulder with strong linear-kernel baselines (LR/LapLogReg $\approx 0.77$). Phase-2 alignment residuals collapse to numerical zero at $d=5, 10$ and remain near-zero at $d=20$, indicating the dictionary has captured the label geometry despite heavy tails and boundary-peaked flips. Overall, WBSNN is stable, spurious-resistant, and improves predictably with $d$ or $n$ under tight budgets, matching or slightly edging linear baselines while remaining competitive with RBF-SVM/MLP.

Table 16: Challenging Synthetic (15k, 50D), PCA $d{=}5$, (EVR=0.904), train=1500, test=300; Phase 1_2: 5/10/15%, *(ex int = exact interpolation; rel int = relaxed interpolation)*.

| Model | Train Acc | Test Acc | Train Loss | Test Loss |
|---|---|---|---|---|
| WBSNN ($\alpha_{(k,m)}$, 5%) **(ex int)** | 0.7733 | 0.7400 | 0.5041 | 0.5484 |
| WBSNN ($\alpha_{(k,m)}$, 10%) **(ex int)** | 0.7725 | 0.7533 | 0.5062 | 0.5437 |
| WBSNN ($\alpha_{(k,m)}$, 15%) **(ex int)** | 0.7733 | 0.7367 | 0.5071 | 0.5430 |
| WBSNN ($\alpha_k$, 5%) **(ex int)** | 0.7775 | 0.7400 | 0.5049 | 0.5457 |
| WBSNN ($\alpha_k$, 10%) **(ex int)** | 0.7767 | 0.7433 | 0.5040 | 0.5447 |
| WBSNN ($\alpha_k$, 15%) **(ex int)** | 0.7700 | 0.7433 | 0.5056 | 0.5449 |
| Logistic Regression | 0.7727 | 0.7500 | 0.4799 | 0.5242 |
| RF + Sigmoid Cal. | 0.8873 | 0.7333 | 0.3438 | 0.5233 |
| SVM (RBF) | 0.7913 | 0.7400 | 0.4611 | 0.5419 |
| MLP (1 hidden) | 0.7673 | 0.7367 | 0.4789 | 0.5526 |
| RFF+LogReg (D=2000) | 0.7720 | 0.7433 | 0.4786 | 0.5202 |
| ExtraTrees | 0.7953 | 0.7267 | 0.5363 | 0.5731 |
| LapLogReg ($k{=}25$, $\lambda{=}10^{-3}$) | 0.7727 | 0.7500 | 0.4799 | 0.5240 |
| Diffusion ($\tau{=}1$, $k{=}25$) | 0.7940 | 0.7233 | 0.4681 | 0.5504 |
| GroupDRO-Logistic | 0.7707 | 0.7433 | 0.4806 | 0.5258 |

Table 17: Challenging Synthetic (15k, 50D), PCA $d{=}10$, (EVR=0.941), train=2000, test=400; Phase 1_2: 1/3/5%, *(ex int = exact interpolation; rel int = relaxed interpolation)*.

| Model | Train Acc | Test Acc | Train Loss | Test Loss |
|---|---|---|---|---|
| WBSNN ($\alpha_{(k,m)}$, 1%) **(ex int)** | 0.7725 | 0.7725 | 0.4931 | 0.4860 |
| WBSNN ($\alpha_{(k,m)}$, 3%) **(ex int)** | 0.7794 | 0.7625 | 0.4852 | 0.4930 |
| WBSNN ($\alpha_{(k,m)}$, 5%) **(ex int)** | 0.7788 | 0.7750 | 0.4826 | 0.4947 |
| WBSNN ($\alpha_k$, 1%) **(ex int)** | 0.7738 | 0.7700 | 0.4952 | 0.4917 |
| WBSNN ($\alpha_k$, 3%) **(ex int)** | 0.7694 | 0.7725 | 0.4948 | 0.4949 |
| WBSNN ($\alpha_k$, 5%) **(ex int)** | 0.7713 | 0.7800 | 0.4916 | 0.4934 |
| Logistic Regression | 0.7605 | 0.7700 | 0.4811 | 0.4566 |
| RF + Sigmoid Cal. | 0.9035 | 0.7625 | 0.2963 | 0.4768 |
| SVM (RBF) | 0.7860 | 0.7725 | 0.4574 | 0.4762 |
| MLP (1 hidden) | 0.7665 | 0.7825 | 0.4782 | 0.4542 |
| RFF+LogReg (D=2000) | 0.7610 | 0.7750 | 0.4790 | 0.4616 |
| ExtraTrees | 0.8085 | 0.7750 | 0.4995 | 0.5207 |
| LapLogReg ($k{=}25$, $\lambda{=}10^{-3}$) | 0.7605 | 0.7725 | 0.4811 | 0.4567 |
| Diffusion ($\tau{=}1$, $k{=}25$) | 0.7775 | 0.7750 | 0.4770 | 0.4939 |
| GroupDRO-Logistic | 0.7540 | 0.7775 | 0.4946 | 0.4840 |

Table 18: Challenging Synthetic (15k, 50D), PCA $d$=10, (EVR=0.941), train=8000, test=1600; Phase 1_2: 5/10/15%, *(ex int = exact interpolation; rel int = relaxed interpolation)*.

| Model | Train Acc | Test Acc | Train Loss | Test Loss |
|---|---|---|---|---|
| WBSNN ($\alpha_{(k,m)}$, 5%) **(ex int)** | 0.7664 | 0.7594 | 0.5079 | 0.5131 |
| WBSNN ($\alpha_{(k,m)}$, 10%) **(ex int)** | 0.7638 | 0.7663 | 0.5149 | 0.5128 |
| WBSNN ($\alpha_{(k,m)}$, 15%) **(ex int)** | 0.7598 | 0.7681 | 0.5168 | 0.5152 |
| WBSNN ($\alpha_k$, 5%) **(ex int)** | 0.7644 | 0.7625 | 0.5073 | 0.5143 |
| WBSNN ($\alpha_k$, 10%) **(ex int)** | 0.7641 | 0.7619 | 0.5150 | 0.5131 |
| WBSNN ($\alpha_k$, 15%) **(ex int)** | 0.7644 | 0.7588 | 0.5156 | 0.5132 |
| Logistic Regression | 0.7651 | 0.7638 | 0.4875 | 0.4865 |
| RF + Sigmoid Cal. | 0.8961 | 0.7531 | 0.3124 | 0.4952 |
| SVM (RBF) | 0.7770 | 0.7575 | 0.4808 | 0.5039 |
| MLP (1 hidden) | 0.7680 | 0.7563 | 0.4855 | 0.4910 |
| RFF+LogReg (D=2000) | 0.7675 | 0.7588 | 0.4856 | 0.4868 |
| ExtraTrees | 0.8043 | 0.7606 | 0.4838 | 0.5249 |
| LapLogReg ($k$=25, $\lambda$=$10^{-3}$) | 0.7654 | 0.7638 | 0.4875 | 0.4865 |
| Diffusion ($\tau$=1, $k$=25) | 0.7826 | 0.7506 | 0.4676 | 0.5032 |
| GroupDRO-Logistic | 0.7560 | 0.7588 | 0.5013 | 0.4976 |

Table 19: Challenging Synthetic (15k, 50D), PCA $d$=5, (EVR=0.904), train=12,000 (full), test=3000; Phase 1_2: 1/3/10%, *(ex int = exact interpolation; rel int = relaxed interpolation)*.

| Model | Train Acc | Test Acc | Train Loss | Test Loss |
|---|---|---|---|---|
| WBSNN ($\alpha_{(k,m)}$, 1%) **(ex int)** | 0.7588 | 0.7633 | 0.5192 | 0.5176 |
| WBSNN ($\alpha_{(k,m)}$, 3%) **(ex int)** | 0.7573 | 0.7597 | 0.5220 | 0.5175 |
| WBSNN ($\alpha_{(k,m)}$, 10%) **(ex int)** | 0.7576 | 0.7647 | 0.5236 | 0.5202 |
| WBSNN ($\alpha_k$, 1%) **(ex int)** | 0.7582 | 0.7603 | 0.5198 | 0.5171 |
| WBSNN ($\alpha_k$, 3%) **(ex int)** | 0.7576 | 0.7587 | 0.5207 | 0.5175 |
| WBSNN ($\alpha_k$, 10%) **(ex int)** | 0.7566 | 0.7620 | 0.5226 | 0.5172 |
| Logistic Regression | 0.7563 | 0.7610 | 0.5006 | 0.4929 |
| RF + Sigmoid Cal. | 0.8678 | 0.7497 | 0.3547 | 0.5084 |
| SVM (RBF) | 0.7614 | 0.7573 | 0.5056 | 0.5097 |
| MLP (1 hidden) | 0.7544 | 0.7570 | 0.5054 | 0.4982 |
| RFF+LogReg (D=2000) | 0.7569 | 0.7617 | 0.4994 | 0.4930 |
| ExtraTrees | 0.7848 | 0.7583 | 0.5153 | 0.5367 |
| LapLogReg ($k$=25, $\lambda$=$10^{-3}$) | 0.7563 | 0.7610 | 0.5006 | 0.4929 |
| Diffusion ($\tau$=1, $k$=25) | 0.7785 | 0.7570 | 0.4701 | 0.5064 |
| GroupDRO-Logistic | 0.7553 | 0.7603 | 0.5008 | 0.4933 |

Table 20: Challenging Synthetic (15k, 50D), PCA $d$=20, (EVR=0.965), train=12,000 (full), test=3000; Phase 1_2: 1/3/10%, *(ex int = exact interpolation; rel int = relaxed interpolation)*.

| Model | Train Acc | Test Acc | Train Loss | Test Loss |
|---|---|---|---|---|
| WBSNN ($\alpha_{(k,m)}$, 1%) **(rel int)** | 0.7894 | 0.7637 | 0.4759 | 0.5176 |
| WBSNN ($\alpha_{(k,m)}$, 3%) **(rel int)** | 0.7884 | 0.7593 | 0.4755 | 0.5189 |
| WBSNN ($\alpha_{(k,m)}$, 10%) **(rel int)** | 0.7816 | 0.7670 | 0.4892 | 0.5151 |
| WBSNN ($\alpha_k$, 1%) **(rel int)** | 0.7845 | 0.7630 | 0.4847 | 0.5112 |
| WBSNN ($\alpha_k$, 3%) **(rel int)** | 0.7873 | 0.7630 | 0.4842 | 0.5119 |
| WBSNN ($\alpha_k$, 10%) **(rel int)** | 0.7779 | 0.7670 | 0.4947 | 0.5115 |
| Logistic Regression | 0.7658 | 0.7697 | 0.4874 | 0.4810 |
| RF + Sigmoid Cal. | 0.9208 | 0.7580 | 0.2817 | 0.4924 |
| SVM (RBF) | 0.7791 | 0.7653 | 0.4779 | 0.5005 |
| MLP (1 hidden) | 0.7637 | 0.7657 | 0.4948 | 0.4890 |
| RFF+LogReg (D=2000) | 0.7662 | 0.7677 | 0.4850 | 0.4807 |
| ExtraTrees | 0.8243 | 0.7550 | 0.4787 | 0.5334 |
| LapLogReg ($k$=25, $\lambda$=$10^{-3}$) | 0.7659 | 0.7703 | 0.4874 | 0.4810 |
| Diffusion ($\tau$=1, $k$=25) | 0.7817 | 0.7607 | 0.4665 | 0.4971 |
| GroupDRO-Logistic | 0.7553 | 0.7617 | 0.5013 | 0.4940 |

**FI-2010 (LOB) setup and summary.** We use the FI-2010 limit-order-book benchmark (June 1–14, 2010; five Helsinki stocks) with 148 normalized LOB features and a chronological 80/20 split (no look-ahead). Features are standardized and PCA-compressed to $d \in \{10, 20, 40\}$ (EVR $\approx$ 0.61/0.71/0.85); Phase 1/2 discovery budgets are 1–10% with training sizes $n \in \{1k, 2k, 10k, 30k\}$. At small $n$, performance is modest but consistent with limited information: $d$=10, $n$=1k yields $\alpha_{(k,m)} \approx 0.458$ and $\alpha_k \approx 0.404 \to 0.476$, near LR/RBF-SVM/MLP (0.40–0.47). Increasing representation and data firms results: $d$=20, $n$=2k gives $\alpha_{(k,m)} \approx 0.488$ and $\alpha_k \approx 0.491$; pushing to $d$=40, $n$=2k shows a small-$n$, high-$d$ wobble ($\alpha_{(k,m)} \approx 0.446$, $\alpha_k \approx 0.463/0.392$), with MLP at $\approx 0.516$. Scaling $n$ is the main driver: at $d$=20, $n$=10k, both heads reach $\approx 0.618$–$0.620$; at $d$=20, $n$=30k, $\alpha_{(k,m)} \approx 0.651$ and $\alpha_k \approx 0.647$, while linear and generic baselines remain in the 0.44–0.59 range. A 20-seed study at $d$=10, $n$=2k, 10% yields $0.504 \pm 0.046$ ($\alpha_{(k,m)}$) and $0.500 \pm 0.051$ ($\alpha_k$), confirming higher variance in low-$n$, low-$d$ settings. Throughout, subset size is capped at $|D_k| \leq 5$, trading exact alignment for more local prototypes; this stabilizes generalization under tight budgets and explains the lack of Phase-2 residual collapse at $d$=40. Overall, WBSNN accuracy scales smoothly with information $(d, n)$ under tiny budgets, while remaining auditable via $D_k$ and alignment diagnostics.

Table 21: FI-2010 LOB: $d = 10$ (EVR=0.613), $n_{\text{train}} = 1000$.

| Model | Train Acc | Test Acc | Train Loss | Test Loss |
|---|---|---|---|---|
| WBSNN ($\alpha_{(k,m)}$, Phase1_2:5%) | 0.642500 | 0.458002 | 0.794756 | 1.253171 |
| WBSNN ($\alpha_{(k,m)}$, Phase1_2:10%) | 0.662500 | 0.458320 | 0.781225 | 1.257423 |
| WBSNN ($\alpha_k$, Phase1_2:5%) | 0.610000 | 0.404222 | 0.838781 | 1.260667 |
| WBSNN ($\alpha_k$, Phase1_2:10%) | 0.642500 | 0.476118 | 0.795929 | 1.312865 |
| Logistic Regression | 0.529000 | 0.426587 | 0.983144 | 1.077829 |
| Random Forest | 1.000000 | 0.468074 | 0.231690 | 1.052151 |
| SVM (RBF) | 0.564000 | 0.401021 | 0.907460 | 1.077469 |
| MLP (1 hidden layer) | 0.631000 | 0.469578 | 0.864141 | 1.080732 |
| Discretized DeepONet | 0.559000 | 0.411741 | 0.906888 | 1.132754 |
| NAIS-Net | 0.645000 | 0.465494 | 0.785776 | 1.112392 |
| Kernel Ridge (Nyström RBF) | 0.747000 | 0.434396 | 0.716720 | 1.171197 |
| Linformer | 0.788000 | 0.453270 | 0.522147 | 1.615529 |

Table 22: FI-2010 LOB: $d = 20$ (EVR=0.709), $n_{\text{train}} = 2000$.

| Model | Train Acc | Test Acc | Train Loss | Test Loss |
|---|---|---|---|---|
| WBSNN ($\alpha_{(k,m)}$, Phase1_2:5%) | 0.661875 | 0.487514 | 0.746566 | 1.154213 |
| WBSNN ($\alpha_{(k,m)}$, Phase1_2:10%) | 0.686250 | 0.477014 | 0.709236 | 1.158808 |
| WBSNN ($\alpha_k$, Phase1_2:5%) | 0.653750 | 0.490563 | 0.761693 | 1.138800 |
| WBSNN ($\alpha_k$, Phase1_2:10%) | 0.653750 | 0.463549 | 0.751237 | 1.145708 |
| Logistic Regression | 0.543500 | 0.450731 | 0.949991 | 1.063273 |
| Random Forest | 1.000000 | 0.478077 | 0.231360 | 1.022263 |
| SVM (RBF) | 0.590000 | 0.427194 | 0.871935 | 1.057475 |
| MLP (1 hidden layer) | 0.581500 | 0.457354 | 0.862921 | 1.055610 |
| Discretized DeepONet | 0.561500 | 0.434506 | 0.891528 | 1.093523 |
| NAIS-Net | 0.648500 | 0.469274 | 0.789911 | 1.046069 |
| Kernel Ridge (Nyström RBF) | 0.799000 | 0.447710 | 0.662295 | 1.103524 |
| Linformer | 0.677500 | 0.488052 | 0.731564 | 1.119059 |

Table 23: FI-2010 LOB: $d = 40$ (EVR=0.850), $n_{\text{train}} = 2000$.

| Model | Train Acc | Test Acc | Train Loss | Test Loss |
|---|---|---|---|---|
| WBSNN ($\alpha_{(k,m)}$, Phase1_2:7%) | 0.553125 | 0.446689 | 1.213871 | 1.301656 |
| WBSNN ($\alpha_{(k,m)}$, Phase1_2:10%) | 0.551875 | 0.445764 | 1.206899 | 1.310211 |
| WBSNN ($\alpha_k$, Phase1_2:7%) | 0.477500 | 0.463355 | 1.300635 | 1.299468 |
| WBSNN ($\alpha_k$, Phase1_2:10%) | 0.503125 | 0.392039 | 1.289421 | 1.344629 |
| Logistic Regression | 0.579500 | 0.460734 | 0.915455 | 1.058701 |
| Random Forest | 1.000000 | 0.461231 | 0.236870 | 1.031504 |
| SVM (RBF) | 0.634500 | 0.439873 | 0.814431 | 1.052494 |
| MLP (1 hidden layer) | 0.706500 | 0.516377 | 0.680691 | 1.043847 |
| Discretized DeepONet | 0.596500 | 0.464073 | 0.863832 | 1.062425 |
| NAIS-Net | 0.727000 | 0.487569 | 0.656201 | 1.089246 |
| Kernel Ridge (Nyström RBF) | 0.865500 | 0.481236 | 0.565842 | 1.065495 |
| Linformer | 0.776000 | 0.467522 | 0.561651 | 1.264335 |

Table 24: FI-2010 LOB: $d = 20$ (EVR=0.679), $n_{\text{train}} = 10000$.

| Model | Train Acc | Test Acc | Train Loss | Test Loss |
|---|---|---|---|---|
| WBSNN ($\alpha_{(k,m)}$, Phase1_2:1%) | 0.673375 | 0.617163 | 0.742650 | 0.916936 |
| WBSNN ($\alpha_{(k,m)}$, Phase1_2:5%) | 0.668875 | 0.618322 | 0.749830 | 0.923892 |
| WBSNN ($\alpha_k$, Phase1_2:1%) | 0.659375 | 0.616198 | 0.775500 | 0.938115 |
| WBSNN ($\alpha_k$, Phase1_2:5%) | 0.663875 | 0.619605 | 0.756335 | 0.919158 |
| Logistic Regression | 0.522400 | 0.443240 | 0.987157 | 1.055377 |
| Random Forest | 0.999800 | 0.514708 | 0.228968 | 0.986638 |
| SVM (RBF) | 0.575400 | 0.467977 | 0.884310 | 1.016771 |
| MLP (1 hidden layer) | 0.618500 | 0.581898 | 0.850882 | 0.949489 |
| Discretized DeepONet | 0.531200 | 0.468377 | 0.936686 | 1.031931 |
| NAIS-Net | 0.598100 | 0.534382 | 0.870932 | 0.968862 |
| Kernel Ridge (Nyström RBF) | 0.736900 | 0.498165 | 0.741503 | 1.055312 |
| Linformer | 0.590100 | 0.516046 | 0.875227 | 1.009025 |

Table 25: FI-2010 LOB: $d = 20$ (EVR=0.673), $n_{\text{train}} = 30000$.

| Model | Train Acc | Test Acc | Train Loss | Test Loss |
|---|---|---|---|---|
| WBSNN ($\alpha_{(k,m)}$, Phase1_2:1%) | 0.653958 | 0.650552 | 0.774376 | 0.856203 |
| WBSNN ($\alpha_{(k,m)}$, Phase1_2:3%) | 0.654042 | 0.643115 | 0.778076 | 0.869560 |
| WBSNN ($\alpha_k$, Phase1_2:1%) | 0.649917 | 0.647213 | 0.785796 | 0.871965 |
| WBSNN ($\alpha_k$, Phase1_2:3%) | 0.649792 | 0.635817 | 0.783198 | 0.896413 |
| Logistic Regression | 0.528867 | 0.440866 | 0.986256 | 1.059975 |
| Random Forest | 0.999867 | 0.518433 | 0.219621 | 0.981985 |
| SVM (RBF) | 0.590367 | 0.478228 | 0.881934 | 1.018261 |
| MLP (1 hidden layer) | 0.620967 | 0.587445 | 0.854811 | 0.937810 |
| Discretized DeepONet | 0.535267 | 0.464280 | 0.965054 | 1.032401 |
| NAIS-Net | 0.584700 | 0.498744 | 0.896916 | 0.999075 |
| Kernel Ridge (Nyström RBF) | 0.707233 | 0.510058 | 0.776045 | 1.033917 |
| Linformer | 0.586833 | 0.525304 | 0.885875 | 0.992044 |

Table 26: FI-2010 LOB: Error bars at $d = 10$,
Phase1_2:10% (20 seeds).

| Model | Test Acc (mean $\pm$ std) |
|---|---|
| WBSNN ($\alpha_{(k,m)}$) | $0.5046 \pm 0.0458$ |
| WBSNN ($\alpha_k$) | $0.5001 \pm 0.0508$ |

**PRSA2017 setup and summary.** We use the Beijing Multi-Site PRSA-2017 data (Mar 2013–Feb 2017) with a chronological split: the last 20% of time ($\sim 82{,}325$ hours) is the fixed test window; inputs are four meteorological features (TEMP, PRES, DEWP, WSPM), standardized, with $d=4$ (EVR$= 1.00$). Under this strict out-of-time protocol and small training windows ($n \in \{1\text{k}, 3\text{k}, 10\text{k}\}$), WBSNN's test $R^2$ scales smoothly: $\sim 0.27$ (1k, 7–25%) $\to 0.29-0.30$ (3k, 5–15%) $\to 0.32-0.33$ (10k, 1–7%), with MAE $\approx 0.58-0.60$ and low sensitivity to head budget; the ordering $\alpha_{k,m} \geq \alpha_k$(shared) $\geq \alpha_k$(separate) holds across scales. Baselines show expected behavior under drift: ExtraTrees reaches near-1.0 train $R^2$ but only $\sim 0.22-0.28$ test, LSTM can turn negative (to $\approx -0.15$), EDMD hovers near $\sim 0.25$, while MLP/GB/Transformer-MLP top out around 0.30–0.34. Proportional 80/20 ablations (e.g., 3k/600 and 10k/2k) yield noisier metrics but the same WBSNN ordering, with $\alpha_{k,m}$ at $\approx 0.25-0.26$ (3k/600) and $\approx 0.31$ (10k/2k), close to the strongest MLP (0.327). Exact vs relaxed interpolation shows a clear trade-off: exact is stronger at 1k ($\approx 0.283-0.284$ vs $0.269-0.271$), roughly tied at 3k, and relaxed wins at 10k ($\approx 0.326-0.332$ vs $0.297-0.320$), indicating that small slack improves robustness to seasonal and episodic drift as data grow.

Table 27: Beijing PRSA2017: $d = 4$ (EVR=1.000), $n_{\text{train}} = 1000$, $n_{\text{test}} = 82{,}325$. Notation: "rel int" = relaxed interpolation; "ex int" = exact interpolation; "sh" = shared backbone; "sep" = separate backbone.

| Model | Train MSE | Test MSE | Train MAE | Test MAE | Train $R^2$ | Test $R^2$ |
|---|---|---|---|---|---|---|
| WBSNN ($\alpha_{(k,m)}$, 7%, *rel int*) | 0.618570 | 0.641911 | 0.559533 | 0.583941 | 0.391600 | 0.269558 |
| WBSNN ($\alpha_{(k,m)}$, 10%, *rel int*) | 0.638253 | 0.642331 | 0.566655 | 0.581238 | 0.372241 | 0.269080 |
| WBSNN ($\alpha_{(k,m)}$, 25%, *rel int*) | 0.662111 | 0.640665 | 0.576373 | 0.582264 | 0.348775 | 0.270976 |
| WBSNN ($\alpha_k$, sh, 7%, *rel int*) | 0.624012 | 0.689124 | 0.577389 | 0.615779 | 0.386247 | 0.215833 |
| WBSNN ($\alpha_k$, sh, 10%, *rel int*) | 0.638667 | 0.663569 | 0.574222 | 0.601165 | 0.371834 | 0.244913 |
| WBSNN ($\alpha_k$, sh, 25%, *rel int*) | 0.658739 | 0.659042 | 0.585030 | 0.604381 | 0.352091 | 0.250064 |
| WBSNN ($\alpha_k$, sep, 7%, *rel int*) | 0.644322 | 0.680975 | 0.587492 | 0.618892 | 0.366272 | 0.225107 |
| WBSNN ($\alpha_k$, sep, 10%, *rel int*) | 0.666288 | 0.659224 | 0.591238 | 0.607368 | 0.344667 | 0.249858 |
| WBSNN ($\alpha_k$, sep, 25%, *rel int*) | 0.672573 | 0.657045 | 0.596266 | 0.609083 | 0.338485 | 0.252337 |
| WBSNN ($\alpha_{(k,m)}$, 7%, *ex int*) | 0.766807 | 0.630164 | 0.627476 | 0.576859 | 0.245800 | 0.282925 |
| WBSNN ($\alpha_{(k,m)}$, 10%, *ex int*) | 0.767275 | 0.628828 | 0.626281 | 0.574256 | 0.245341 | 0.284445 |
| WBSNN ($\alpha_{(k,m)}$, 25%, *ex int*) | 0.780345 | 0.632802 | 0.634366 | 0.576375 | 0.232485 | 0.279924 |
| WBSNN ($\alpha_k$, sh, 7%, *ex int*) | 0.774323 | 0.651270 | 0.653145 | 0.611013 | 0.238408 | 0.258908 |
| WBSNN ($\alpha_k$, sh, 10%, *ex int*) | 0.779560 | 0.651303 | 0.655450 | 0.610426 | 0.233257 | 0.258870 |
| WBSNN ($\alpha_k$, sh, 25%, *ex int*) | 0.786984 | 0.658663 | 0.656443 | 0.612784 | 0.225956 | 0.250496 |
| WBSNN ($\alpha_k$, sep, 7%, *ex int*) | 0.801370 | 0.662512 | 0.660155 | 0.611078 | 0.211806 | 0.246116 |
| WBSNN ($\alpha_k$, sep, 10%, *ex int*) | 0.813325 | 0.676132 | 0.665465 | 0.616010 | 0.200048 | 0.230618 |
| WBSNN ($\alpha_k$, sep, 25%, *ex int*) | 0.821104 | 0.680059 | 0.670086 | 0.617575 | 0.192396 | 0.226149 |
| Linear Regression | 0.792422 | 0.740826 | 0.640207 | 0.637274 | 0.207578 | 0.157001 |
| Gradient Boosting | 0.379893 | 0.693600 | 0.447361 | 0.602252 | 0.620107 | 0.210740 |
| MLP Baseline | 0.485877 | 0.653235 | 0.507794 | 0.566028 | 0.514123 | 0.256673 |
| Transformer MLP | 0.544369 | 0.633099 | 0.528648 | 0.555000 | 0.455631 | 0.279585 |
| TabTransformer | 0.453626 | 0.713115 | 0.487507 | 0.589505 | 0.546374 | 0.188534 |
| ExtraTreesRegressor | 0.000006 | 0.685417 | 0.000112 | 0.599094 | 0.999994 | 0.220052 |
| HistGB (CatBoost fallback) | 0.238742 | 0.744962 | 0.351445 | 0.622866 | 0.761258 | 0.152295 |
| LSTM (lag=24) | 0.878913 | 1.040005 | 0.679906 | 0.764515 | 0.105908 | -0.147716 |
| EDMD (Poly lift, lag=24, deg=2) | 0.693338 | 0.660922 | 0.615977 | 0.605961 | 0.293871 | 0.248127 |

Table 28: Beijing PRSA2017: $d = 4$ (EVR=1.000), $n_{\text{train}} = 3000$, $n_{\text{test}} = 82{,}325$. Notation: "rel int" = relaxed interpolation; "ex int" = exact interpolation; "sh" = shared backbone; "sep" = separate backbone.

| Model | Train MSE | Test MSE | Train MAE | Test MAE | Train $R^2$ | Test $R^2$ |
|---|---|---|---|---|---|---|
| WBSNN ($\alpha_{(k,m)}$, 5%, *rel int*) | 0.656149 | 0.703912 | 0.577091 | 0.597687 | 0.319430 | 0.292502 |
| WBSNN ($\alpha_{(k,m)}$, 10%, *rel int*) | 0.657450 | 0.696231 | 0.578698 | 0.594983 | 0.318081 | 0.300221 |
| WBSNN ($\alpha_{(k,m)}$, 15%, *rel int*) | 0.654483 | 0.701072 | 0.578030 | 0.597031 | 0.321158 | 0.295356 |
| WBSNN ($\alpha_k$, sh, 5%, *rel int*) | 0.666619 | 0.713751 | 0.592692 | 0.613276 | 0.308570 | 0.282612 |
| WBSNN ($\alpha_k$, sh, 10%, *rel int*) | 0.661710 | 0.713105 | 0.585848 | 0.606728 | 0.313663 | 0.283262 |
| WBSNN ($\alpha_k$, sh, 15%, *rel int*) | 0.659058 | 0.710773 | 0.586762 | 0.609358 | 0.316413 | 0.285606 |
| WBSNN ($\alpha_k$, sep, 5%, *rel int*) | 0.681605 | 0.723789 | 0.604193 | 0.623168 | 0.293027 | 0.272523 |
| WBSNN ($\alpha_k$, sep, 10%, *rel int*) | 0.670411 | 0.711685 | 0.591879 | 0.610020 | 0.304638 | 0.284689 |
| WBSNN ($\alpha_k$, sep, 15%, *rel int*) | 0.674022 | 0.718627 | 0.596054 | 0.614956 | 0.300891 | 0.277712 |
| WBSNN ($\alpha_{(k,m)}$, 5%, *ex int*) | 0.745485 | 0.702362 | 0.630674 | 0.598891 | 0.226769 | 0.294059 |
| WBSNN ($\alpha_{(k,m)}$, 10%, *ex int*) | 0.748913 | 0.704946 | 0.632946 | 0.600194 | 0.223213 | 0.291462 |
| WBSNN ($\alpha_{(k,m)}$, 15%, *ex int*) | 0.757309 | 0.713810 | 0.637084 | 0.603791 | 0.214505 | 0.282553 |
| WBSNN ($\alpha_k$, sh, 5%, *ex int*) | 0.753109 | 0.735135 | 0.651284 | 0.639383 | 0.218861 | 0.261120 |
| WBSNN ($\alpha_k$, sh, 10%, *ex int*) | 0.754657 | 0.736362 | 0.651861 | 0.639748 | 0.217255 | 0.259886 |
| WBSNN ($\alpha_k$, sh, 15%, *ex int*) | 0.762029 | 0.741674 | 0.655449 | 0.640611 | 0.209609 | 0.254547 |
| WBSNN ($\alpha_k$, sep, 5%, *ex int*) | 0.773028 | 0.750112 | 0.658795 | 0.641117 | 0.198201 | 0.246066 |
| WBSNN ($\alpha_k$, sep, 10%, *ex int*) | 0.777798 | 0.754868 | 0.661792 | 0.643159 | 0.193253 | 0.241286 |
| WBSNN ($\alpha_k$, sep, 15%, *ex int*) | 0.785708 | 0.766764 | 0.665657 | 0.647759 | 0.185049 | 0.229329 |
| Linear Regression | 0.810602 | 0.828868 | 0.649523 | 0.663098 | 0.189398 | 0.166909 |
| Gradient Boosting | 0.542878 | 0.705065 | 0.525823 | 0.601151 | 0.457122 | 0.291343 |
| MLP Baseline | 0.590280 | 0.692952 | 0.561343 | 0.599555 | 0.409720 | 0.303517 |
| Transformer MLP | 0.618047 | 0.685839 | 0.548472 | 0.578823 | 0.381953 | 0.310667 |
| TabTransformer | 0.573379 | 0.698697 | 0.529078 | 0.576363 | 0.426621 | 0.297743 |
| ExtraTreesRegressor | 0.000083 | 0.721530 | 0.000813 | 0.604455 | 0.999917 | 0.274794 |
| HistGB (CatBoost fallback) | 0.319057 | 0.761739 | 0.406342 | 0.618528 | 0.680943 | 0.234379 |
| LSTM (lag=24) | 0.907224 | 1.098229 | 0.704622 | 0.725680 | 0.085741 | -0.097355 |
| EDMD (Poly lift, lag=24, deg=2) | 0.890765 | 0.890046 | 0.694420 | 0.663599 | 0.102068 | 0.107608 |

Table 29: Beijing PRSA2017: $d = 4$ (EVR=1.000), $n_{\text{train}} = 10000$, $n_{\text{test}} = 82{,}325$. Notation: "rel int" = relaxed interpolation; "ex int" = exact interpolation; "sh" = shared backbone; "sep" = separate backbone.

| Model | Train MSE | Test MSE | Train MAE | Test MAE | Train $R^2$ | Test $R^2$ |
|---|---|---|---|---|---|---|
| WBSNN ($\alpha_{(k,m)}$, 1%, *rel int*) | 0.695077 | 0.685453 | 0.589800 | 0.595902 | 0.303611 | 0.329147 |
| WBSNN ($\alpha_{(k,m)}$, 3%, *rel int*) | 0.685843 | 0.682126 | 0.583978 | 0.592003 | 0.312863 | 0.332403 |
| WBSNN ($\alpha_{(k,m)}$, 7%, *rel int*) | 0.697026 | 0.688120 | 0.589651 | 0.596024 | 0.301658 | 0.326537 |
| WBSNN ($\alpha_k$, sh, 1%, *rel int*) | 0.706775 | 0.701345 | 0.601163 | 0.610782 | 0.291890 | 0.313594 |
| WBSNN ($\alpha_k$, sh, 3%, *rel int*) | 0.696073 | 0.691676 | 0.596378 | 0.605414 | 0.302612 | 0.323057 |
| WBSNN ($\alpha_k$, sh, 7%, *rel int*) | 0.699206 | 0.698812 | 0.597525 | 0.607717 | 0.299474 | 0.316072 |
| WBSNN ($\alpha_k$, sep, 1%, *rel int*) | 0.714830 | 0.709229 | 0.609328 | 0.618346 | 0.283821 | 0.305878 |
| WBSNN ($\alpha_k$, sep, 3%, *rel int*) | 0.704452 | 0.699359 | 0.601241 | 0.611087 | 0.294218 | 0.315537 |
| WBSNN ($\alpha_k$, sep, 7%, *rel int*) | 0.712420 | 0.706685 | 0.604686 | 0.612484 | 0.286235 | 0.308368 |
| WBSNN ($\alpha_{(k,m)}$, 1%, *ex int*) | 0.747853 | 0.695087 | 0.621322 | 0.598003 | 0.250735 | 0.319719 |
| WBSNN ($\alpha_{(k,m)}$, 3%, *ex int*) | 0.761869 | 0.708098 | 0.629487 | 0.605235 | 0.236692 | 0.306985 |
| WBSNN ($\alpha_{(k,m)}$, 7%, *ex int*) | 0.775455 | 0.718634 | 0.634631 | 0.607564 | 0.223081 | 0.296673 |
| WBSNN ($\alpha_k$, sh, 1%, *ex int*) | 0.772541 | 0.733919 | 0.652491 | 0.641724 | 0.226000 | 0.281714 |
| WBSNN ($\alpha_k$, sh, 3%, *ex int*) | 0.780478 | 0.741464 | 0.654748 | 0.643332 | 0.218048 | 0.274329 |
| WBSNN ($\alpha_k$, sh, 7%, *ex int*) | 0.785426 | 0.745418 | 0.655936 | 0.642755 | 0.213091 | 0.270460 |
| WBSNN ($\alpha_k$, sep, 1%, *ex int*) | 0.785592 | 0.741192 | 0.657928 | 0.642786 | 0.212925 | 0.274595 |
| WBSNN ($\alpha_k$, sep, 3%, *ex int*) | 0.792648 | 0.749911 | 0.660078 | 0.644612 | 0.205855 | 0.266062 |
| WBSNN ($\alpha_k$, sep, 7%, *ex int*) | 0.802719 | 0.762370 | 0.664623 | 0.648652 | 0.195765 | 0.253868 |
| Linear Regression | 0.825971 | 0.848414 | 0.654933 | 0.667590 | 0.174029 | 0.169657 |
| Gradient Boosting | 0.640431 | 0.694948 | 0.567352 | 0.600901 | 0.359569 | 0.319855 |
| MLP Baseline | 0.642532 | 0.677827 | 0.579014 | 0.597316 | 0.357468 | 0.336610 |
| Transformer MLP | 0.653198 | 0.687728 | 0.564441 | 0.580408 | 0.346802 | 0.326921 |
| TabTransformer | 0.615521 | 0.700102 | 0.554664 | 0.589814 | 0.384479 | 0.314810 |
| ExtraTreesRegressor | 0.001810 | 0.730956 | 0.004799 | 0.610842 | 0.998190 | 0.284613 |
| HistGB (CatBoost fallback) | 0.466135 | 0.722155 | 0.482938 | 0.606517 | 0.533865 | 0.293227 |
| LSTM (lag=24) | 0.976328 | 0.969949 | 0.717147 | 0.694659 | 0.022372 | 0.050946 |
| EDMD (Poly lift, lag=24, deg=2) | 0.957590 | 1.021665 | 0.719162 | 0.688723 | 0.039986 | 0.008594 |

Table 30: Beijing PRSA2017 (80/20 ablation): $d = 4$ (EVR=1.000), $n_{\text{train}} = 3000$, $n_{\text{test}} = 600$. Notation: "rel int" = relaxed interpolation; "ex int" = exact interpolation; "sh" = shared backbone; "sep" = separate backbone.

| Model | Train MSE | Test MSE | Train MAE | Test MAE | Train $R^2$ | Test $R^2$ |
|---|---|---|---|---|---|---|
| WBSNN ($\alpha_{(k,m)}$, 5%, *rel int*) | 0.645593 | 0.667178 | 0.569555 | 0.582276 | 0.330379 | 0.258177 |
| WBSNN ($\alpha_{(k,m)}$, 10%, *rel int*) | 0.650208 | 0.671631 | 0.572116 | 0.586320 | 0.325592 | 0.253226 |
| WBSNN ($\alpha_{(k,m)}$, 15%, *rel int*) | 0.662848 | 0.672192 | 0.577766 | 0.589588 | 0.312482 | 0.252602 |
| WBSNN ($\alpha_k$, sh, 5%, *rel int*) | 0.654200 | 0.684182 | 0.584280 | 0.602350 | 0.321452 | 0.239271 |
| WBSNN ($\alpha_k$, sh, 10%, *rel int*) | 0.651857 | 0.676491 | 0.581193 | 0.596277 | 0.323882 | 0.247822 |
| WBSNN ($\alpha_k$, sh, 15%, *rel int*) | 0.662495 | 0.670283 | 0.586935 | 0.594059 | 0.312848 | 0.254725 |
| WBSNN ($\alpha_k$, sep, 5%, *rel int*) | 0.664068 | 0.684722 | 0.590508 | 0.604292 | 0.311217 | 0.238670 |
| WBSNN ($\alpha_k$, sep, 10%, *rel int*) | 0.664752 | 0.687558 | 0.590080 | 0.603413 | 0.310507 | 0.235517 |
| WBSNN ($\alpha_k$, sep, 15%, *rel int*) | 0.676833 | 0.685301 | 0.596375 | 0.604291 | 0.297976 | 0.238027 |
| WBSNN ($\alpha_{(k,m)}$, 5%, *ex int*) | 0.729119 | 0.650090 | 0.618568 | 0.576760 | 0.243744 | 0.277177 |
| WBSNN ($\alpha_{(k,m)}$, 10%, *ex int*) | 0.749758 | 0.655918 | 0.632943 | 0.582051 | 0.222337 | 0.270697 |
| WBSNN ($\alpha_{(k,m)}$, 15%, *ex int*) | 0.757270 | 0.659168 | 0.637463 | 0.584623 | 0.214545 | 0.267083 |
| WBSNN ($\alpha_k$, sh, 5%, *ex int*) | 0.748659 | 0.681922 | 0.647053 | 0.613969 | 0.223477 | 0.241784 |
| WBSNN ($\alpha_k$, sh, 10%, *ex int*) | 0.756328 | 0.697906 | 0.652550 | 0.624691 | 0.215523 | 0.224012 |
| WBSNN ($\alpha_k$, sh, 15%, *ex int*) | 0.761052 | 0.700683 | 0.654605 | 0.625023 | 0.210623 | 0.220924 |
| WBSNN ($\alpha_k$, sep, 5%, *ex int*) | 0.762630 | 0.696238 | 0.653581 | 0.620080 | 0.208986 | 0.225866 |
| WBSNN ($\alpha_k$, sep, 10%, *ex int*) | 0.774878 | 0.709209 | 0.660706 | 0.626774 | 0.196282 | 0.211444 |
| WBSNN ($\alpha_k$, sep, 15%, *ex int*) | 0.787496 | 0.716508 | 0.666144 | 0.628306 | 0.183194 | 0.203328 |
| Linear Regression | 0.810602 | 0.801751 | 0.649523 | 0.656842 | 0.189398 | 0.108548 |
| Gradient Boosting | 0.542878 | 0.654196 | 0.525823 | 0.588998 | 0.457122 | 0.272612 |
| MLP Baseline | 0.598596 | 0.636876 | 0.558970 | 0.585497 | 0.401404 | 0.291870 |
| Transformer MLP | 0.627384 | 0.600555 | 0.545739 | 0.546079 | 0.372616 | 0.332254 |
| TabTransformer | 0.572121 | 0.688862 | 0.536453 | 0.598654 | 0.427879 | 0.234067 |
| ExtraTreesRegressor | 0.000083 | 0.677715 | 0.000813 | 0.600068 | 0.999917 | 0.246461 |
| HistGB (CatBoost fallback) | 0.319057 | 0.730615 | 0.406342 | 0.612856 | 0.680943 | 0.187643 |
| LSTM (lag=24) | 0.951910 | 0.951943 | 0.719389 | 0.710372 | 0.040771 | -0.043745 |
| EDMD (Poly lift, lag=24, deg=2) | 0.890765 | 0.972033 | 0.694420 | 0.704489 | 0.102068 | -0.048958 |

Table 31: Beijing PRSA2017 (80/20 ablation): $d = 4$ (EVR=1.000), $n_{\text{train}} = 10000$, $n_{\text{test}} = 2000$. Notation: "rel int" = relaxed interpolation; "ex int" = exact interpolation; "sh" = shared backbone; "sep" = separate backbone.

| Model | Train MSE | Test MSE | Train MAE | Test MAE | Train $R^2$ | Test $R^2$ |
|---|---|---|---|---|---|---|
| WBSNN ($\alpha_{(k,m)}$, 1%, *rel int*) | 0.688236 | 0.655980 | 0.584978 | 0.596126 | 0.310464 | 0.310791 |
| WBSNN ($\alpha_{(k,m)}$, 3%, *rel int*) | 0.692875 | 0.656443 | 0.586684 | 0.597915 | 0.305816 | 0.310304 |
| WBSNN ($\alpha_{(k,m)}$, 7%, *rel int*) | 0.694702 | 0.660850 | 0.590564 | 0.602104 | 0.303987 | 0.305674 |
| WBSNN ($\alpha_k$, sh, 1%, *rel int*) | 0.701060 | 0.676044 | 0.602139 | 0.618658 | 0.297617 | 0.289710 |
| WBSNN ($\alpha_k$, sh, 3%, *rel int*) | 0.693652 | 0.661718 | 0.592816 | 0.606239 | 0.305039 | 0.304762 |
| WBSNN ($\alpha_k$, sh, 7%, *rel int*) | 0.705541 | 0.677173 | 0.603850 | 0.618590 | 0.293127 | 0.288524 |
| WBSNN ($\alpha_k$, sep, 1%, *rel int*) | 0.707769 | 0.675720 | 0.606517 | 0.618128 | 0.290894 | 0.290050 |
| WBSNN ($\alpha_k$, sep, 3%, *rel int*) | 0.705138 | 0.671601 | 0.597960 | 0.611218 | 0.293531 | 0.294378 |
| WBSNN ($\alpha_k$, sep, 7%, *rel int*) | 0.715818 | 0.686460 | 0.609633 | 0.622808 | 0.282831 | 0.278766 |
| WBSNN ($\alpha_{(k,m)}$, 1%, *ex int*) | 0.750836 | 0.671190 | 0.622406 | 0.605548 | 0.247746 | 0.294810 |
| WBSNN ($\alpha_{(k,m)}$, 3%, *ex int*) | 0.764128 | 0.679701 | 0.629616 | 0.609747 | 0.234429 | 0.285868 |
| WBSNN ($\alpha_{(k,m)}$, 7%, *ex int*) | 0.776267 | 0.688466 | 0.635503 | 0.613222 | 0.222267 | 0.276658 |
| WBSNN ($\alpha_k$, sh, 1%, *ex int*) | 0.775078 | 0.705512 | 0.653073 | 0.644694 | 0.223459 | 0.258749 |
| WBSNN ($\alpha_k$, sh, 3%, *ex int*) | 0.780285 | 0.709379 | 0.655377 | 0.646636 | 0.218242 | 0.254687 |
| WBSNN ($\alpha_k$, sh, 7%, *ex int*) | 0.784600 | 0.711542 | 0.655817 | 0.645119 | 0.213919 | 0.252414 |
| WBSNN ($\alpha_k$, sep, 1%, *ex int*) | 0.787757 | 0.710749 | 0.658015 | 0.644609 | 0.210756 | 0.253247 |
| WBSNN ($\alpha_k$, sep, 3%, *ex int*) | 0.792103 | 0.715195 | 0.659679 | 0.645810 | 0.206402 | 0.248576 |
| WBSNN ($\alpha_k$, sep, 7%, *ex int*) | 0.805869 | 0.726732 | 0.665550 | 0.648075 | 0.192610 | 0.236454 |
| Linear Regression | 0.825971 | 0.800074 | 0.654933 | 0.663877 | 0.174029 | 0.159397 |
| Gradient Boosting | 0.640431 | 0.667842 | 0.567352 | 0.604143 | 0.359569 | 0.298328 |
| MLP Baseline | 0.640004 | 0.658328 | 0.566900 | 0.583914 | 0.361744 | 0.327477 |
| Transformer MLP | 0.641096 | 0.648514 | 0.561987 | 0.589036 | 0.358904 | 0.318635 |
| TabTransformer | 0.619942 | 0.664196 | 0.559440 | 0.592730 | 0.380058 | 0.302158 |
| ExtraTreesRegressor | 0.001810 | 0.713591 | 0.004799 | 0.626595 | 0.998190 | 0.250261 |
| HistGB (CatBoost fallback) | 0.466135 | 0.707566 | 0.482938 | 0.620893 | 0.533865 | 0.256591 |
| LSTM (lag=24) | 0.945213 | 0.984680 | 0.720387 | 0.738673 | 0.052946 | -0.030103 |
| EDMD (Poly lift, lag=24, deg=2) | 0.957590 | 0.966950 | 0.719162 | 0.721628 | 0.039986 | -0.007857 |

**ISOLET setup and summary.** The Isolet dataset maps spoken letters A–Z to 26 classes (OpenML v1): 6,238 train / 1,558 test. We standardize features, apply PCA to $d \in \{5, 10, 20, 35\}$ (components z-scored), and evaluate budgets of 1–15% under three regimes: subsets ($d$=5, $n$=2000; $d$=10, $n$=4000) and full-data sweeps ($d$=5, 20; plus $d$=35, EVR≈ 0.76) with small budgets (1–5%). A 20-seed study at $d$=5, $n$=2000, 10% reports the test-accuracy distribution for both heads. Across settings, $\alpha_{k,m}$ leads; $\alpha_k$ with a shared trunk is consistently stronger and stabler than separate backbones, especially at low $d$/low budget. As $d$ and $n$ grow, WBSNN closes on strong baselines, while Random Forest/k-NN show classic train=100% vs lower test, indicating overfit.

Table 32: ISOLET: PCA $d = 5$, (EVR=0.420), $n_{\text{train}} = 2000$, $n_{\text{test}} = 400$. Notation: "sh" = shared backbone; "sep" = separate backbone.

| Model | Train Acc | Test Acc | Train Loss | Test Loss |
|---|---|---|---|---|
| WBSNN ($\alpha_{(k,m)}$, 5%) | 0.602500 | 0.620000 | 1.031726 | 1.000247 |
| WBSNN ($\alpha_{(k,m)}$, 10%) | 0.638125 | 0.635000 | 0.933222 | 0.935203 |
| WBSNN ($\alpha_{(k,m)}$, 15%) | 0.652500 | 0.647500 | 0.905966 | 0.910387 |
| WBSNN ($\alpha_k$, sh, 5%) | 0.487500 | 0.495000 | 1.685710 | 1.700538 |
| WBSNN ($\alpha_k$, sh, 10%) | 0.578125 | 0.575000 | 1.147624 | 1.121298 |
| WBSNN ($\alpha_k$, sh, 15%) | 0.621875 | 0.637500 | 0.979887 | 0.967730 |
| WBSNN ($\alpha_k$, sep, 5%) | 0.450000 | 0.457500 | 1.846356 | 1.868448 |
| WBSNN ($\alpha_k$, sep, 10%) | 0.545000 | 0.565000 | 1.290820 | 1.256991 |
| WBSNN ($\alpha_k$, sep, 15%) | 0.594375 | 0.590000 | 1.092682 | 1.058226 |
| Logistic Regression | 0.626000 | 0.625000 | 1.064762 | 1.023552 |
| Random Forest | 1.000000 | 0.632500 | 0.235798 | 1.080969 |
| SVM (RBF) | 0.680000 | 0.670000 | 0.871917 | 0.911077 |
| MLP (1 hidden layer) | 0.634000 | 0.642500 | 0.982878 | 0.939872 |
| Kernel Ridge (Nyström RBF) | 0.566000 | 0.557500 | 2.818404 | 2.839033 |
| Label Propagation (RBF) | 0.667500 | 0.617500 | 0.797632 | 1.524892 |
| k-NN (k=15, dist) | 1.000000 | 0.642500 | 0.000000 | 1.568800 |

Table 33: ISOLET: PCA $d = 10$ (EVR=0.545), $n_{\text{train}} = 4000$, $n_{\text{test}} = 800$. Notation: "sh" = shared backbone; "sep" = separate backbone.

| Model | Train Acc | Test Acc | Train Loss | Test Loss |
|---|---|---|---|---|
| WBSNN ($\alpha_{(k,m)}$, 5%) | 0.821562 | 0.771250 | 0.472241 | 0.660406 |
| WBSNN ($\alpha_{(k,m)}$, 10%) | 0.841875 | 0.753750 | 0.422335 | 0.673265 |
| WBSNN ($\alpha_{(k,m)}$, 15%) | 0.854062 | 0.763750 | 0.388083 | 0.651753 |
| WBSNN ($\alpha_k$, sh, 5%) | 0.694063 | 0.641250 | 0.954818 | 1.122597 |
| WBSNN ($\alpha_k$, sh, 10%) | 0.801875 | 0.742500 | 0.555484 | 0.731559 |
| WBSNN ($\alpha_k$, sh, 15%) | 0.828438 | 0.752500 | 0.447987 | 0.673833 |
| WBSNN ($\alpha_k$, sep, 5%) | 0.649375 | 0.593750 | 1.101077 | 1.254728 |
| WBSNN ($\alpha_k$, sep, 10%) | 0.760938 | 0.711250 | 0.681077 | 0.822034 |
| WBSNN ($\alpha_k$, sep, 15%) | 0.793750 | 0.748750 | 0.556685 | 0.732286 |
| Logistic Regression | 0.772000 | 0.752500 | 0.663921 | 0.737691 |
| Random Forest | 1.000000 | 0.701250 | 0.204508 | 0.895178 |
| SVM (RBF) | 0.849250 | 0.776250 | 0.452041 | 0.635348 |
| MLP (1 hidden layer) | 0.819750 | 0.762500 | 0.495664 | 0.632494 |
| Kernel Ridge (Nyström RBF) | 0.738000 | 0.723750 | 2.527632 | 2.560071 |
| Label Propagation (RBF) | 0.775250 | 0.711250 | 0.584864 | 1.139011 |
| k-NN (k=15, dist) | 1.000000 | 0.730000 | 0.000000 | 1.055602 |

Table 34: ISOLET: PCA $d = 5$ (EVR=0.420), full data $n_{\text{train}} = 6238$, $n_{\text{test}} = 1559$. Notation: "sh" = shared backbone; "sep" = separate backbone.

| Model | Train Acc | Test Acc | Train Loss | Test Loss |
|---|---|---|---|---|
| WBSNN ($\alpha_{(k,m)}$, 1%) | 0.595591 | 0.592046 | 1.165179 | 1.147161 |
| WBSNN ($\alpha_{(k,m)}$, 3%) | 0.644088 | 0.643361 | 0.924612 | 0.915943 |
| WBSNN ($\alpha_{(k,m)}$, 10%) | 0.653106 | 0.652983 | 0.892467 | 0.901375 |
| WBSNN ($\alpha_k$, sh, 1%) | 0.440080 | 0.456062 | 1.879926 | 1.884090 |
| WBSNN ($\alpha_k$, sh, 3%) | 0.621844 | 0.617704 | 1.049085 | 1.032445 |
| WBSNN ($\alpha_k$, sh, 10%) | 0.644489 | 0.636947 | 0.922908 | 0.918281 |
| WBSNN ($\alpha_k$, sep, 1%) | 0.413427 | 0.430404 | 2.007891 | 2.011616 |
| WBSNN ($\alpha_k$, sep, 3%) | 0.608216 | 0.617062 | 1.109980 | 1.086602 |
| WBSNN ($\alpha_k$, sep, 10%) | 0.636673 | 0.633098 | 0.946605 | 0.931952 |
| Logistic Regression | 0.632575 | 0.637588 | 1.003948 | 0.971847 |
| Random Forest | 1.000000 | 0.646568 | 0.217297 | 1.055366 |
| SVM (RBF) | 0.678102 | 0.664529 | 0.827640 | 0.857115 |
| MLP (1 hidden layer) | 0.678903 | 0.655548 | 0.793894 | 0.847480 |
| Kernel Ridge (Nyström RBF) | 0.565085 | 0.565747 | 2.773107 | 2.793578 |
| Label Propagation (RBF) | 0.693011 | 0.627967 | 0.722494 | 1.619359 |
| k-NN (k=15, dist) | 1.000000 | 0.637588 | 0.000000 | 1.574948 |

Table 35: ISOLET: PCA $d = 20$ (EVR=0.670), full data $n_{\text{train}} = 6238$, $n_{\text{test}} = 1559$. Notation: "sh" = shared backbone; "sep" = separate backbone.

| Model | Train Acc | Test Acc | Train Loss | Test Loss |
|---|---|---|---|---|
| WBSNN ($\alpha_{(k,m)}$, 1%) | 0.823246 | 0.796665 | 0.551880 | 0.645463 |
| WBSNN ($\alpha_{(k,m)}$, 3%) | 0.934269 | 0.887749 | 0.187796 | 0.344252 |
| WBSNN ($\alpha_{(k,m)}$, 10%) | 0.960321 | 0.899936 | 0.119269 | 0.324099 |
| WBSNN ($\alpha_k$, sh, 1%) | 0.441082 | 0.436818 | 2.097107 | 2.147292 |
| WBSNN ($\alpha_k$, sh, 3%) | 0.689980 | 0.658114 | 1.134449 | 1.252120 |
| WBSNN ($\alpha_k$, sh, 10%) | 0.905411 | 0.881334 | 0.276886 | 0.403693 |
| WBSNN ($\alpha_k$, sep, 1%) | 0.356713 | 0.347659 | 2.311550 | 2.342138 |
| WBSNN ($\alpha_k$, sep, 3%) | 0.651303 | 0.642078 | 1.254789 | 1.356887 |
| WBSNN ($\alpha_k$, sep, 10%) | 0.870942 | 0.842848 | 0.393063 | 0.484338 |
| Logistic Regression | 0.908945 | 0.892239 | 0.303193 | 0.323313 |
| Random Forest | 1.000000 | 0.868505 | 0.183865 | 0.738272 |
| SVM (RBF) | 0.948862 | 0.909557 | 0.174038 | 0.290399 |
| MLP (1 hidden layer) | 0.928182 | 0.898012 | 0.228892 | 0.286440 |
| Kernel Ridge (Nyström RBF) | 0.897884 | 0.885183 | 2.156269 | 2.204649 |
| Label Propagation (RBF) | 0.861815 | 0.842207 | 0.413167 | 0.713350 |
| k-NN (k=15, dist) | 1.000000 | 0.855035 | 0.000000 | 0.725293 |

Table 36: ISOLET: PCA $d = 35$ (EVR=0.756), full data $n_\text{train} = 6238$, $n_\text{test} = 1559$. Notation: "sh" = shared backbone; "sep" = separate backbone.

| Model | Train Acc | Test Acc | Train Loss | Test Loss |
|---|---|---|---|---|
| WBSNN ($\alpha_{(k,m)}$, 3%) | 0.976353 | 0.919179 | 0.077702 | 0.340435 |
| WBSNN ($\alpha_{(k,m)}$, 5%) | 0.988978 | 0.924310 | 0.042586 | 0.352441 |
| WBSNN ($\alpha_k$, sh, 3%) | 0.660321 | 0.622194 | 1.343763 | 1.496550 |
| WBSNN ($\alpha_k$, sh, 5%) | 0.787575 | 0.740218 | 0.803531 | 1.016032 |
| WBSNN ($\alpha_k$, sep, 3%) | 0.580561 | 0.551636 | 1.574497 | 1.679204 |
| WBSNN ($\alpha_k$, sep, 5%) | 0.733467 | 0.707505 | 0.974586 | 1.153906 |
| Logistic Regression | 0.958480 | 0.934573 | 0.149727 | 0.223072 |
| Random Forest | 1.000000 | 0.907633 | 0.193623 | 0.774041 |
| SVM (RBF) | 0.983168 | 0.952534 | 0.077960 | 0.201624 |
| MLP (1 hidden layer) | 0.959122 | 0.932008 | 0.144147 | 0.211379 |
| Kernel Ridge (Nyström RBF) | 0.959763 | 0.932649 | 1.923177 | 1.987090 |
| Label Propagation (RBF) | 0.910388 | 0.878768 | 0.334925 | 0.551099 |
| k-NN (k=15, dist) | 1.000000 | 0.899936 | 0.000000 | 0.597966 |

Table 37: ISOLET: Error bars PCA5, 20 seeds,
2k train/400 test, "sh/sep" = shared/separate backbone.

| Model | Test Acc (mean $\pm$ std) |
|---|---|
| WBSNN ($\alpha_{(k,m)}$, 10%) | $0.6201 \pm 0.0108$ |
| WBSNN ($\alpha_k$, **sh**, 10%) | $0.5857 \pm 0.01918$ |
| WBSNN ($\alpha_k$, **sep**, 10%) | $0.5531 \pm 0.0217$ |

**Swiss Roll + RFF: setup and summary ($\alpha_k$ = shared-backbone).** We map $(x, y, z)$ through Random Fourier Features (Gaussian bandwidth $\sigma$=5.0, fixed seed) using $\cos/\sin$ pairs, then compress with PCA to $d \in \{10, 15\}$; features are standardized. Each variant uses a fixed-seed 80/20 split; from the 80% pool we draw $M_\text{train}$ and set $M_\text{test}$=0.2 $M_\text{train}$; WBSNN's Phase 1/2 discovery sees only 7–20% of $M_\text{train}$ (classification uses one-hot targets in Phase 2; regression targets are standardized). *Noisy 5-class:* small run (RFF20→PCA10, 800, 15%) yields $\alpha_{(k,m)} \approx 0.95$, $\alpha_k \approx 0.938$; large run (30→15, 30k, 7%) lands both heads at ~0.96, on par with RBF-SVM/MLP. *Low-sample + 20% label noise:* with 400–1k train, $\alpha_k$ often *edges* $\alpha_{(k,m)}$ at higher $d/n$ (e.g., ~0.75 @ RFF20→PCA15, 1k, 10%), suggesting the shared head's bias helps under uniform flips. *Multi-roll (4 spirals):* 2k/10% → ~0.94–0.95; 20k/7% → ~0.97, matching strong kernel/MLP baselines with tiny anchor budgets. *Regression (heteroskedastic):* at 1k, $R^2 \approx 0.78$; at 20k, $\alpha_k \approx 0.83$ vs. $\alpha_{(k,m)} \approx 0.78$, indicating better bias–variance from the shared head for 1-D targets. **Takeaway:** widening the RFF→PCA lens and increasing $n$ raise accuracy/$R^2$ under small budgets; $\alpha_{(k,m)}$ leads on complex classification, while $\alpha_k$ is sturdier under label flips and in regression.

Table 38: Swiss Roll (RFF) — noisy_5class. Embedding: RFF 20 → PCA 10. $n_{\text{train}} = 800$. WBSNN percentage refers to Phase 1&2 budget.

| Model | Train Acc | Test Acc | Train Loss | Test Loss |
|---|---|---|---|---|
| WBSNN ($\alpha_{(k,m)}$, 15%) | 0.970313 | 0.95000 | 0.085 | 0.173 |
| WBSNN ($\alpha_k$, 15%) | 0.957812 | 0.93750 | 0.112 | 0.178 |
| Logistic Regression | 0.925000 | 0.90000 | 0.267 | 0.268 |
| Random Forest | 1.000000 | 0.93125 | 0.075 | 0.213 |
| SVM (RBF) | 0.961250 | 0.95000 | 0.121 | 0.148 |
| MLP (1 hidden layer) | 0.892500 | 0.87500 | 0.572 | 0.574 |
| k-NN (k=15, dist) | 1.000000 | 0.95000 | 1e-15 | 0.138 |
| NAIS-Net | 1.000000 | 0.91875 | 0.002 | 0.645 |
| KRR | 0.933750 | 0.93125 | 0.085 | 0.173 |

Table 39: Swiss Roll (RFF) — noisy_5class. Embedding: RFF 30 → PCA 15. $n_{\text{train}} = 30000$.

| Model | Train Acc | Test Acc | Train Loss | Test Loss |
|---|---|---|---|---|
| WBSNN ($\alpha_{(k,m)}$, 7%) | 0.957917 | 0.96100 | 0.106 | 0.104 |
| WBSNN ($\alpha_k$, 7%) | 0.958000 | 0.95983 | 0.107 | 0.103 |
| Logistic Regression | 0.905867 | 0.90867 | 0.316 | 0.298 |
| Random Forest | 1.000000 | 0.95700 | 0.038 | 0.132 |
| SVM (RBF) [20k cap] | 0.959810 | 0.96150 | 0.119 | 0.115 |
| MLP (1 hidden layer) | 0.959233 | 0.96283 | 0.099 | 0.093 |
| k-NN (k=15, dist) | 1.000000 | 0.95917 | 1e-15 | 0.173 |
| NAIS-Net | 0.963333 | 0.96033 | 0.085 | 0.097 |
| KRR | 0.958267 | 0.96117 | 0.106 | 0.104 |

Table 40: Swiss Roll (RFF) — low_sample_label_noise (10-class, 20% label noise). Embedding: RFF 20 → PCA 10. $n_{\text{train}} = 400$.

| Model | Train Acc | Test Acc | Train Loss | Test Loss |
|---|---|---|---|---|
| WBSNN ($\alpha_{(k,m)}$, 20%) | 0.80625 | 0.6875 | 0.648 | 1.463 |
| WBSNN ($\alpha_k$, 20%) | 0.75938 | 0.6750 | 0.956 | 1.363 |
| Logistic Regression | 0.73750 | 0.7000 | 1.042 | 1.448 |
| Random Forest | 0.98250 | 0.6750 | 0.480 | 1.256 |
| SVM (RBF) | 0.78750 | 0.7125 | 0.878 | 1.230 |
| MLP (1 hidden layer) | 0.71250 | 0.7125 | 1.151 | 1.329 |
| k-NN (k=15, dist) | 1.00000 | 0.7000 | 1e-15 | 6.12 |
| Label Propagation (RBF) | 0.76000 | 0.7000 | 0.736 | 4.044 |
| KRR | 0.76000 | 0.7250 | 0.648 | 1.463 |

Table 41: Swiss Roll (RFF) — low_sample_label_noise (10-class, 20% label noise). Embedding: RFF 20 → PCA 10. $n_{\text{train}} = 800$.

| Model | Train Acc | Test Acc | Train Loss | Test Loss |
|---|---|---|---|---|
| WBSNN ($\alpha_{(k,m)}$, 15%) | 0.80625 | 0.73125 | 0.682 | 1.329 |
| WBSNN ($\alpha_k$, 15%) | 0.78750 | 0.74375 | 0.792 | 1.320 |
| Logistic Regression | 0.77375 | 0.73125 | 1.036 | 1.229 |
| Random Forest | 0.99500 | 0.70625 | 0.350 | 1.146 |
| SVM (RBF) | 0.80375 | 0.73125 | 0.841 | 1.124 |
| MLP (1 hidden layer) | 0.75875 | 0.68125 | 1.059 | 1.189 |
| k-NN (k=15, dist) | 1.00000 | 0.71875 | 1e-15 | 5.44 |
| Label Propagation (RBF) | 0.79625 | 0.68750 | 0.616 | 3.656 |
| KRR | 0.77500 | 0.72500 | 0.682 | 1.329 |

Table 42: Swiss Roll (RFF) — low_sample_label_noise (10-class, 20% label noise). Embedding: RFF 20 → PCA 15. $n_{\text{train}} = 1000$.

| Model | Train Acc | Test Acc | Train Loss | Test Loss |
|---|---|---|---|---|
| WBSNN ($\alpha_{(k,m)}$, 10%) | 0.824 | 0.735 | 0.589 | 1.368 |
| WBSNN ($\alpha_k$, 10%) | 0.810 | 0.750 | 0.772 | 1.301 |
| Logistic Regression | 0.802 | 0.720 | 0.911 | 1.293 |
| Random Forest | 0.988 | 0.730 | 0.348 | 1.131 |
| SVM (RBF) | 0.811 | 0.745 | 0.809 | 1.157 |
| MLP (1 hidden layer) | 0.777 | 0.715 | 1.140 | 1.304 |
| k-NN (k=15, dist) | 1.000 | 0.740 | 1e-15 | 6.59 |
| Label Propagation (RBF) | 0.808 | 0.730 | 0.588 | 4.182 |
| KRR | 0.811 | 0.735 | 0.589 | 1.368 |

Table 43: Swiss Roll (RFF) — multi_roll (10-class, 4-roll). Embedding: RFF 20 → PCA 10. $n_\text{train} = 2000$. Phase 1–2 = 10%.

| Model | Train Acc | Test Acc | Train Loss | Test Loss |
|---|---|---|---|---|
| WBSNN ($\alpha_{(k,m)}$) | 0.977500 | 0.9450 | 0.078 | 0.150 |
| WBSNN ($\alpha_k$) | 0.958125 | 0.9400 | 0.144 | 0.189 |
| Logistic Regression | 0.710000 | 0.7075 | 0.785 | 0.730 |
| Random Forest | 1.000000 | 0.9475 | 0.083 | 0.208 |
| SVM (RBF) | 0.955000 | 0.9525 | 0.125 | 0.123 |
| MLP (1 hidden layer) | 0.922500 | 0.9250 | 0.302 | 0.282 |
| k-NN (k=15, dist) | 1.000000 | 0.9575 | 1e-15 | 0.198 |
| NAIS-Net | 0.999500 | 0.9475 | 0.004 | 0.217 |
| Diffusion Maps + LogReg | 0.710500 | 0.7025 | 0.785 | 0.730 |
| KRR | 0.851500 | 0.8725 | 0.078 | 0.150 |

Table 44: Swiss Roll (RFF) — multi_roll (10-class, 4-roll). Embedding: RFF 30 → PCA 15. $n_\text{train} = 20000$. Phase 1–2 = 7%.

| Model | Train Acc | Test Acc | Train Loss | Test Loss |
|---|---|---|---|---|
| WBSNN ($\alpha_{(k,m)}$) | 0.975562 | 0.97200 | 0.061 | 0.077 |
| WBSNN ($\alpha_k$) | 0.972750 | 0.96675 | 0.072 | 0.085 |
| Logistic Regression | 0.784700 | 0.77450 | 0.641 | 0.684 |
| Random Forest | 1.000000 | 0.97250 | 0.022 | 0.086 |
| SVM (RBF) | 0.978000 | 0.97400 | 0.054 | 0.064 |
| MLP (1 hidden layer) | 0.978400 | 0.97025 | 0.061 | 0.080 |
| k-NN (k=15, dist) | 1.000000 | 0.96875 | 1e-15 | 0.097 |
| NAIS-Net | 0.985200 | 0.97150 | 0.037 | 0.073 |
| Diffusion Maps + LogReg | 0.602750 | 0.60525 | 1.175 | 1.192 |
| KRR | 0.962700 | 0.95475 | 0.061 | 0.077 |

Table 45: Swiss Roll (RFF) — regression. Embedding: RFF 20 → PCA 10. $n_\text{train} = 1000$. Phase 1–2 = 10%.

| Model | Train Loss | Test Loss | Train MSE | Test MSE | Train $R^2$ | Test $R^2$ |
|---|---|---|---|---|---|---|
| WBSNN ($\alpha_{(k,m)}$) | 0.008 | 0.009 | 0.007816 | 0.009461 | 0.812828 | 0.779148 |
| WBSNN ($\alpha_k$) | 0.008 | 0.009 | 0.007826 | 0.009411 | 0.812575 | 0.780305 |
| Linear Regression | 0.017 | 0.016 | 0.017445 | 0.016068 | 0.576039 | 0.624893 |
| Random Forest | 0.001 | 0.007 | 0.000935 | 0.006833 | 0.977286 | 0.840497 |
| SVR | 0.005 | 0.006 | 0.005012 | 0.006056 | 0.878196 | 0.858630 |
| MLP (1 hidden layer) | 0.008 | 0.009 | 0.008447 | 0.009382 | 0.794704 | 0.780983 |
| k-NN (k=15, dist) | 0.000 | 0.005 | 0.000000 | 0.004952 | 1.000000 | 0.884408 |
| KRR | 0.008 | 0.009 | 0.008866 | 0.008799 | 0.784527 | 0.794587 |

Table 46: Swiss Roll (RFF) — regression. Embedding: RFF 30 → PCA 15. $n_{\text{train}} = 20000$. Phase 1–2 = 7%.

| Model | Train Loss | Test Loss | Train MSE | Test MSE | Train $R^2$ | Test $R^2$ |
|---|---|---|---|---|---|---|
| WBSNN ($\alpha_{(k,m)}$) | 0.009 | 0.009 | 0.008925 | 0.009185 | 0.775105 | 0.777464 |
| WBSNN ($\alpha_k$) | 0.006 | 0.007 | 0.006355 | 0.006979 | 0.839865 | 0.830914 |
| Linear Regression | 0.021 | 0.022 | 0.021082 | 0.022086 | 0.467190 | 0.464876 |
| Random Forest | 0.001 | 0.005 | 0.000689 | 0.005421 | 0.982576 | 0.868646 |
| SVR | 0.005 | 0.005 | 0.004622 | 0.005114 | 0.883175 | 0.876097 |
| MLP (1 hidden layer) | 0.005 | 0.006 | 0.004877 | 0.005561 | 0.876738 | 0.865261 |
| k-NN (k=15, dist) | 0.000 | 0.005 | 0.000000 | 0.005128 | 1.000000 | 0.875762 |
| KRR | 0.009 | 0.009 | 0.004701 | 0.005127 | 0.881200 | 0.875785 |

**Swiss Roll + Polynomial (setup).** From 13k points on the 3D roll with noise $\in \{0.2, 0.5, 0.8\}$, we build a 15-D feature stack: $(x, y, z)$, quadratics $(x^2, y^2, z^2)$, pairwise products $(xy, xz, yz)$, plus six $\mathcal{N}(0, 0.1)$ spurious channels; 80/20 split, $M_{\text{train}} \in \{500, 2k, 5k, 10k\}$, $M_{\text{test}}=0.2M_{\text{train}}$, PCA bottleneck $d \in \{5, 10, 15\}$, z-score, Phase 1–2 discovery 10–20% with Phase 2 one-hot (10-way).

**Why polynomial (vs. RFF).** Deterministic low-degree interactions expose curvature and spurious correlations explicitly; we then ask WBSNN to compress (PCA) and still separate under rising noise and tiny discovery budgets.

**Summary at noise 0.2.** At $d=10$, 10k test 98.05%, tracking RBF-SVM/MLP; LogReg nip at the top, as the poly space is near-linear.

**Summary at noise 0.5.** At $d=10$, 5k→95.3%; widening to $d=15$ at fixed 5k dips to 93.6%—smaller $d$ acts like a low-pass under noise.

**Summary at noise 0.8.** Small-$d$ wins: with 500 points, $d=5$ WBSNN hits 96.0% top performer; Nyström KRR and Random Forest overfit (100% train, low-82% test on Nyström vs. 93% on RF).

**Takeaways.** Prefer small $d$ as noise rises; gains come from more samples and lean discovery, not bigger heads; WBSNN stays competitive/robust while baselines might memorize drift.

**Selected runs (representative runs shown; complete runs (all configs) are in the accompanying GitHub repository).** (i) *Low-noise ceiling:* noise 0.2, $d=10$, $M_{\text{train}}=10$k (shows near-linear separability and WBSNN parity: $\approx 98.05\%$). (ii) *Noise+variance trade-off:* $d=10$ vs. 15 at 5k, noise 0.5 (95.3% vs. 93.6%) to illustrate "small-$d$ denoises." (iii) *High-noise, small-data efficiency:* $d=5$, $M_{\text{train}}=500$, noise 0.8 (96.0%) to highlight discovery-budget efficiency.

Table 47: Swiss Roll (Polynomial) — noise = 0.2. PCA $d=10$, $n_{\text{train}}=10,000$, discovery 10%.

| Model | Train Acc | Test Acc | Train Loss | Test Loss |
|---|---|---|---|---|
| WBSNN ($\alpha_{(k,m)}$) | 0.982625 | 0.9805 | 0.049220 | 0.048053 |
| Logistic Regression | 0.983500 | 0.9820 | 0.058884 | 0.059518 |
| Random Forest | 1.000000 | 0.9805 | 0.013727 | 0.048484 |
| SVM (RBF) | 0.983100 | 0.9775 | 0.043808 | 0.052378 |
| MLP (1 hidden layer) | 0.985400 | 0.9815 | 0.034195 | 0.040197 |
| NAIS-Net | 0.985200 | 0.9825 | 0.037436 | 0.042702 |
| Kernel Ridge (Nyström RBF) | 0.994700 | 0.9485 | 0.818758 | 0.888262 |
| Label Propagation (RBF) | 0.981700 | 0.9760 | 0.051147 | 0.059930 |
| Diffusion Maps + LogReg | 0.979500 | 0.9780 | 0.096964 | 0.096583 |

Table 48: Swiss Roll (Polynomial) — noise = 0.5. PCA $d$=10, $n_{\text{train}}$=5,000, discovery 10%.

| Model | Train Acc | Test Acc | Train Loss | Test Loss |
|---|---|---|---|---|
| WBSNN ($\alpha_{(k,m)}$) | 0.96375 | 0.953 | 0.090803 | 0.114413 |
| Logistic Regression | 0.95880 | 0.954 | 0.104122 | 0.115466 |
| Random Forest | 1.00000 | 0.945 | 0.031183 | 0.159319 |
| SVM (RBF) | 0.95900 | 0.950 | 0.095327 | 0.116598 |
| MLP (1 hidden layer) | 0.96260 | 0.955 | 0.082152 | 0.105968 |
| NAIS-Net | 0.96200 | 0.951 | 0.087675 | 0.122909 |
| Kernel Ridge (Nyström RBF) | 0.99640 | 0.910 | 0.794089 | 0.937682 |
| Label Propagation (RBF) | 0.95740 | 0.943 | 0.099558 | 0.158828 |
| Diffusion Maps + LogReg | 0.95620 | 0.941 | 0.156893 | 0.177376 |

Table 49: Swiss Roll (Polynomial) — noise = 0.5. PCA $d$=15, $n_{\text{train}}$=5,000, discovery 10%.

| Model | Train Acc | Test Acc | Train Loss | Test Loss |
|---|---|---|---|---|
| WBSNN ($\alpha_{(k,m)}$) | 0.98525 | 0.936 | 0.053564 | 0.196517 |
| Logistic Regression | 0.96280 | 0.950 | 0.099798 | 0.119264 |
| Random Forest | 1.00000 | 0.945 | 0.038015 | 0.139064 |
| SVM (RBF) | 0.97060 | 0.937 | 0.080215 | 0.148304 |
| MLP (1 hidden layer) | 0.99840 | 0.938 | 0.011080 | 0.232138 |
| NAIS-Net | 0.96160 | 0.950 | 0.087023 | 0.121348 |
| Kernel Ridge (Nyström RBF) | 0.99480 | 0.910 | 0.797563 | 0.930236 |
| Label Propagation (RBF) | 0.95740 | 0.943 | 0.099558 | 0.158828 |
| Diffusion Maps + LogReg | 0.95620 | 0.941 | 0.156893 | 0.177376 |

Table 50: Swiss Roll (Polynomial) — noise = 0.8. PCA $d$=5, $n_{\text{train}}$=500, discovery 20%.

| Model | Train Acc | Test Acc | Train Loss | Test Loss |
|---|---|---|---|---|
| WBSNN ($\alpha_{(k,m)}$) | 0.9475 | 0.9600 | 0.200217 | 0.162914 |
| Logistic Regression | 0.9040 | 0.9300 | 0.364154 | 0.343107 |
| Random Forest | 1.0000 | 0.9300 | 0.082212 | 0.242832 |
| SVM (RBF) | 0.9120 | 0.9400 | 0.272589 | 0.253670 |
| MLP (1 hidden layer) | 0.9480 | 0.9500 | 0.137418 | 0.139910 |
| NAIS-Net | 0.9800 | 0.9400 | 0.061205 | 0.200780 |
| Kernel Ridge (Nyström RBF) | 1.0000 | 0.8200 | 0.734382 | 1.278720 |
| Label Propagation (RBF) | 0.9060 | 0.8900 | 0.262146 | 0.294173 |
| Diffusion Maps + LogReg | 0.9380 | 0.9400 | 0.321441 | 0.344973 |

**Swiss Roll (raw 3D) — Setup & Summary Swiss Roll (raw 3D) – setup:** Data: raw $(x, y, z)$, standardized; no feature maps. Variants: noisy_3class ($\sigma = 0.5$, 3 bins), low_sample_label_noise (10 bins, 10% label flips), multi_roll (3 spirals, 10 bins), regression (unwrapped $t \in [0, 1]$, $\sigma = 0.1$).

**Splits/budgets:** 80/20 pool; sample $M_{\text{train}}$ with $M_{\text{test}} = 0.2\,M_{\text{train}}$. Phase-1/2 discovery 10–20% at small $M$, shrinking to 7% by 20k.

**Summary:** noisy_3class $\sim 99.4\%$ test at 10% discovery; while logistic $\sim 61\%$.

low_sample_label_noise: 91–91.5% at 500–1k despite 10% flips; small MLP collapses at 500 (46%).

multi_roll: 98.7% as 20k with 7% discovery budget; topology handled well under tight budgets.

regression: shared $\alpha_k$ generalizes best (e.g., $R^2 \approx 0.986$ at 2k vs 0.969 for $\alpha_{(k,m)}$); wider head not needed.

**Selected runs (representative runs shown; complete runs (all configs) are in the accompanying GitHub repository)):** noisy_3class (10k, 10%), low_sample_label_noise (500, 20%) and (1k, 15%), multi_roll (20k, 7%), regression (2k, 10%) and (10k, 10%).

**Why:** geometry win without features; robustness under scarcity+noise; topology+scaling+budget efficiency; clear bias–variance readout.

Table 51: Swiss Roll (raw 3D) — `noisy_3class`. $n_{\text{train}}$=10,000, discovery 10%.

| Model | Train Acc | Test Acc | Train Loss | Test Loss |
|---|---|---|---|---|
| WBSNN ($\alpha_{(k,m)}$) | 0.994625 | 0.9935 | 0.014 | 0.018 |
| WBSNN ($\alpha_k$) | 0.994125 | 0.9935 | 0.014 | 0.016 |
| Logistic Regression | 0.624200 | 0.6100 | 0.611 | 0.622 |
| Random Forest | 1.000000 | 0.9940 | 0.006 | 0.024 |
| SVM (RBF) | 0.994500 | 0.9945 | 0.014 | 0.015 |
| MLP (1 hidden layer) | 0.994100 | 0.9930 | 0.055 | 0.056 |
| k-NN (k=15, dist) | 1.000000 | 0.9915 | 1e-15 | 0.014 |
| NAIS-Net | 0.994800 | 0.9930 | 0.012 | 0.015 |

Table 52: Swiss Roll (raw 3D) — `low_sample_label_noise` (10% flips). $n_{\text{train}}$=500, discovery 20%.

| Model | Train Acc | Test Acc | Train Loss | Test Loss |
|---|---|---|---|---|
| WBSNN ($\alpha_{(k,m)}$) | 0.89500 | 0.910 | 0.488 | 0.787 |
| WBSNN ($\alpha_k$) | 0.90000 | 0.910 | 0.512 | 0.900 |
| Logistic Regression | 0.88000 | 0.880 | 0.939 | 1.120 |
| Random Forest | — | — | — | — |
| SVM (RBF) | 0.89400 | 0.890 | 0.616 | 0.715 |
| MLP (1 hidden layer) | 0.54600 | 0.460 | 1.605 | 1.753 |
| k-NN (k=15, dist) | 1.00000 | 0.870 | 9.99e-16 | 3.04 |
| Label Propagation (RBF) | 0.87400 | 0.840 | 0.412 | 1.931 |

Table 53: Swiss Roll (raw 3D) — `low_sample_label_noise` (10% flips). $n_{\text{train}}$=1,000, discovery 15%.

| Model | Train Acc | Test Acc | Train Loss | Test Loss |
|---|---|---|---|---|
| WBSNN ($\alpha_{(k,m)}$) | 0.90250 | 0.915 | 0.483 | 0.654 |
| WBSNN ($\alpha_k$) | 0.90125 | 0.915 | 0.508 | 0.652 |
| Logistic Regression | 0.87400 | 0.900 | 1.008 | 1.082 |
| Random Forest | 0.99600 | 0.920 | 0.204 | 0.516 |
| SVM (RBF) | 0.88800 | 0.915 | 0.588 | 0.603 |
| MLP (1 hidden layer) | 0.87700 | 0.875 | 1.012 | 1.034 |
| k-NN (k=15, dist) | 1.00000 | 0.905 | 1e-15 | 2.47 |
| Label Propagation (RBF) | 0.88400 | 0.885 | 0.381 | 1.657 |

Table 54: Swiss Roll (raw 3D) — `multi_roll` (3 roll). $n_{\text{train}}$=20,000, discovery 7%.

| Model | Train Acc | Test Acc | Train Loss | Test Loss |
|---|---|---|---|---|
| WBSNN ($\alpha_{(k,m)}$) | 0.989688 | 0.98675 | 0.028 | 0.033 |
| WBSNN ($\alpha_k$) | 0.987500 | 0.98600 | 0.035 | 0.040 |
| Logistic Regression | 0.984550 | 0.98525 | 0.103 | 0.099 |
| Random Forest | 1.000000 | 0.99100 | 0.010 | 0.040 |
| SVM (RBF) | 0.990950 | 0.98850 | 0.027 | 0.031 |
| MLP (1 hidden layer) | 0.988800 | 0.98700 | 0.056 | 0.057 |
| k-NN (k=15, dist) | 1.000000 | 0.98100 | 1e-15 | 0.052 |
| NAIS-Net | 0.993650 | 0.98975 | 0.016 | 0.021 |
| Diffusion Maps + LogReg | 0.990200 | 0.99000 | 0.048 | 0.048 |

Table 55: Swiss Roll (raw 3D) - `regression` ($t/t_{\max}$). $n_{\text{train}} = 2000$, discovery 10%.

| Model | Train Loss | Test Loss | Train MSE | Test MSE | Train $R^2$ | Test $R^2$ |
|---|---|---|---|---|---|---|
| WBSNN ($\alpha_{(k,m)}$) | 0.001 | 0.001 | 0.001436 | 0.001281 | 0.960250 | 0.968505 |
| WBSNN ($\alpha_k$) | 0.001 | 0.001 | 0.000597 | 0.000581 | 0.983486 | 0.985709 |
| Linear Regression | 0.034 | 0.038 | 0.034445 | 0.038237 | 0.064788 | 0.060128 |
| Random Forest | 0.000 | 0.000 | 0.000030 | 0.000002 | 0.999198 | 0.999956 |
| SVR | 0.007 | 0.007 | 0.006636 | 0.006758 | 0.819834 | 0.833899 |
| MLP (1 hidden layer) | 0.001 | 0.001 | 0.001272 | 0.001383 | 0.965477 | 0.966017 |
| k-NN (k=15, dist) | 0.000 | 7.23e-6 | 0.000000 | 0.000007 | 1.000000 | 0.999822 |

Table 56: Swiss Roll (raw 3D) — `regression` ($t/t_{\max}$). $n_{\text{train}}$=10,000, discovery 10%.

| Model | Train Loss | Test Loss | Train MSE | Test MSE | Train $R^2$ | Test $R^2$ |
|---|---|---|---|---|---|---|
| WBSNN ($\alpha_{(k,m)}$) | 0.002 | 0.002 | 0.002452 | 0.002416 | 0.934906 | 0.937793 |
| WBSNN ($\alpha_k$) | 0.002 | 0.002 | 0.001532 | 0.001530 | 0.959314 | 0.960600 |
| Linear Regression | 0.035 | 0.036 | 0.035059 | 0.035947 | 0.070930 | 0.074468 |
| Random Forest | 0.000 | 0.000 | 0.000003 | 0.000001 | 0.999911 | 0.999966 |
| SVR | 0.006 | 0.006 | 0.006382 | 0.006499 | 0.830884 | 0.832676 |
| MLP (1 hidden layer) | 0.000 | 0.000 | 0.000479 | 0.000476 | 0.987302 | 0.987735 |
| k-NN (k=15, dist) | 0.000 | 1.65e-6 | 0.000000 | 0.000002 | 1.000000 | 0.999958 |

