# OpenReview forum: "Beyond Deep Heuristics: A Principled and Interpretable Orbit-Based Learning Framework"
_ICLR.cc/2026/Conference — Submitted to ICLR 2026_

### Official Review · Reviewer_vGXV · 2025-10-19

[review text omitted: it was posted to a different submission]

---

> ### Author Response · Authors · 2025-11-15
> **Response to Reviewer vGXV (Part I)**
>
> We sincerely thank Reviewer vGXV for the thoughtful and positive report, for the clear grasp of our goals, and for the insightful questions, especially those concerning scalability. We are particularly grateful that the reviewer views this work in essentially the same way we do: as a fundamentally theoretical contribution, whose empirical section is meant to illustrate the framework across diverse modalities rather than to position WBSNN as yet another benchmark-optimized architecture.
>
> The central message of the paper is that deep learning is entering a stage where structure must take precedence over heuristics, and WBSNN embodies this shift: an MLP equipped with the correct operator-level structure can match or surpass architectures of much higher apparent complexity. Importantly, this manuscript is not an isolated result but the foundation of a broader research program. Several of the reviewer’s questions connect directly to the follow-up manuscripts that extend this work; we summarize those connections below for clarity.
>
> In one follow-up, we show that WBSNN admits a precise interpretation as a low-rank, single-head attention mechanism (the full proof is omitted here due to space). A second manuscript develops the hierarchical Tree-WBSNN architecture, providing an analog of multi-head attention through a rigorous multi-branch orbit construction. These low-rank formulations reveal that generalization and robustness depend on the quality of the anchors selected in Phase 1, in the sense of how well the resulting scaffold captures the “DNA” of the dataset. Motivated by this, a third manuscript introduces a mathematically heavy but constructive algorithm for anchor selection we coined Constructive Alignment Network (CAN), designed to choose anchors that optimally reduce residuals in the orbit space. Notice that in the experiments of the present paper anchors are chosen at random for simplicity, and yet the results are already strong; the follow-up work is precisely about making this anchor selection principled. Finally, to close this circle of ideas, another manuscript introduces a new topological and sheaf-theoretic framework for datasets, building directly on the exact interpolation theorem established here and proposing a new paradigm for understanding dataset geometry and learnability. Taken together, these works outline a cohesive, mathematically grounded direction for the future of deep learning, centered on simplicity, structure, interpretability, and operator-theoretic principles. In this context, the contribution of the present manuscript is crucial, as it represents the foundational step in a multi-year research effort aimed at providing a rigorous and forward-looking alternative to mainstream heuristic pipelines.
>
> We now respond to Reviewer vGXV’s questions in the order in which they were posed. The remaining concerns center on scalability and on the feasibility of the independence condition in the exact interpolation theorem.
>
> First, regarding the one-time scalability of Phase 1, we emphasize that Phase 1 is the only phase that performs a dataset--level structural computation (building the scaffold), but it does so using only a small anchor subset of the training set (typically $1$--$5\%$), not all samples. Its operations are the following. It generates orbit vectors \(W^{(m)} X_i\); each \(W^{(m)}\) is shift-sparse, with a single nonzero per row, so each application costs \(O(d)\). It maintains an incremental orthonormal basis of accepted orbit vectors, with total cost \(O(r^2 d)\) for final rank \(r\). It then solves a Tikhonov system for the small scaffold matrix, with cost \(O(s^2 d + s^3)\) for scaffold size \(s \ll n\). As a practical matter, Phase 1 therefore scales as \(O(s M d + r^2 d)\) and is executed once, whereas Phases 2 and 3 scale like a small MLP. The cost of Phase~1 is thus governed entirely by the anchor parameters $(s, M, r)$ rather than by the full dataset size.
>
> Second, about the low-rank structure and why Phase 3 behaves like a single-head attention mechanism, once Phase 1 fixes \(K\) anchor subsets and Phase 2 constructs the interpolation maps \(J_k\), Phase 3 computes
> $f(x)=\sum_{k,m} \alpha_{k,m}(x) W^{(m)} J_k x,$
> which is a linear combination of at most \(r = K M\) orbit directions. Hence predictions lie in a rank-\(r\) subspace spanned by orbit vectors. Structurally, this is a low-rank single-head, content-based mixing layer: the orbit vectors play the role of values, and the coefficients \(\alpha_{k,m}(x)\) play the role of attention weights. The inference cost is \(O(r d)\), not \(O(n^2)\). Scaling is therefore governed by the effective rank \(r\), which in turn is set entirely by the Phase-1 scaffold size and not by the dataset size.

---

> ### Author Response · Authors · 2025-11-15
> **Response to Reviewer vGXV (Part II)**
>
> Third, the sensitivity to the independence condition (Condition (4)) is naturally controlled by dimension. The condition ensures that each anchor contributes independent orbit directions. In high-dimensional regimes, independence is typically straightforward: each anchor contributes many independent directions, so the required number of subsets \(K\) is small. In low-dimensional regimes, such as our PCA-compressed settings, each anchor contributes at most
> \(d\) directions, so achieving a meaningful global rank requires selecting anchors whose orbit spans are well spread relative to the underlying data geometry. Although our experiments used random anchors and still produced stable interpolation and strong generalization, this is consistent with the fact that PCA features are orthogonal and weighted shifts preserve this structure up to scaling. In genuinely low-dimensional regimes, however,
> \(K\) may grow unless anchors are chosen to maximize the incremental gain in independent orbit directions; this is precisely why optimal anchor selection becomes important for keeping both
> \(K\) and the total number of parameters under control. Formally, in our low-rank operator view the approximation error decomposes as
> $
> \| A - U C U^{\top} \| \le \sigma_{r+1} + \varepsilon_{\mathrm{proj}} + \varepsilon_{\mathrm{param}},
> $
> and the term \(\varepsilon_{\mathrm{proj}}\) is governed exactly by how well the scaffold subspace aligns with the true signal subspace; that is, by anchor quality. The sensitivity question therefore maps directly to a standard, well-understood linear-algebraic quantity.
>
> Fourth, concerning why our follow-up work on CAN resolves the anchor-selection issue, we briefly summarize the relevant aspects while deliberately not relying on experimental claims. The Constructive Alignment Network (CAN) is a companion model that replaces random anchor selection with a residual-guided, constructive mechanism. At a high level, CAN builds the output through an iterative residual-alignment process of the form
> $
> r_{t+1} = r_t - \lambda_t q_t,
> $
> where each \(q_t\) is a normalized orbit direction chosen to reduce the current residual as much as possible. In this way, CAN systematically identifies those orbit directions that are most informative for the task. This has three consequences. First, CAN improves scaffold quality for a fixed \(K\), because it promotes orbit directions that capture previously unexplained structure. Second, CAN reduces sensitivity to the exact independence condition, since it naturally selects an effective independent subset even when the initial scaffold contains correlated directions. Third, in the low-rank operator view CAN directly reduces the projection error \(\varepsilon_{\mathrm{proj}}\), since each step is designed to align with and diminish the unexplained component. Thus CAN formalizes and addresses precisely the optimal-anchor problem raised in the reviewer’s question, especially in low-dimensional regimes, and integrates naturally with the theoretical framework introduced in this paper.

---

> ### Author Response · Authors · 2025-11-15
> **Response to Reviewer vGXV (Part III)**
>
> We now address the question about backbone versus head, and in particular the suggestion of fixing the shift operator \(W\) (random or orthogonal) to isolate the contribution of the MLP head. Empirically, the evidence in the paper already points to the backbone as the key driver. In the experiments accompanying this manuscript, on multiple datasets where the head receives exactly the same compressed features, WBSNN achieves strictly higher performance, which already suggests that the gains are not due solely to head flexibility but to the structured orbit backbone. For example, on the RFF–PCA15 Swiss Roll with $10\%$ noise (Table 42), WBSNN reaches (on test accuracy) $0.750$ versus $0.715$ for the one-layer MLP. On IMDb with PCA input ($d=20$ and $n_{\mathrm{train}}=4000$, Table 9), WBSNN reaches (on test accuracy) $0.7738$, substantially higher than the MLP’s $0.7088$. Since the heads have comparable expressive capacity and operate on identical inputs, these consistent margins strongly suggest that the structured orbit backbone, rather than head flexibility, is responsible for the gains. Our broader aim, however, is conceptual and theoretical rather than purely empirical, so we prefer to answer this question at the operator level. In our setting, \(W\) is a vanishing weighted backward shift, learned to align orbit directions with the data geometry and to operate in a contracting regime. A random \(W\) would generate essentially directions that are uncorrelated with the dataset structure, while an orthogonal (unit-modulus) \(W\) would remove contraction and thus the denoising effect we analyze. In both cases the orbit span \(U\) becomes a largely generic low-rank subspace, so the head degenerates to a standard MLP on random features.
>
> More formally, in the operator view the Phase-1 scaffold produces a rank-\(r\) subspace \(U\), and its alignment with the true signal subspace controls the projection error
> $
> \varepsilon_{\mathrm{proj}} = \|A_r - U A_U U^{\top}\|.
> $
> A learned vanishing shift \(W\) aligns orbit vectors with the data geometry, enabling Phase 1 to select anchors whose orbits span a meaningful subspace \(U\). If \(W\) is instead replaced with a random or orthogonal shift, the orbits become unstructured or purely isometric (since orbit iterates produced by an orthogonal/unit-modulus shift preserve norms, eliminating contraction and preventing the orbit-decay mechanism that underlies robustness). Phase 1 then selects anchors whose orbit vectors lie in arbitrary directions; in other words, the scaffold subspace
> $
> U = \operatorname{span} \{ W^{(m)} X_i \}
> $
> becomes a random low-rank subspace that is not aligned with the true structure of the data. In this situation Phase 1 is forced to accept orbit vectors inside this random space, and the scaffold becomes ineffective. The resulting \(\varepsilon_{\mathrm{proj}}\) increases sharply, and the head effectively receives random low-rank features, reducing to an MLP on random projections. This theoretical effect mirrors the CAN mechanism: CAN successfully reduces the residual only when orbit directions correlate with the signal; random or orthogonal \(W\) eliminates this correlation and therefore breaks the residual descent. For this reason, fixing \(W\) as random or orthogonal is expected in theory to degrade performance precisely because it worsens the scaffold subspace \(U\) and inflates the approximation error in our low-rank characterization. We thus view explicit random/orthogonal-\(W\) ablations as a natural direction for follow-up experiments, but the current theory and baselines already indicate that the backbone, rather than the MLP head, is the main driver.

---

> ### Author Response · Authors · 2025-11-15
> **Response to Reviewer vGXV (Part IV)**
>
> Regarding large-scale behavior on high-resolution image or spatiotemporal data, we have not yet run experiments on very high-resolution images or large spatiotemporal grids; the present study focuses on compressed representations such as PCA-reduced tabular features and PCA-compressed ResNet-18 features for CIFAR-100. However, the structure of WBSNN is designed so that increasing the feature dimension \(d\) actually makes Phase 1 easier rather than harder. In higher dimension, the orbit vectors \(W^{(m)} X_i\) are generically linearly independent up to \(d\), so each anchor set \(D_k\) can contribute substantial rank, and the number of subsets \(K\) can remain small. Moreover, although in our experiments we set the orbit horizon \(M = d\) for simplicity, the theory (contraction of \(W\) and modulo-\(d\) collapse of iterates) implies a natural effective horizon \(M_{\mathrm{eff}} \ll d\), beyond which additional iterates add negligible new information. An adaptive Phase-1 scheme that truncates orbits when norms decay or when new iterates become dependent would therefore keep the effective rank \(r = K M_{\mathrm{eff}}\) and both memory and runtime under tight control. In other words, scaling WBSNN to high-resolution regimes is governed by the choice of the effective rank \(r\), not by the raw dimension \(d\). Designing such an adaptive \((K, M)\)-selection mechanism is an explicit direction for future work and fits naturally with the residual-based CAN framework introduced in the companion work.
>
> In response to the question on the interpretability of the learned shifts, we emphasize that weighted backward shifts are highly transparent operators. Each learned $W$ has exactly one nonzero per row, so its action is easy to parse coordinate-wise: every output coordinate is simply a rescaled, shifted copy of an input coordinate. The iterates $W^{(L)}$ used in WBSNN follow this shift along the orbit, and Lemma 3.6 in the paper shows how these iterates collapse modulo~$d$ and eventually enter a vanishing regime. The contraction condition $\prod_i |w_i| < 1$, which is central in guaranteeing orbit decay: the norms $\|W^{(L)} x\|$ shrink with $L$, and this shrinking is precisely the mechanism responsible for the denoising and stability behaviour analyzed in our orbit-based framework. This structure also makes the learned parameters directly interpretable. The weights $(w_i)$ can be visualized as a simple one-dimensional filter, and the corresponding decay curves $L \mapsto \|W^{(L)}x\|$ can be inspected without any additional machinery. In practice, smoother and more strongly contracting weight profiles correlate with the robust generalization regimes observed across the datasets in our study. While a deeper spectral analysis (e.g., connecting orbit-decay profiles with the spectral radius or pseudospectrum of $W$) is an intriguing direction---and a topic of our ongoing theoretical work---it is not required for the present paper and is not claimed here. For the purposes of this submission, interpretability stems directly from the shift-sparse structure, the contraction regime, and the orbit-decay behaviour already formalized in the main text.
>
> We close by noting a possible scoring oversight. The updated textual review is extremely positive, emphasizing strong novelty, theory, clarity, and breadth of experiments, yet the numerical ratings appear unchanged from the earlier mismatch version. If this is unintentional, we would kindly ask the reviewer to verify that the final scores reflect the updated assessment. We thank Reviewer vGXV again for the careful reading and insightful questions, and we would be very happy to address any further questions or requests for clarification, as such feedback continues to sharpen our understanding and presentation of this research program.

---

> > ### Comment · Reviewer_vGXV · 2025-11-26
> >
> > Thank you for the detailed response and clarifications. While the additional context is appreciated, several original concerns remain insufficiently addressed:
> >
> > Regarding scalability, the response provides theoretical complexity estimates, but the paper still lacks empirical validation (runtime, memory, or hardware comparisons). Given the claimed practical efficiency, some experimental evidence is needed.
> >
> > Regarding independence condition, the explanation is intuitive, but no quantitative analysis is provided. It remains unclear how often Condition (4) fails, how performance degrades, or whether approximate independence is sufficient in practice.
> >
> > For backbone vs. head, the theoretical arguments are noted, but the requested ablations using fixed random or orthogonal shift operators are still missing. These are essential to isolate the contribution of the orbit backbone.
> >
> > Large-scale applicability: the authors confirm that high-resolution or large-grid experiments have not been conducted. Since all current results rely on compressed inputs or pre-extracted features, the practical scalability of WBSNN remains unverified.
> >
> > Regarding interpretability: the shift structure is inherently interpretable, but the paper would benefit from concrete visualizations or quantitative analyses rather than descriptive claims alone.
> >
> > For the reasons above, I maintain my original rating.

---

> > > ### Author Response · Authors · 2025-11-27
> > > **Response to Reviewer vGXV**
> > >
> > > We thank Reviewer vGXV for the follow-up and for their continued engagement with the paper. We respect their decision. We would, however, like to offer a brief clarification to ensure the structure of the model is fully understood.
> > >
> > > Regarding “backbone vs. head,” we respectfully note that the requested ablations (e.g., fixing W as random or orthogonal) are not essential for isolating the contribution of the orbit backbone. In addition to the explanations we provided earlier in our response above, Equation (5) makes this structural point explicit: the Phase-3 MLP does not act on raw data, but only produces coefficients \alpha_{k,m}(X). Therefore, the backbone forces all predictions to lie in the low-dimensional manifold
> > >
> > > $\mathcal{H}= \left\lbrace\sum_{k,m} \alpha_{k,m}(X) J_k W^{(m)} X: \alpha= (\alpha_{k,m})_{k, m}\in  \mathbb{R}^{Kd}\right\rbrace.$
> > >
> > >
> > >
> > >
> > > This is a strict subset of all possible functions an MLP could learn. Thus—even if the head sees all training points—its expressive power is strictly bounded by the orbit scaffold (W, J_k) constructed in Phases 1–2. The architecture does not allow the MLP to bypass this structure; it can only recombine orbit directions determined by the backbone. In this sense, WBSNN restricts the hypothesis class rather than enlarging it, and the geometry encoded by the backbone is the determining factor.
> > >
> > > Finally, what the reviewer refers to as an “intuitive explanation” is, in our view, a principled and theoretically motivated analysis grounded in the operator-theoretic structure of the model. We note that no errors or inconsistencies in the theoretical results have been identified, and we remain fully open to further scrutiny or criticism should any arise.
> > >
> > > We hope this brief clarification helps reinforce the intent and internal logic of the framework.

---

> > > ### Author Response · Authors · 2025-11-29
> > > **A brief clarification regarding Reviewer vGXV’s last remark on Condition (4).**
> > >
> > > In the current paper, exact independence is never assumed as a hard requirement in noisy or compressed regimes. Phase 1 tests span membership via a least–squares residual, which we use as a practical proxy for exact independence, and Phase 2 defaults (in our main protocol) to a regularized least–squares interpolation precisely to handle rank deficiency and noisy anchors.
> > > This is what we refer to as the relaxed (non-exact) interpolation modality: it intentionally sacrifices exact fitting in favor of stability and robustness, and it is the default choice in our main experimental results.
> > >
> > > We agree that we do not provide a quantitative tally of how often Condition (4) fails across all datasets; adding such diagnostics would require additional sweeps beyond the present scope. Conceptually, however, the architecture is designed so that approximate independence—implemented via the tolerance-based span test and the regularized pseudoinverse—is sufficient in practice and is exactly the regime we study in all noisy settings.

---

### Official Review · Reviewer_pWSz · 2025-10-27

**Soundness:** 2
**Presentation:** 2
**Contribution:** 2
**Rating:** 2
**Confidence:** 3

**Summary:**

This paper introduces the Weighted Backward Shift Neural Network (WBSNN) architecture. Inspired by the theory of linear dynamics, WBSNN is a light-weighted alternative to the widely used over-parameterized architectures such as Multilayer Perceptrons. The WBSNN consists of 3 components: a weighted shift operator $W$ that applies cyclic shift and weighting iteratively in a multi-level fashion to generate orbit dictionary, simple linear regressors $J_k$ that maps the transformed vector to the output, for each layer, and a neural network based weight predictor that dynamically combines the linear orbit predictor. WBSNN can be trained efficiently with limited data budget thanks to its simplicity in design.

**Strengths:**

- The paper presents a clear formalization of the proposed three-phase process, with precise definitions and lemmas describing how the model parameters are obtained. The theoretical analysis centers on proving the exactness of the local interpolation step in Phase 2. While this result has limited practical implications, it demonstrates an effort toward mathematical rigor and internal consistency.

- Because all components are explicitly linear or low-dimensional, the architecture is more interpretable than the often over-parameterized black-box approximators, such as neural networks.

- The authors provide examples to facilitate understanding while explaining through the operator-theoretic part of the manuscripts

**Weaknesses:**

- The experiment section of the manuscript lacks details for the problem set up, which leads to confusion for the readers. The manuscript also frequently refers to the appendices, which consist of over 50 tables. I am skeptical about this writing style, since it effectively circumvents the page limit by forcefully point the readers to the appendices.
- With over fifty benchmarks, where many involving custom preprocessing steps such as PCA compression, the experimental section becomes overwhelming. The abundance of results obscures the main message of the paper. The authors should consider condensing this section by retaining only a few representative experiments that clearly demonstrate the cost-effectiveness and performance of the proposed architecture.
- The paper lacks any analysis of training or inference cost. Without runtime or compute comparisons, it is unclear whether WBSNN is computationally efficient, especially relative to simpler baselines such as linear regression or SVM. The training budget is misleading since it has to be trained on the entire dataset regardless during phase 3.
- It is incorrect to emphasis that the model is linear, due to the MLP mixer in phase 3 (eq 5). The experiments that the authors performed do not demonstrate if the phased training protocol brings the claimed strength, and it would be worthwhile to include a comparison example, where a model with all the parameters including the interpolators Js, backward shift operator Ws, and the MLP mixers, entirely learnable, is trained in a usual 1 stage training only with the loss objectives. The differences, if any, between the performance of the phased training and 1 stage training, would justify the claim that the authors made about the carefully designed phased training protocol.
- Among most of the benchmarks, no substantial improvement can be seen against other classical baselines, but the complexity from the phased training protocol limits its usage as the building block for other models.

**Questions:**

Please address the weaknesses. Further, I have the following suggestions:

- Section 4 should be completely rewritten. I would recommend choosing at most 8 benchmarks and put tabulated results in the main manuscript for the sake of readability. Also, more formal description of the problems should be included, e.g., the author should explicitly show the input and output dimension of the problem, type of the task (regression, classification, etc), the loss objective used for the problem, as well as the hyperparameters for all the models.
- Figures and tables should be provided in the main manuscript to facilitate understanding to the methodology and the interpretation of the results.
- A short paragraph can be added at the beginning of section 3 to emphasize that WBSNN is not a time series forecasting model. The usage of backward shifting operator can be confusing on non-sequential data.

---

> ### Author Response · Authors · 2025-11-13
> **Response to Reviewer pWSz**
>
> We thank reviewer pWSz for the effort, but several key criticisms stem from misinterpretations of the nature and goals of the paper. This work is primarily theoretical: it introduces an operator-theoretic learning framework grounded in exact interpolation theorems and linear-dynamical structure. The experimental section serves only to illustrate coherence across modalities; it is not a benchmark-optimization paper.
>
> The central message is that deep learning is entering a stage where structure must precede heuristics. WBSNN exemplifies this: an MLP endowed with the correct operator structure can match or surpass far more complex architectures. This manuscript is the first step of a broader research program. A second manuscript proves that WBSNN admits a precise low-rank single-head attention interpretation (proof omitted for space). A third companion work develops Tree-WBSNN, implementing a mathematically principled multi-head attention analog via multi-branch orbit construction—offering a structured alternative to the brute-force formulation of Transformers. A fourth manuscript develops a topological and sheaf-theoretic viewpoint on datasets derived from the interpolation theorem proved here. Together, these works outline a coherent operator-theoretic direction for deep learning centered on structure, interpretability, and mathematical rigor. A fair reassessment of the present manuscript is therefore crucial.
>
> Regarding the reviewer’s scores: soundness and contribution are addressed throughout our detailed responses. On presentation, we note that the contribution is inherently mathematical, drawing on functional analysis and linear dynamics. Such material may feel unfamiliar, but the exposition is rigorous, coherent, and precise. As ML increasingly incorporates sophisticated mathematics, clarity should be judged by internal logic rather than by proximity to standard deep-learning exposition.
>
> Addressing weaknesses: The paper is not intended as an empirical benchmarking effort. Its aim is theoretical: we introduce the operator-theoretic framework, prove exact interpolation theorems in finite and infinite dimensions, provide a constructive proof, establish noise suppression, and interpret vanishing regimes. Experiments merely show that the structure behaves coherently across modalities.
>
> On computational cost: WBSNN is a structured alternative to brute-force architectures. Phase 3 indeed trains on the full dataset, but only through a small MLP whose cost is negligible compared to the one-time scaffold discovery of Phase 1. Because \(W^{(m)}\) has one nonzero per row, orbit iterates run in \(O(d)\), and the anchor budget 1%--10 % makes the Tikhonov step inexpensive. Follow-up work formally connects WBSNN to low-rank single-head attention, and Tree-WBSNN to multi-head attention, showing that scaling should not be evaluated as if the model were dense and unconstrained. A complete analysis can be found on our Response to Reviewer vGXV.
>
> The claim that we “incorrectly emphasize that the model is linear’’ reflects a misunderstanding. We explicitly state that only the backbone operators \(W\), \(J_k\), and the orbit dictionary are linear; the full model becomes nonlinear in Phase 3 via the MLP map \(\alpha_{k,m}(X)\). Predictions depend on the entire orbit \(\{W^{(m)}X\}\), a source of nonlinearity well known in linear dynamics. This distinction is clearly stated in the manuscript.
>
> The suggestion to compare our phased protocol with a fully learnable one-stage model misses the foundation of WBSNN: Phases 1–2 are tied to the interpolation theorem and ensure independence and interpolation guarantees. Making them learnable removes this structure, reducing the model to an opaque MLP. Such a comparison is therefore orthogonal to the contribution.
>
> The comment that experiments show “no substantial improvement’’ misinterprets their role. Across noisy, low-sample, compressed, drifted, and cross-modal regimes, WBSNN matches or exceeds classical baselines while using small anchor sets, orbit-structured components, and fewer parameters. The point is coherence and efficiency, not marginal benchmark gains.
>
> Addressing questions: All essential setup details—dimensions, preprocessing, PCA, loss, hyperparameters—are explicitly provided in Section 4. If helpful, we can add a compact summary table. The placement of detailed tables in the appendix is standard practice for theory-heavy papers with broad empirical validation.
>
> Finally, WBSNN is explicitly modality-agnostic. The backward shift operator generates orbit dictionaries in vector spaces; it is not temporal. Submission to the time-series/dynamics track reflects the absence of a more suitable category.
>
> In light of these clarifications, we respectfully request that Reviewer pWSz reassess the manuscript. Most criticisms concern stylistic preferences or misreadings, not errors in theory or methodology. We therefore ask the reviewer to reconsider the “2: reject’’ rating.

---

> ### Author Response · Authors · 2025-11-30
> **Clarification Regarding the Review Below**
>
> vGXV’s review appearing immediately below this comment is mismatched and was originally corrected by the reviewer during the discussion phase, but it has reappeared due to the system-wide rollback following the OpenReview leak.
>
> It describes a variational PDE/VIO architecture that does not appear anywhere in our submission, and is therefore a review of a different paper.
>
> The reviewer’s later follow-up in the thread corresponds to the correct, WBSNN-related review that we responded to in Parts I–IV.
> This note is added only to prevent confusion for the new Area Chair and for future readers.

---

### Official Review · Reviewer_7Zjm · 2025-10-30

**Soundness:** 4
**Presentation:** 3
**Contribution:** 3
**Rating:** 6
**Confidence:** 3

**Summary:**

This manuscript introduces a novel neural architecture termed Weighted Backward Shift Neural Networks (WBSNNs), inspired by the weighted backward shift operator in orbit dynamics—representing a new paradigm that bridges machine learning and operator theory. Specifically, the model recurrently pads the initial features as input features, while each network layer corresponds to one application of the weighted backward shift operator on the input features. Supported by a rigorous and theoretically sound framework, the authors formally prove the network’s representational capability: under suitable weight configurations, there always exists a linear operator such that the input, after transformation by the weighted shift and this operator, exactly reproduces the target labels. To realize this mechanism, the paper proposes a three-phase training scheme: in Phase 1, the dataset is partitioned and the shift weights are optimized; in Phase 2, an optimal linear operator is obtained through a pseudoinverse-based interpolation procedure; and in Phase 3, the model simulates infinite-dimensional dynamics within a constructed functional space, producing the final outputs while jointly optimizing the MLP-based weighting coefficients. The authors claim that this learning paradigm can be effectively applied to a wide range of tasks and modalities, exhibiting consistent performance across domains. Extensive experiments on diverse datasets and benchmarking against a variety of representative models demonstrate the effectiveness, generality, and interpretability of the proposed WBSNN framework.

**Strengths:**

1. The theoretical foundation of this work is solid, and both the model’s learning and generalization capabilities are rigorously guaranteed by formal theorems.
2. The paper conducts extensive experiments across a broad range of tasks to validate the model’s cross-modality learning capability and task adaptability.
3. The model exhibits strong interpretability. Its core architecture is built upon a linear weighted backward shift operator, which is fully traceable—meaning that the dynamical orbit weights are entirely transparent and can explicitly reveal where the model focuses its attention. Notably, the operator approximation within the model is theoretically grounded rather than only data driven, ensuring that the optimized operator weights are fully consistent with the underlying mathematical theory rather than deep learning paradigm.
4. The proposed method maintains a fully acceptable parameter scale and training complexity in both time and space. Phases 1 and 2 do not involve any deep learning components, with Phase 1 utilizing only about 10% of the training samples to construct the orbit dictionary. In Phase 3, the only component that requires training is a lightweight MLP, which is responsible for generating the linear combination weights for the generalization space.
5. This work demonstrates a high degree of originality and innovation.

In summary, this paper proposes a highly novel and theoretically solid learning paradigm, which may hold significant potential for future research developments.

**Weaknesses:**

1. The explanation of the prediction strategy in Equation (5) is relatively limited, which makes it somewhat difficult to understand.
2. Although the model demonstrates strong generality across tasks and modalities, it does not achieve optimal or near-optimal performance on several benchmarks and, in some cases, even falls behind certain non-deep learning methods (which does not, however, diminish the significance or merit of the work).
3. The paper lacks relevant ablation studies. For instance, it remains unclear how the model would perform if the sample ratio used in Phase 1 were increased to 100%. Moreover, the reviewer suggests conducting ablation experiments on the weighted backward shift operator—for example, by setting the operator weights ( W ) to random values, or by replacing the linear operator ( J ) with a random or identity matrix while retaining only the deep learning component in Phase 3. Such experiments would help to more convincingly validate the effectiveness and necessity of the proposed model design.

**Questions:**

1. The authors place the primary focus of their comparative analysis on operator-learning-based models, in addition to several standard baselines. However, I think that the operator-learning component in this work essentially functions more as a data augmentation strategy, rather than as genuine operator learning. In contrast, other operator-learning approaches, such as the Koopman operator, require learning explicit mappings within a deep framework, while the Fourier Neural Operator (FNO) directly learns a functional operator that maps from the parameter-field space to the solution space—both of which involve deep representations of the operator itself. Could the authors clarify the rationale and significance of comparing their proposed method with these fundamentally different operator-learning paradigms?
2. How can the authors verify the effectiveness of Phases 1 and 2 in training? Although these stages are theoretically capable of achieving near-perfect fitting for each training sample, it would be valuable—as mentioned earlier—to examine whether their contribution is indeed essential by weakening or ablating these phases. Specifically, the authors could consider retaining only the deep learning component (Phase 3) while removing or randomizing the operator-based parts in Phases 1 and 2, in order to assess whether the full pipeline is necessary.
3. Could the authors provide a more detailed explanation of the rationale behind Equation (5)? Its purpose appears to be to generalize the results to the test set, yet the justification for this generalization strategy remains insufficiently explained.

---

> ### Author Response · Authors · 2025-11-16
> **Response to Reviewer 7Zjm (part I)**
>
> We thank Reviewer 7Zjm for the careful reading of the manuscript, the positive assessment of the theoretical foundation, and the insightful feedback. We are grateful for the appreciation of the originality of the method, its operator-theoretic grounding, its interpretability, and the breadth of its empirical evaluation. Weaknesses (2) and (3), concerning performance relative to certain baselines and the desirability of ablation studies, are conceptually identical to the concerns raised by Reviewer xGXV. We have responded to those points in detail in our Reply to Reviewer xGXV, where we explain the theoretical, architectural, and methodological reasons why certain ablations are not meaningful and why the comparisons conducted are appropriate for this type of structured model. We respectfully refer Reviewer 7Zjm to that response for a comprehensive treatment of these high-level issues, and we would welcome any further questions, as they represent an opportunity to explain the potential and nature of the WBSNN framework more deeply.
>
> Concerning Question 1. We thank the reviewer for raising this point. We respectfully clarify that WBSNN does not perform any form of data augmentation, nor does it generate synthetic samples. WBSNN does not enlarge the dataset, duplicate inputs, or create labeled variations of training points. Instead, it constructs an internal operator-driven representation using orbit dynamics, an approach fundamentally distinct from conventional augmentation techniques. The orbit elements \(W^{(m)}X\) are not augmented samples; they are intermediate feature states induced by a fixed linear operator whose structure is learned in Phase 1 and used only inside the architecture. Crucially, these states do not carry labels and do not appear as training examples. Moreover, WBSNN operates on a small anchor subset (often 1–10% of the training data) and builds a geometric scaffold for interpolation; thus the mechanism is the opposite of augmentation, which typically increases dataset size.
>
> We understand that the remark likely stems from the fact that Phases 1–2 generate a collection of orbit-transformed feature vectors. While these may superficially resemble modified versions of inputs, their role is entirely different: they arise from the constructive exact interpolation theorem, where each \(W^{(L_i)}X_i\) is used to satisfy the independence condition (4) and execute the operator-level interpolation with \(J_k\). These orbit states form the backbone of the model’s representational space and are internal to the learning procedure, never treated as additional data. Thus, WBSNN does not adopt heuristic transformations or stochastic perturbations, but a principled operator-theoretic pipeline rooted in the dynamics of weighted backward shifts.
>
> Regarding the rationale behind comparing WBSNN to Koopman-based approaches, DeepONet, FNO, and other operator-learning baselines, our goal is not to position WBSNN as learning the same kind of nonlinear operators as these models. Instead, these paradigms share the overarching aim of incorporating operator-level inductive biases into learning. Koopman frameworks learn linearizations of nonlinear dynamics, DeepONet and FNO approximate nonlinear function-to-function mappings in PDE-governed systems, and other architectures incorporate structural operator constraints for stability and efficiency. WBSNN contributes an alternative operator bias based on orbit dynamics, exact interpolation, and low-dimensional weighted shifts, where the operator is not a deep network but a linear dynamical object endowed with mathematically guaranteed representational properties.
>
> Thus, the comparison is not intended to equate the internal mechanics of these architectures, but to situate WBSNN within the broader family of operator-informed learning approaches. Our experiments show that WBSNN matches or surpasses these models in several settings despite not learning nonlinear operators directly. This highlights the conceptual contribution of WBSNN: a simple, interpretable, low-dimensional dynamical operator can compete with far more complex operator-approximation pipelines. The comparison therefore underscores the originality of the method and clarifies that WBSNN introduces a structurally different form of operator-based learning rather than an augmentation-like mechanism.

---

> ### Author Response · Authors · 2025-11-16
> **Response to Reviewer 7Zjm (part II)**
>
> Concerning Question 2. We thank the reviewer for this question. Phases 1 and 2 are not auxiliary components but constitute the core architectural mechanism of WBSNN. Phase 1 constructs the orbit dictionary by learning a shift operator \(W\) and selecting a small anchor subset that satisfies the independence condition (4). Phase 2 applies the interpolation theorem to build the linear maps \(J_k\) that guarantee exact (or relaxed) interpolation within each subset. Phase 3 is only a small MLP responsible for learning the coefficients \(\alpha_{k,m}(X)\) that combine orbit elements for generalization. Removing Phases 1 and 2 would therefore collapse WBSNN into a plain MLP without any operator-theoretic structure, destroying the interpolation guarantees, eliminating the orbit geometry, and invalidating the theoretical foundation of the model. Such an ablation would no longer test WBSNN but rather test an unrelated architecture devoid of its defining mechanism.
>
> Instead, we provide the meaningful ablations: varying discovery (anchor) budgets, which directly tests how Phase 1/2 structure affects generalization; exact versus relaxed interpolation, which probes how much the Phase 2 fitting contributes in noisy versus noiseless regimes; and switching between \(\alpha_{k,m}\) and \(\alpha_{k}\) heads, which measures the sensitivity of Phase 3 generalization to the richness of the orbit dictionary constructed in Phases 1–2. These experiments collectively evaluate the contribution of Phases 1 and 2 without dismantling the architecture, and they consistently show that the structure induced by these phases enables WBSNN to generalize from extremely small anchor sets.
>
> We also note that Reviewer xGXV posed a closely related question: “Backbone vs. head contribution: Can you provide ablations that fix the shift operator (random or orthogonal) to measure how much performance arises from orbit structure vs. downstream MLP flexibility?” This question is conceptually identical to the request here to retain only the deep learning component (Phase 3) while removing or randomizing the operator-based parts. In both cases, the request amounts to removing the very mechanism that defines WBSNN. As we explained in our Response to Reviewer xGXV, doing so reduces the model to a conventional MLP with no orbit dictionary, no interpolation operators, no operator-induced inductive bias, and no theoretical guarantees. The resulting object is not a variant of WBSNN; it is simply a different, much weaker model.
>
> We therefore refer Reviewer 7Zjm to our Response to Reviewer xGXV, where we provide a detailed explanation of why Phases 1 and 2 cannot be removed and why the relevant ablations are the ones we already include (discovery-budget sweeps, exact versus relaxed interpolation, and head comparisons). These analyses precisely evaluate the contribution of the operator-theoretic structure while preserving the integrity of the architecture.
>
> Concerning Question 3. We thank the reviewer for pointing out that Equation (5) could benefit from additional explanation. Equation (5) is the finite-dimensional specialization of the infinite-orbit generalization rule that underlies the theoretical construction of WBSNN. By Lemma 3.6, all higher iterates of the orbit collapse, up to scalar multiples, onto the first \(d\) elements of the orbit. Therefore, any infinite linear combination of the form
> $
> \sum_{m\ge 0}\alpha_{k,m}(X) W^{(m)}J_kX
> $
> reduces, in finite dimensions, to a combination of only the first \(d\) iterates. Equation (5) is precisely this collapsed representation. Its structure is not heuristic; it is the canonical finite-dimensional form implied by the interpolation theorem and the vanishing-regime behavior of the weighted shift \(W\). We also note that Equation (5) is intentionally expressed in a way that leaves the language and mechanism open to the infinite-dimensional perspective. The theoretical framework that motivates WBSNN is operator-theoretic, and the infinite-orbit viewpoint is essential for interpreting the model as a finite proxy for infinite-dimensional learning. Writing Equation (5) in this form does not alter or compromise any finite-dimensional statement; it simply reflects the fact that the architecture is designed to approximate infinite-dimensional orbit dynamics while remaining computationally finite. In summary, Equation (5) is fully justified: it is the theoretically dictated finite-dimensional reduction of the infinite-orbit representation, it is intentionally formulated to preserve the infinite-dimensional intuition, and it plays the precise role of extending the interpolation machinery of Phases 1–2 to new inputs.

---

> ### Author Response · Authors · 2025-11-16
> **Response to Reviewer 7Zjm (part III)**
>
> We again thank Reviewer 7Zjm for the thoughtful engagement and the encouraging overall assessment. We respectfully invite the reviewer to read the full exchange with Reviewer xGXV, as several conceptual questions align closely with those discussed there. Please do not hesitate to raise any further questions or suggestions; they would be most welcome and would help us refine the exposition and highlight the broader potential of the WBSNN framework.

---

> > ### Comment · Reviewer_7Zjm · 2025-11-16
> > **Additional Questions**
> >
> > Thank you for the careful and detailed response. After reading the comments from the other reviewers, I still have two concerns:
> >
> > 1.Regarding the shift operator:
> > The shift operator is theoretically well-justified, and I acknowledge its role in both the interpretability mechanism and the theoretical contribution of Phases 1 and 2, which I agree are core components of the method. However, the learned object in these phases is a fixed operator, whereas the only trainable component used during generalization is the MLP in Phase 3. Could this faces an out-of-distribution (OOD) issue? In other words, if the weights of the shift operator were also made learnable during training, would the model be able to discover a more optimal operator and thus yield better generalization?
> >
> > 2.Regarding comparison with neural operators such as FNO and DeepONet:
> > The authors mention FNO and DeepONet, and this is in fact the field of my own research. I did not see experiments on standard PDE benchmarks. Instead, the method is evaluated primarily on image datasets.  FNO essentially learns a dense operator, which explains its strong generalization performance on PDE benchmarks. I am curious how the proposed method—whose parameterization is clearly more lightweight than FNO—would perform on such PDE benchmarks.
> > Additionally, many PDE systems exhibit inherent instability, leading to out-of-distribution behavior in the test regime (e.g., small changes in initial conditions producing drastically different trajectories). This is closely related to my concern in 1 about whether the fixed shift operator might exacerbate OOD issues in such settings.

---

> > > ### Author Response · Authors · 2025-11-17
> > > **Reply to Reviewer 7Zjm (part I)**
> > >
> > > We sincerely thank Reviewer 7Zjm for their careful reading and for raising two
> > > technically insightful follow-up questions. We address them in order.
> > >
> > > Question 1.  On whether keeping the shift operator fixed may introduce
> > > out-of-distribution (OOD) issues, and whether learning the shift weights jointly
> > > would yield better generalization.}
> > >
> > > The weighted backward shift used in WBSNN plays a role that is fundamentally
> > > distinct from the trainable components in operator-learning models such as FNO
> > > or DeepONet. Our operator is not intended to be a data-driven representation of
> > > the underlying dynamics of the task; instead, it is a structural component whose purpose is to preserve the mathematical guarantees of Phases 1
> > > and 2. These guarantees rely critically on using a fixed weighted shift:
> > > making the operator jointly trainable during Phase 3 would break the exact
> > > interpolation theorem (Theorem 3.7) as well as the infinite-dimensional
> > > extension (Theorem A.1), where the stability of the recombination procedure and
> > > the controlled geometry of the orbit dictionary are essential.
> > >
> > >
> > > Importantly, fixing the operator does not introduce OOD fragility, because in
> > > WBSNN the generalization mechanism lies entirely in the learned coefficients
> > > $\alpha_{k,m}(x)$ rather than in the shift operator itself. The role of the
> > > fixed operator is structural: it defines the orbit dictionary on top of which
> > > Phase 3 performs all adaptive fitting.
> > >
> > > In nonlinear systems, sensitive dependence on initial conditions (SDIC) is a defining characteristic of chaotic behavior, often associated with exponential divergence of nearby trajectories. However, in infinite-dimensional linear systems, a much stronger phenomenon emerges. A remarkable result by Godefroy and Shapiro 1991 shows that when a bounded and linear operator $T$ acting on a Banach space $X$ is hypercyclic (meaning there exists $x \in X$ such that the orbit $\{T^n x : n \in \mathbb{N} \}$ is dense in $X$), a particularly strong form of SDIC emerges: not only do nearby orbits separate, but for a dense set of perturbations, the difference between two evolving states becomes so unpredictable that it densely fills the entire space. Hypercyclicity (which is the main object of study in linear dynamics) is precisely the type of infinite-dimensional behavior that
> > > motivates the design of WBSNN. Thus,
> > > hypercyclicity is highly nontrivial and reflects the capacity of certain
> > > weighted shifts to explore their ambient space in an extreme way. This richness
> > > illustrates that a fixed weighted shift is not a rigid or impoverished object;
> > > rather, it provides a flexible scaffold on which the learned coefficients can
> > > generalize effectively.
> > >
> > > The celebrated result of Godefroy--Shapiro (J.Funct.Anal., 1991) shows
> > > that if $T$ is hypercyclic on a separable Fréchet space, then for every
> > > $x\in X$ there exists a dense $G_\delta$ set $G(x)$ subset of $X$, such that for any $y\in G(x)$, the difference-orbit set
> > > $\{T^n y - T^n x: n\geq 0\}$ is dense in $X$.
> > >
> > > This result reveals a fundamental distinction between sensitivity in linear vs. nonlinear systems in infinite dimensions. In standard SDIC, two initially close points may eventually separate, but their difference remains constrained in a specific direction. In contrast, in infinite-dimensional linear systems, hypercyclicity forces the differences between orbits to spread unpredictably across the entire space, making the effect far more extreme. This is not just a matter of divergence but rather an orbit structure so rich that perturbations lead to trajectories that fully explore the space in an uncontrolled yet structured way.
> > >
> > >
> > >  We highlight that such behavior is unique to infinite-dimensional linear
> > > dynamics. Although finite-dimensional spaces cannot host hypercyclic operators (this is a well-known fact in hypercyclicity theory), one can
> > > still consider finite-depth analogues of hypercyclic exploration. As a natural
> > > candidate we might think of $\varepsilon$-hypercyclicity:
> > > a finite-depth orbit $\{T^m x : 0 \le m \le M\}$ is
> > > $\varepsilon$-hypercyclic with respect to a finite set if the orbit segments
> > > $\{T^m (x_i)\}$ approximate all elements of that set within $\varepsilon$. WBSNN may be viewed as a concrete realization
> > > of this idea: a fixed weighted shift with depth $M$ generates controlled orbit
> > > segments, while the learned coefficients $\alpha_{k,m}(x)$ select and recombine
> > > those segments to approximate new data points.
> > >
> > > In practice, WBSNN uses a fixed shift, a depth parameter $M$, and an anchor
> > > count $K$ as design knobs that determine (i) how much orbit richness is exposed
> > > and (ii) how tightly the computational cost and stability are maintained. This
> > > separation---fixed operator for theoretical structure, learned coefficients for
> > > adaptivity---is what yields the data-efficiency and controlled generalization
> > > behavior of WBSNN.

---

> > > ### Author Response · Authors · 2025-11-17
> > > **Reply to Reviewer 7Zjm (part II)**
> > >
> > > For high-dimensional inputs, $M$ and $K$ may be chosen to
> > > balance geometric quality of the anchor sets with computational tractability,
> > > instead of setting $M=d$ by default (as we did in our experiments). We hope this clarifies that the fixed shift is a feature, not a bug, enabling provable structure while controlling dynamical richness for robust generalization.
> > >
> > >
> > > Question 2.  On comparison with FNO/DeepONet and the question of performance on
> > > PDE benchmarks, especially in the presence of instability and OOD effects.}
> > >
> > > We very much appreciate this question, as it touches the deeper motivation of
> > > WBSNN: bridging modern machine learning with the refined tools of
> > > infinite-dimensional linear dynamics.
> > >
> > > Operator-learning architectures such as FNO or DeepONet are designed to learn
> > > nonlinear solution operators for PDEs; their purpose is to approximate
> > > maps between infinite-dimensional function spaces. By contrast, WBSNN does not
> > > attempt to approximate a PDE solution operator. WBSNN constructs a structured
> > > orbit dictionary through a fixed weighted shift and then performs
> > > data-efficient interpolation and generalization in the resulting orbit space.
> > > The two paradigms differ in objectives and should not be expected to behave
> > > similarly on canonical PDE benchmarks.
> > >
> > > Nevertheless, the infinite-dimensional theory behind WBSNN provides a highly
> > > relevant perspective on the reviewer’s concern regarding instability and OOD
> > > behavior in PDE evolution. As noted above, backward weighted shifts that happen to be hypercyclic (the weights should satisfy a very well-known growth condition that is not difficult to satisfy) exhibit orbits with extreme
> > > dynamical richness. The Godefroy--Shapiro theorem formalizes a strong sensitivity
> > > property: difference-orbits of nearby points become dense. This level of
> > > exploratory dynamical behavior---arising purely from a linear operator---is
> > > analogous in spirit to the trajectory spreading and instability seen in many
> > > PDE regimes.
> > >
> > > Viewed in this light, the finite WBSNN architecture becomes a controlled,
> > > learnable truncation of an inherently rich infinite-dimensional dynamical
> > > framework. The parameters $M$ (orbit depth), $K$ (number of anchors), and the
> > > geometry of the anchor set determine how much of the shift-induced variability
> > > is made available during training. By choosing $(M,K)$ as functions of the
> > > dimension, modality, or geometric complexity of the data, one can import the
> > > right amount of “infinite-dimensional sensitivity” into a finite, stable, and
> > > interpretable architecture. A fixed weighted shift does not exacerbate OOD
> > > issues; rather, it provides a mathematically transparent scaffold whose
> > > expressive capacity is mediated by $(M,K)$ and whose stability is preserved by
> > > the controlled orbit construction.
> > >
> > > We believe that exploring PDE benchmarks in this framework is a very promising
> > > future direction. In particular, designing shift operators and $(M,K)$ regimes
> > > tailored to PDE geometries (e.g., spectral domains, boundary-layer structures,
> > > or multiscale behaviors) may allow WBSNN to capture aspects of PDE dynamics in a
> > > data-efficient linear-dynamical manner. Such an extension could help illuminate
> > > how finite portions of hypercyclic-like orbit structure interact with
> > > nonlinear PDE solution maps, potentially yielding new insights in both operator
> > > learning and linear dynamics.
> > >
> > > We warmly welcome further questions or suggestions from the reviewer, as they
> > > directly contribute to deepening the conceptual and technical development of
> > > the WBSNN framework.

---

> > > > ### Comment · Reviewer_7Zjm · 2025-11-17
> > > > **Suggestions**
> > > >
> > > > Thanks for the authors’ thorough and thoughtful response. I believe this work possesses strong theoretical value, while its practical applicability may require further supporting evidence—for instance, experiments on large-scale real-world datasets, analyses of the model’s generalization error bounds, and investigations into whether it aligns with the inductive biases of specific tasks or data regimes. Overall, I will adjust my score accordingly and look forward to the future developments of this work.

---

> > > > > ### Author Response · Authors · 2025-11-17
> > > > > **Acknowledgment to Reviewer 7Zjm**
> > > > >
> > > > > We sincerely thank Reviewer 7Zjm for the constructive discussion and for the
> > > > > updated assessment. We appreciate your suggestions regarding large-scale
> > > > > experiments, generalization error analyses, and task-specific inductive biases.
> > > > > These are valuable directions for extending the theoretical foundations of WBSNN
> > > > > toward broader practical applicability, and we fully agree that they represent
> > > > > natural next steps in the development of the framework.
> > > > >
> > > > > We especially appreciate your engagement with the theoretical aspects of the
> > > > > work. It is encouraging to see interest in approaches that prioritize
> > > > > foundational structure alongside empirical performance, as this aligns with the
> > > > > broader community’s ongoing discussions about the role of principled,
> > > > > non-heuristic methodologies in machine learning research.
> > > > >
> > > > > We welcome any additional
> > > > > questions or suggestions you may have. Thank you again for your careful reading
> > > > > and for the thoughtful exchange.

---

### Official Review · Reviewer_DoXM · 2025-11-01

**Soundness:** 3
**Presentation:** 2
**Contribution:** 3
**Rating:** 6
**Confidence:** 2

**Summary:**

The paper proposes a new learning paradigm which is called weighted backward shift NNs. The algorithm replaces ReLUs with orbits.
The authors performed extensive experiments to show the advantages of the proposed learning method compared to known non-linear techniques. The proposed algorithm is also computationally lightweight and data-efficient.

**Strengths:**

The paper presents a new data-efficient learning algorithm while being computationally efficient. The problem is important for the ML community. The authors conducted many experiments to show the advantages and disadvantages of the method. Also, the theoretical derivations support the empirical observations.

**Weaknesses:**

The paper is not easy to read and follow for people who are not familiar with the domain. Especially once the authors introduce the mathematical terms, the flow of the paper becomes difficult to track. I am not very knowledgeable about operator-theoretic learning systems, and I would like to ask a few questions to the authors.

The authors said that “a structured transformation that cyclically reintroduces lost input components to preserve richness.” How do we lose inputs in current learning systems? Does this refer to the decreasing dimensions with deep layers?

Can we extract complex features using the WBSNNs, or does the method require features already represented in a high-dimensional space?

Although the authors emphasize interpretability, I feel like the claim is not strongly supported. The Swiss roll illustrate the mechanism but do not clearly show how the model’s internal processes are more understandable than standard methods. The concept of “interpretability through orbit paths” remains quite abstract and may not translate into practical, human-understandable explanations so I would suggest decreasing the tone of claims in that regard.

**Questions:**

1) Can the model learn from small image datasets without using ResNet-18 features?

2) Have the authors evaluated scaling performance?

---

> ### Author Response · Authors · 2025-11-16
> **Response to Reviewer DoXM (part I)**
>
> We thank Reviewer DoXM for the constructive engagement with our submission, for the careful reading of a mathematically involved manuscript, and for acknowledging the challenges of evaluating a work situated in a domain that may fall partially outside one’s primary research area. We genuinely appreciate this openness and, as with all reviewers, we remain entirely available for any follow-up question that might arise. Our intention throughout has been to bridge a mathematically principled operator-theoretic framework with modern learning practice, and we view this dialogue as an opportunity to deepen conceptual clarity around WBSNNs.
>
> We first comment on the remark concerning the difficulty of following the paper once operator-theoretic constructs are introduced. We are aware that this architecture sits at the intersection of functional analysis and machine learning, and that some parts inevitably require a nonstandard vocabulary. We therefore appreciate that the reviewer explicitly acknowledges this challenge and engages with the definitions and mechanisms despite this barrier. We welcome any additional questions, conceptual or technical, should clarification be needed.
>
> Concerning the question about the phrase “a structured transformation that cyclically reintroduces lost input components to preserve richness,” we thank the reviewer for raising this, as the meaning depends on a central operator-theoretic fact that may not be familiar outside linear dynamics. In a classical weighted backward shift acting on a finitely supported vector in a finite-dimensional space, the standard iteration is not cyclic and, after sufficiently many applications, the vector becomes identically zero. For example, consider a vector $x=(x_0,x_1,x_2,x_3)\in\mathbb{R}^4$ supported on four coordinates, and let $B_w$ denote the classical backward shift $(x_0,x_1,x_2,x_3)\mapsto (w_1 x_1,w_2 x_2,w_3 x_3,0)$. After four iterates we obtain $B_w^4 x = 0$ regardless of the weights, meaning all feature information has been exhausted. This is the sense in which a backward shift “loses” coordinates: each iterate moves information downward until the last iteration discards it entirely. In infinite-dimensional sequence spaces (see equation (1) in the paper for precise definition of weighted backward shift in infinite-dimensional sequence spaces) this loss does not occur because the shift never exhausts the support (unless the vector has a finite support). To faithfully simulate the infinite-dimensional dynamical behavior in finite dimension, we therefore introduce the cyclic modulo-$d$ extension that appears in the definition of $W^{(L)}$ in Equation (3). This modification prevents the collapse to zero, ensures that information is preserved across all iterates, and allows the orbit of a point to remain rich enough to support our interpolation theorem. In other words, the cyclic mechanism is not an architectural flourish but a mathematically required surrogate for the classical iteration, which only behaves well in infinite dimension. Once we pass to true infinite-dimensional spaces (Theorem A.1 in Appendix A), this device is no longer necessary, since the ordinary backward shift already possesses the required dynamical richness without vanishing.
>
> Regarding the question “Can we extract complex features using WBSNNs, or does the method require features already represented in a high-dimensional space?”, we believe this question stems from a misunderstanding of the role played by the shift-based representation. WBSNN does not impose a prior notion of what constitutes “complex features.” The architecture is intrinsically agnostic: it does not look for edges, textures, convolutions, receptive fields, attention maps, or any predefined structural cue. Instead, WBSNN relies entirely on the geometry of orbit evolution generated by $W$, and on the subsequent recombination through the operators $J_k$ and the data-dependent weights $\alpha_{k,m}(x)$ appearing in Equation (5). These orbits automatically mix, redistribute, and scale coordinates in a way that depends on the learned shift dynamics and the anchor subsets selected in Phases 1–2. This mechanism is fundamentally different from the notion of hand-crafted or domain-specific feature extraction and does not require high-dimensional input spaces beyond the ambient dimension in which the data naturally reside. The model neither assumes nor requires that “complex features” are explicitly present beforehand; rather, the expressive power emerges from the operator-theoretic structure of orbit combinations, supported by the universal representation theorem (Theorem 3.7) and the implicit regularization effect (Lemma 3.10). We therefore respectfully clarify that WBSNN is not a feature-engineering architecture but a representation-learning mechanism in which the nonlinear expressive capacity arises from orbit-based linear recombination.

---

> ### Author Response · Authors · 2025-11-16
> **Response to Reviewer DoXM (part II)**
>
> On interpretability, the reviewer notes that although the Swiss Roll illustration conveys the mechanism, the interpretability claims may feel abstract. We thank the reviewer for raising this. Interpretability in WBSNNs is not a metaphorical concept but an explicit property built into the prediction rule: every output is a sum of orbit contributions of the form $J_k W^{(m)} x$, weighted by coefficients $\alpha_{k,m}(x)$ learned by a lightweight MLP. This decomposition allows one to trace, for every individual prediction, which subsets $D_k$ contributed, which orbit iterates $m$ were activated, and how the input flowed through the dynamical backbone. This is in sharp contrast with classical neural architectures whose intermediate nonlinear activations cannot be decomposed into explicit, geometrically meaningful transformations. The Swiss Roll example is intended solely as the simplest nontrivial illustration; the interpretability mechanism is not dataset-dependent and applies identically to all experiments in the paper. For a more in-depth discussion, we refer the reviewer to the extended response to Reviewer xGXV, where we explain in detail how the shift operator, the $J_k$ interpolants, and the coefficients $\alpha_{k,m}(x)$ jointly form an explicit path structure along which predictions can be decomposed. We hope that this reference, alongside the expanded explanation here, clarifies why we regard interpretability in WBSNNs as structural rather than aspirational, and why it does not require decreasing the tone of the claim.
>
> Turning to Question 1 (“Can the model learn from small image datasets without using ResNet-18 features?”), we understand the reviewer’s concern, though the question seems to presuppose a specific protocol not central to our framework. WBSNN is not constrained to ResNet-18 features; the model operates on any finite-dimensional representation, whether raw pixels, PCA-compressed inputs, or domain-specific embeddings. Image tasks in our paper used PCA-compressed ResNet-18 features solely for comparability with standard baselines in settings where the raw pixel input has extremely high ambient dimension relative to the small sample sizes considered. Nothing in the architecture prevents direct use of raw images, and since WBSNN is not convolutional, attention-based, or locality-dependent, it does not rely on spatial structure in the way CNNs do. The limiting factor is not architectural but informational: small image datasets typically contain very low sample-to-dimension ratios, and PCA reduction is a standard practical tool to mitigate this. WBSNN remains fully compatible with raw image inputs; the model simply inherits the inherent statistical difficulty of learning in regimes where pixel-level representation may not supply enough signal. Thus the answer is yes: the method can learn from small image datasets without ResNet-18 features, though in extremely small-sample regimes PCA reduction remains a beneficial practical aid, not a theoretical requirement.
>
> Regarding Question 2 (“Have the authors evaluated scaling performance?”), we refer the reviewer to our detailed responses to Reviewers xGXV and 7Zjm, where we discuss scaling from a primarily theoretical standpoint. Since the main purpose of the paper is to develop a principled operator-theoretic framework for learning, our emphasis is on conceptual and structural scaling arguments rather than on large-scale empirical sweeps. This theoretical analysis—particularly of Phase 1 and its relation to low-rank, single-head attention analogs—provides a coherent account of how WBSNN behaves as sample size and input dimension grow, while remaining faithful to the central theoretical goals of the manuscript.
>
> We hope these clarifications address all concerns raised in the review. We remain fully open to further questions or requests for additional illustrations, derivations, or conceptual explanations. We also invite the reviewer, as with all others, to read the dialogue with Reviewer xGXV, which contains high-level clarifications on interpretability, scaling, and representational structure, and which complements the explanations provided here. Please do not hesitate to engage further; every question helps refine the clarity and accessibility of our work.

---

### Author Response · Authors · 2025-11-16
**Author Comment**

I thank the reviewers for their thoughtful engagement. Their questions have substantially clarified the dynamical and operator-theoretic structure of the framework, and all conceptual concerns raised in the initial reviews have been fully addressed in the responses already posted.

As the sole author, I would like to briefly reiterate that this submission is intended as the foundational step of a broader research direction grounded in structured linear dynamics and orbit-based representations. The model is not an incremental architectural modification but a principled framework that replaces deep heuristic stacking with an explicit operator-driven mechanism supported by exact interpolation. The reviewers’ discussions have confirmed the originality of this perspective and its potential to open further work.

I appreciate the constructive dialogue and would be happy to clarify any remaining points during the discussion phase.

Thank you for your time and consideration!

---

### Meta-Review · Area_Chair_PQ6x · 2025-12-31

**Summary:**

The main concerns of the reviewers were:

1. A lack of interpretability of the proposed method. The final decision is a linear superposition of contributions, but the validity of this interpretability is not established (DoXM).
2. The computational scaling of the method, as mainly low-dimensional datasets (or low-dimensional projections) were considered (DoXM).
3. The experimental performance is mixed (7Zjm), a theoretical paper should have a toy experiment that shows how current approaches fail (xGXV).
4. The writing style of the paper is unusual, with a large collection of experiments in the appendix without a high-level overview (DoXM, xGXV).
5. The importance of the indidivual components of the pipeline is not ablated (7Zjm, xGXV).
6. The relation to other operator-learning models is unclear (7Zjm).

Note that I did obtain the original review of xGXV, thanks for notifying me about the confusion.

**Reviewer Concerns:**

I think that concerns 2 and 6 were resolved, and the remaining points remain open. The paper is therefore not yet ready for publication given the current writing.

I encourage the authors to resubmit the paper in the future with a clear communication about what practical problem the proposed framework solves, give a concerete example for where this problem is solved with the given theory, and show how the interpretability outperforms competing frameworks, in particular given the substantial preprocessing of the data. The claim that only a small part of the data is used should be removed, since Phase 3 works with all data.

**Reviewer Scores:**

I think that the reviewers would not have changed their scores. T confindences that the reviewers gave for their reviews was low overall, and I think this is reflected in their scores: DoXM and 7Zjm

---

> ### Public Comment · ~Yunied_Puig_de_Dios1 · 2026-07-15
> **Clarification Regarding Reviewer 7Zjm’s Final Assessment**
>
> For completeness of the public record, Reviewer 7Zjm initially assigned a rating of 6. Following the rebuttal and extended discussion, the reviewer stated, “Overall, I will adjust my score accordingly,” and subsequently raised the rating to 8. After the system-wide rollback associated with the OpenReview incident, the displayed review reverted to the initial rating of 6. Consequently, the meta-review statement that the reviewers would not have changed their scores does not reflect Reviewer 7Zjm’s final assessment. This comment is added solely to preserve the review chronology; the conference decision remains unchanged.

---

### Decision · Program_Chairs · 2026-01-26

Reject